# Last-iterate Convergence in Extensive-Form Games

**Chung-Wei Lee**
University of Southern California
`leechung@usc.edu`

**Christian Kroer**
Columbia University
`christian.kroer@columbia.edu`

**Haipeng Luo**
University of Southern California
`haipengl@usc.edu`

## Abstract

Regret-based algorithms are highly efficient at finding approximate Nash equilibria in sequential games such as poker games. However, most regret-based algorithms, including counterfactual regret minimization (CFR) and its variants, rely on iterate averaging to achieve convergence. Inspired by recent advances on *last-iterate* convergence of optimistic algorithms in zero-sum normal-form games, we study this phenomenon in sequential games, and provide a comprehensive study of last-iterate convergence for zero-sum extensive-form games with perfect recall (EFGs), using various optimistic regret-minimization algorithms over treeplexes. This includes algorithms using the vanilla entropy or squared Euclidean norm regularizers, as well as their dilated versions which admit more efficient implementation. In contrast to CFR, we show that all of these algorithms enjoy last-iterate convergence, with some of them even converging *exponentially* fast. We also provide experiments to further support our theoretical results.

## 1 Introduction

Extensive-form games (EFGs) are an important class of games in game theory and artificial intelligence which can model imperfect information and sequential interactions. EFGs are typically solved by finding or approximating a *Nash equilibrium*. Regret-minimization algorithms are among the most popular approaches to approximate Nash equilibria. The motivation comes from a classical result which says that in two-player zero-sum games, when both players use *no-regret* algorithms, *the average strategy* converges to Nash equilibrium [Freund and Schapire, 1999, Hart and Mas-Colell, 2000, Zinkevich et al., 2007]. Counterfactual Regret Minimization (CFR) [Zinkevich et al., 2007] and it variants such as CFR+ [Tammelin, 2014] are based on this motivation.

However, due to their ergodic convergence guarantee, theoretical convergence rates of regret-minimization algorithms are typically limited to $\mathcal{O}(1/\sqrt{T})$ or $\mathcal{O}(1/T)$ for $T$ rounds, and this is also the case in practice [Brown and Sandholm, 2019a, Burch et al., 2019]. In contrast, it is known that linear convergence rates are achievable for certain other first-order algorithms [Tseng, 1995, Gilpin et al., 2008]. Additionally, the averaging procedure can create complications. It not only increases the computational and memory overhead [Bowling et al., 2015], but also makes things difficult when incorporating neural networks in the solution process, where averaging is usually not possible. Indeed, to address this issue, Brown et al. [2019] create a separate neural network to approximate the average strategy in their Deep CFR model.

Therefore, a natural idea is to design regret-minimization algorithms whose last strategy converges (we call this *last-iterate convergence*), ideally at a faster rate than the average iterate. Unfortunately, many regret minimization algorithms such as regret matching, regret matching+, and hedge, are known not

to satisfy this property empirically and theoretically even for normal-form games. Although Bowling et al. [2015] find that in Heads-Up Limit Hold'em poker the last strategy of CFR+ is better than the average strategy, and Farina et al. [2019b] observe in some experiments the last-iterate of optimistic OMD and FTRL converge fast, a theoretical understanding of this phenomenon is still absent for EFGs.

In this work, inspired by recent results on last-iterate convergence in normal-form games [Wei et al., 2021], we greatly extend the theoretical understanding of last-iterate convergence of regret-minimization algorithms in two-player zero-sum extensive-form games with perfect recall, and open up many interesting directions both in theory and practice. First, we show that any optimistic online mirror-descent algorithm instantiated with a strongly convex regularizer that is continuously differentiable on the EFG strategy space provably enjoys last-iterate convergence, while CFR with either regret matching or regret matching+ fails to converge. Moreover, for some of the optimistic algorithms, we further show explicit convergence rates. In particular, we prove that optimistic mirror descent instantiated with the 1-strongly-convex dilated entropy regularizer [Kroer et al., 2020], which we refer to as *Dilated Optimistic Multiplicative Weights Update* (DOMWU), has a *linear* convergence rate under the assumption that there is a unique Nash equilibrium; we note that this assumption was also made by Daskalakis and Panageas [2019], Wei et al. [2021] in order to achieve similar results for normal-form games.

## 2 Related Work

**Extensive-form Games**   Here we focus on work related to two-player zero-sum perfect-recall games. Although there are many game-solving techniques such as abstraction [Kroer and Sandholm, 2014, Ganzfried and Sandholm, 2014, Brown et al., 2015], endgame solving [Burch et al., 2014, Ganzfried and Sandholm, 2015], and subgame solving [Moravcik et al., 2016, Brown and Sandholm, 2019b], these methods all rely on scalable methods for computing approximate Nash equilibria. There are several classes of algorithms for computing approximate Nash equilibria, such as double-oracle methods [McMahan et al., 2003], fictitious play [Brown, 1951, Heinrich and Silver, 2016], first-order methods [Hoda et al., 2010, Kroer et al., 2020], and CFR methods [Zinkevich et al., 2007, Lanctot et al., 2009, Tammelin, 2014]. Notably, variants of the CFR approach have achieved significant success in poker games [Bowling et al., 2015, Moravčík et al., 2017, Brown and Sandholm, 2018]. Underlying the first-order and CFR approaches is the sequence-form representation [von Stengel, 1996], which allows the problem to be represented as a bilinear saddle-point problem. This leads to algorithms based on smoothing techniques and other first-order methods [Gilpin et al., 2008, Kroer et al., 2017, Gao et al., 2021], and enables CFR via the theorem connecting no-regret guarantees to Nash equilibrium.

**Online Convex Optimization and Optimistic Regret Minimization**   Online convex optimization [Zinkevich, 2003] is a framework for repeated decision making where the goal is to minimize regret. When applied to repeated two-player zero-sum games, it is known that the average strategy converges to Nash Equilibria at the rate of $\mathcal{O}(1/\sqrt{T})$ when both players apply regret-minimization algorithms whose regret grows on the order of $\mathcal{O}(\sqrt{T})$ [Freund and Schapire, 1999, Hart and Mas-Colell, 2000, Zinkevich et al., 2007]. Moreover, when the players use optimistic regret-minimization algorithms, the convergence rate is improved to $\mathcal{O}(1/T)$ [Rakhlin and Sridharan, 2013, Syrgkanis et al., 2015]. Recent works have applied optimism ideas to EFGs, such as optimistic algorithms with dilated regularizers [Kroer et al., 2020, Farina et al., 2019b], CFR-like local optimistic algorithms [Farina et al., 2019a], and optimistic CFR algorithms [Burch, 2018, Brown and Sandholm, 2019a, Farina et al., 2021a]. However, the theoretical results in all these existing papers consider the average strategy, while we are the first to consider last-iterate convergence in EFGs.

**Last-iterate Convergence in Saddle-point Optimization**   As mentioned previously, two-player zero-sum games can be formulated as saddle-point optimization problems. Saddle-point problems have recently gained a lot of attention due to their applications in machine learning, for example in generative adversarial networks [Goodfellow et al., 2014]. Basic algorithms, including gradient descent ascent and multiplicative weights update, diverge even in simple instances [Mertikopoulos et al., 2018, Bailey and Piliouras, 2018]. In contrast, their optimistic versions, optimistic gradient descent ascent (OGDA) [Daskalakis et al., 2018, Mertikopoulos et al., 2019, Wei et al., 2021] and optimistic multiplicative weights update (OMWU) [Daskalakis and Panageas, 2019, Lei et al., 2021,

Wei et al., 2021] have been shown to enjoy attractive last-iterate convergence guarantees. However, almost none of these results apply to the case of EFGs: Wei et al. [2021] show a result that implies linear convergence of vanilla OGDA in EFGs (see Corollary 5), but no results are known for vanilla OMWU or more importantly for algorithms instantiated with *dilated* regularizers which lead to fast iterate updates in EFGs. In this work we extend the existing results on normal-form games to EFGs, including the practically-important dilated regularizers.

## 3 Problem Setup

We start with some basic notation. For a vector $\boldsymbol{z}$, we use $z_i$ to denote its $i$-th coordinate and $\|\boldsymbol{z}\|_p$ to denote its $p$-norm (with $\|\boldsymbol{z}\|$ being a shorthand for $\|\boldsymbol{z}\|_2$). For a convex function $\psi$, the associated Bregman divergence is define as $D_\psi(\boldsymbol{u}, \boldsymbol{v}) = \psi(\boldsymbol{u}) - \psi(\boldsymbol{v}) - \langle \nabla\psi(\boldsymbol{v}), \boldsymbol{u} - \boldsymbol{v}\rangle$, and $\psi$ is called $\kappa$-strongly convex with respect to the $p$-norm if $D_\psi(\boldsymbol{u}, \boldsymbol{v}) \geq \frac{\kappa}{2}\|\boldsymbol{u} - \boldsymbol{v}\|_p^2$ holds for all $\boldsymbol{u}$ and $\boldsymbol{v}$ in the domain. The Kullback-Leibler divergence, which is the Bregman divergence with respect to the entropy function, is denoted by $\mathrm{KL}(\cdot, \cdot)$. Finally, we use $\Delta_P$ to denote the $(P - 1)$-dimensional simplex and $[N]$ to denote the set $\{1, 2 \ldots, N\}$ for some positive integer $N$.

**Extensive-form Games as Bilinear Saddle-point Optimization**   We consider the problem of finding a Nash equilibrium of a two-player zero-sum extensive-form game (EFG) with perfect recall. Instead of formally introducing the definition of an EFG (see Appendix A for an example), for the purpose of this work, it suffices to consider an equivalent formulation, which casts the problem as a simple bilinear saddle-point optimization [von Stengel, 1996]:

$$\min_{\boldsymbol{x}\in\mathcal{X}} \max_{y\in\mathcal{Y}} \boldsymbol{x}^\top \boldsymbol{G}\boldsymbol{y} = \max_{y\in\mathcal{Y}} \min_{\boldsymbol{x}\in\mathcal{X}} \boldsymbol{x}^\top \boldsymbol{G}\boldsymbol{y}, \tag{1}$$

where $\boldsymbol{G} \in [-1, +1]^{M\times N}$ is a known matrix, and $\mathcal{X} \subset \mathbb{R}^M$ and $\mathcal{Y} \subset \mathbb{R}^N$ are two polytopes called *treeplexes* (to be defined soon). The set of Nash equilibria is then defined as $\mathcal{Z}^* = \mathcal{X}^* \times \mathcal{Y}^*$, where $\mathcal{X}^* = \operatorname{argmin}_{\boldsymbol{x}\in\mathcal{X}} \max_{y\in\mathcal{Y}} \boldsymbol{x}^\top \boldsymbol{G}\boldsymbol{y}$ and $\mathcal{Y}^* = \operatorname{argmax}_{y\in\mathcal{Y}} \min_{\boldsymbol{x}\in\mathcal{X}} \boldsymbol{x}^\top \boldsymbol{G}\boldsymbol{y}$. Our goal is to find a point $z \in \mathcal{Z} = \mathcal{X} \times \mathcal{Y}$ that is close to the set of Nash equilibria $\mathcal{Z}^*$, and we use the Bregman divergence (of some function $\psi$) between $\boldsymbol{z}$ and the closest point in $\mathcal{Z}^*$ to measure the closeness, that is, $\min_{\boldsymbol{z}^*\in\mathcal{Z}^*} D_\psi(\boldsymbol{z}^*, \boldsymbol{z})$.

For notational convenience, we let $P = M + N$ and $F(\boldsymbol{z}) = (\boldsymbol{G}\boldsymbol{y}, -\boldsymbol{G}^\top \boldsymbol{x})$ for any $\boldsymbol{z} = (\boldsymbol{x}, \boldsymbol{y}) \in \mathcal{Z} \subset \mathbb{R}^P$. Without loss of generality, we assume $\|F(\boldsymbol{z})\|_\infty \leq 1$ for all $\boldsymbol{z} \in \mathcal{Z}$ (which can always be ensured by normalizing the entries of $\boldsymbol{G}$ accordingly).

**Treeplexes**   The structure of the EFG is implicitly captured by the treeplexes $\mathcal{X}$ and $\mathcal{Y}$, which are generalizations of simplexes that capture the sequential structure of an EFG. The formal definition is as follows. (In Appendix A, we provide more details on the connection between treeplexes and the structure of the EFG, as well as concrete examples of treeplexes for better illustrations.)

**Definition 1** (Hoda et al. [2010]). *A treeplex is recursively constructed via the following three operations:*

1. *Every probability simplex is a treeplex.*

2. *Given treeplexes $\mathcal{Z}_1, \ldots, \mathcal{Z}_K$, the Cartesian product $\mathcal{Z}_1 \times \cdots \times \mathcal{Z}_K$ is a treeplex.*

3. *(Branching) Given treeplexes $\mathcal{Z}_1 \subset \mathbb{R}^M$ and $\mathcal{Z}_2 \subset \mathbb{R}^N$, and any $i \in [M]$,*

$$\mathcal{Z}_1 \boxed{i} \mathcal{Z}_2 = \left\{ (\boldsymbol{u}, u_i \cdot \boldsymbol{v}) \in \mathbb{R}^{M+N} : \boldsymbol{u} \in \mathcal{Z}_1, \ \boldsymbol{v} \in \mathcal{Z}_2 \right\}$$

   *is a treeplex.*

By definition, a treeplex is a tree-like structure built with simplexes, which intuitively represents the tree-like decision space of a single player, and an element in the treeplex represents a strategy for the player. Let $\mathcal{H}^{\mathcal{Z}}$ denote the collection of all the simplexes in treeplex $\mathcal{Z}$, which following Definition 1 can be recursively defined as: $\mathcal{H}^{\mathcal{Z}} = \{\mathcal{Z}\}$ if $\mathcal{Z}$ is a simplex; $\mathcal{H}^{\mathcal{Z}} = \bigcup_{k=1}^K \mathcal{H}^{\mathcal{Z}_i}$ if $\mathcal{Z}$ is a Cartesian product $\mathcal{Z}_1 \times \cdots \times \mathcal{Z}_K$; and $\mathcal{H}^{\mathcal{Z}} = \mathcal{H}^{\mathcal{Z}_1} \cup \mathcal{H}^{\mathcal{Z}_2}$ if $\mathcal{Z} = \mathcal{Z}_1 \boxed{i} \mathcal{Z}_2$. In EFG terminology, $\mathcal{H}^{\mathcal{X}}$ and $\mathcal{H}^{\mathcal{Y}}$ are the collections of *information sets* for player $\boldsymbol{x}$ and player $\boldsymbol{y}$ respectively, which are the decision

points for the players, at which they select an action within the simplex. For any $h \in \mathcal{H}^{\mathcal{Z}}$, we let $\Omega_h$ denote the set of indices belonging to $h$, and for any $z \in \mathcal{Z}$, we let $z_h$ be the slice of $z$ whose indices are in $\Omega_h$. For each index $i$, we also let $h(i)$ be the simplex $i$ falls into, that is, $i \in \Omega_{h(i)}$.

In Definition 1, the last branching operation naturally introduces the concept of a *parent variable* for each $h \in \mathcal{H}^{\mathcal{Z}}$, which can again be recursively defined as: if $\mathcal{Z}$ is a simplex, then it has no parent; if $\mathcal{Z}$ is a Cartesian product $\mathcal{Z}_1 \times \cdots \times \mathcal{Z}_K$, then the parent of $h \in \mathcal{H}^{\mathcal{Z}}$ is the same as the parent of $h$ in the treeplex $\mathcal{Z}_k$ that $h$ belongs to (that is, $h \in \mathcal{H}^{\mathcal{Z}_k}$); finally, if $\mathcal{Z} = \{(u, u_i \cdot v) : u \in \mathcal{Z}_1, \, v \in \mathcal{Z}_2\}$, then for all $h \in \mathcal{H}^{\mathcal{Z}_2}$ without a parent, their parent in $\mathcal{Z}$ is $u_i$, and for all other $h$, their parents remain the same as in $\mathcal{Z}_1$ or $\mathcal{Z}_2$. We denote by $\sigma(h)$ the index of the parent variable of $h$, and let it be 0 if $h$ has no parent. For convenience, we let $z_0 = 1$ for all $z \in \mathcal{Z}$ (so that $z_{\sigma(h)}$ is always well-defined). Also define $\mathcal{H}_i = \{h \in \mathcal{H}^{\mathcal{Z}} : \sigma(h) = i\}$ to be the collection of simplexes whose parent index is $i$.

Similarly, for an index $i$, its parent index is defined as $p_i = \sigma(h(i))$, and $i$ is called a *terminal index* if it is not a parent index (that is, $i \neq p_j$ for all $j$). Finally, for an element $z \in \mathcal{Z}$ and an index $i$, we define $q_i = z_i/z_{p_i}$. In EFG terminology, $q_i$ specifies the probability of selecting action $i$ in the information set $h(i)$ according to strategy $z$.

# 4   Optimistic Regret-minimization Algorithms

There are many different algorithms for solving bilinear saddle-point problems over general constrained sets. We focus specifically on a family of regret-minimization algorithms, called Optimistic Online Mirror Descent (OOMD) [Rakhlin and Sridharan, 2013], which are known to be highly efficient. In contrast to the CFR algorithm and its variants, which minimize a local regret notion at each information set (which upper bounds global regret), the algorithms we consider explicitly minimize global regret. As our main results in the next section show, these global regret-minimization algorithms enjoy last-iterate convergence, while CFR provably diverges.

Specifically, given a step size $\eta > 0$ and a convex function $\psi$ (called a *regularizer*), OOMD sequentially performs the following update for $t = 1, 2, \ldots,$

$$x_t = \underset{x \in \mathcal{X}}{\operatorname{argmin}} \left\{ \eta \langle x, G y_{t-1} \rangle + D_\psi(x, \widehat{x}_t) \right\}, \qquad \widehat{x}_{t+1} = \underset{x \in \mathcal{X}}{\operatorname{argmin}} \left\{ \eta \langle x, G y_t \rangle + D_\psi(x, \widehat{x}_t) \right\},$$

$$y_t = \underset{y \in \mathcal{Y}}{\operatorname{argmin}} \left\{ \eta \langle y, -G^\top x_{t-1} \rangle + D_\psi(y, \widehat{y}_t) \right\}, \quad \widehat{y}_{t+1} = \underset{y \in \mathcal{Y}}{\operatorname{argmin}} \left\{ \eta \langle y, -G^\top x_t \rangle + D_\psi(y, \widehat{y}_t) \right\},$$

with $(\widehat{x}_1, \widehat{y}_1) = (x_0, y_0) \in \mathcal{Z}$ being arbitrary. Using shorthands $z_t = (x_t, y_t)$, $\widehat{z}_t = (\widehat{x}_t, \widehat{y}_t)$, $\psi(z) = \psi(x) + \psi(y)$ and recalling the notation $F(z) = (Gy, -G^\top x)$, the updates above can be compactly written as OOMD with regularizer $\psi$ over treeplex $\mathcal{Z}$:

$$z_t = \underset{z \in \mathcal{Z}}{\operatorname{argmin}} \left\{ \eta \langle z, F(z_{t-1}) \rangle + D_\psi(z, \widehat{z}_t) \right\}, \quad \widehat{z}_{t+1} = \underset{z \in \mathcal{Z}}{\operatorname{argmin}} \left\{ \eta \langle z, F(z_t) \rangle + D_\psi(z, \widehat{z}_t) \right\}. \quad (2)$$

Below, we discuss four different regularizers and their resulting algorithms (throughout, we use notations $\Phi$ for regularizers based on Euclidean norm and $\Psi$ for regularizers based on entropy).

**Vanilla Optimistic Gradient Descent Ascent (VOGDA)**   Define the vanilla squared Euclidean norm regularizer as $\Phi^{\text{van}}(z) = \frac{1}{2} \sum_i z_i^2$. We call OOMD instantiated with $\psi = \Phi^{\text{van}}$ Vanilla Optimistic Gradient Descent Ascent (VOGDA). In this case, the Bregman divergence is $D_{\Phi^{\text{van}}}(z, z') = \frac{1}{2} \|z - z'\|^2$ (by definition $\Phi^{\text{van}}$ is thus 1-strongly convex with respect to the 2-norm), and the updates simply become projected gradient descent. For VOGDA there is no closed-form for Eq. (2), since projection onto the treeplex $\mathcal{Z}$ is required. Nevertheless, the solution can still be computed in $\mathcal{O}(P^2 \log P)$ time (recall that $P$ is the dimension of $\mathcal{Z}$) [Gilpin et al., 2008].

**Vanilla Optimistic Multiplicative Weight Update (VOMWU)**   Define the vanilla entropy regularizer as $\Psi^{\text{van}}(z) = \sum_i z_i \ln z_i$. We call OOMD with $\psi = \Psi^{\text{van}}$ Vanilla Optimistic Multiplicative Weights Update (VOMWU). The Bregman divergence in this case is the generalized KL divergence: $D_{\Psi^{\text{van}}}(z, z') = \sum_i z_i \ln(z_i/z_i') - z_i + z_i'$. Although it is well-known that $\Psi^{\text{van}}$ is 1-strongly convex with respect to the 1-norm for the special case when $\mathcal{Z}$ is a simplex, this is not true generally on a treeplex. Nevertheless, it can still be shown that $\Psi^{\text{van}}$ is 1-strongly convex with respect to the 2-norm; see Appendix C.

The name "Multiplicative Weights Update" is inherited from case when $\mathcal{X}$ and $\mathcal{Y}$ are simplexes, in which case the updates in Eq. (2) have a simple multiplicative form. We emphasize, however, that in general VOMWU does not admit a closed-form update. Instead, to solve Eq. (2), one can equivalently solve a simpler dual optimization problem; see [Zimin and Neu, 2013, Proposition 1].

The two regularizers mentioned above ignore the structure of the treeplex. *Dilated Regularizers* [Hoda et al., 2010], on the other hand, take the structure into account and allow one to decompose the update into simpler updates at each information set. Specifically, given any convex function $\psi$ defined over the simplex and a weight parameter $\boldsymbol{\alpha} \in \mathbb{R}_+^{\mathcal{H}^{\mathcal{Z}}}$, the dilated version of $\psi$ defined over $\mathcal{Z}$ is:

$$\psi_{\boldsymbol{\alpha}}^{\mathrm{dil}}(\boldsymbol{z}) = \sum_{h \in \mathcal{H}^{\mathcal{Z}}} \alpha_h \cdot z_{\sigma(h)} \cdot \psi\left(\frac{\boldsymbol{z}_h}{z_{\sigma(h)}}\right). \tag{3}$$

This is well-defined since $\boldsymbol{z}_h/z_{\sigma(h)}$ is indeed a distribution within the simplex $h$ (with $q_i$ for $i \in \Omega_h$ being its entries). It can also be shown that $\psi_{\boldsymbol{\alpha}}^{\mathrm{dil}}$ is always convex in $\boldsymbol{z}$ [Hoda et al., 2010]. Intuitively, $\psi_{\boldsymbol{\alpha}}^{\mathrm{dil}}$ applies the base regularizer $\psi$ to the action distribution in each information set and then scales the value by its parent variable and the weight $\alpha_h$. By picking different base regularizers, we obtain the following two algorithms.

**Dilated Optimistic Gradient Descent Ascent (DOGDA) [Farina et al., 2019b]** Define the dilated squared Euclidean norm regularizer $\Phi_{\boldsymbol{\alpha}}^{\mathrm{dil}}$ as Eq. (3) with $\psi$ being the vanilla squared Euclidean norm $\psi(\boldsymbol{z}) = \frac{1}{2}\sum_i z_i^2$. Direct calculation shows $\Phi_{\boldsymbol{\alpha}}^{\mathrm{dil}}(\boldsymbol{z}) = \frac{1}{2}\sum_i \alpha_{h(i)} z_i q_i$. We call OOMD with regularizer $\Phi_{\boldsymbol{\alpha}}^{\mathrm{dil}}$ the Dilated Optimistic Gradient Descent Ascent algorithm (DOGDA). It is known that there exists an $\boldsymbol{\alpha}$ such that $\Phi_{\boldsymbol{\alpha}}^{\mathrm{dil}}$ is 1-strongly convex with respect to the 2-norm [Farina et al., 2019b]. Importantly, DOGDA decomposes the update Eq. (2) into simpler gradient descent-style updates at each information set, as shown below.

**Lemma 1** (Hoda et al. [2010]). *If* $\boldsymbol{z}' = \operatorname{argmin}_{\boldsymbol{z} \in \mathcal{Z}}\{\eta\langle \boldsymbol{z}, \boldsymbol{f}\rangle + D_{\Phi_{\boldsymbol{\alpha}}^{dil}}(\boldsymbol{z}, \widehat{\boldsymbol{z}})\}$, then for every $h \in \mathcal{H}^{\mathcal{Z}}$, the corresponding vector $\boldsymbol{q}'_h = \frac{\boldsymbol{z}'_h}{z'_{\sigma(h)}}$ can be computed by:*

$$\boldsymbol{q}'_h = \underset{\boldsymbol{q}_h \in \Delta_{|\Omega_h|}}{\operatorname{argmin}}\left\{\eta\langle \boldsymbol{q}_h, \boldsymbol{L}_h\rangle + \frac{\alpha_h}{2}\|\boldsymbol{q}_h - \widehat{\boldsymbol{q}}_h\|^2\right\}, \tag{4}$$

*where* $\widehat{\boldsymbol{q}}_h = \frac{\widehat{\boldsymbol{z}}_h}{\widehat{z}_{\sigma(h)}}$, $\boldsymbol{L}_h$ *is the slice of* $\boldsymbol{L}$ *whose entries are in* $\Omega_h$, *and* $\boldsymbol{L}$ *is defined through:*

$$L_i = f_i + \sum_{h \in \mathcal{H}_i}\left(\langle \boldsymbol{q}'_h, \boldsymbol{L}_h\rangle + \frac{\alpha_h}{2\eta}\|\boldsymbol{q}'_h - \widehat{\boldsymbol{q}}_h\|^2\right).$$

While the definitions of $\boldsymbol{q}'_h$ and $\boldsymbol{L}$ are seemingly recursive, one can verify that they can in fact be computed in a "bottom-up" manner, starting with the terminal indices. Although Eq. (4) still does not admit closed-form solution, it only requires projection onto a simplex, which can be solved efficiently, see e.g. [Condat, 2016]. Finally, with $\boldsymbol{q}'_h$ computed for all $h$, $\boldsymbol{z}'$ can be calculated in a "top-down" manner by definition.

**Dilated Optimistic Multiplicative Weight Update (DOMWU) [Kroer et al., 2020]** Finally, define the dilated entropy regularizer $\Psi_{\boldsymbol{\alpha}}^{\mathrm{dil}}$ as Eq. (3) with $\psi$ being the vanilla entropy $\psi(\boldsymbol{z}) = \sum_i z_i \ln z_i$. Direct calculation shows $\Psi_{\boldsymbol{\alpha}}^{\mathrm{dil}}(\boldsymbol{z}) = \sum_i \alpha_{h(i)} z_i \ln q_i$. We call OOMD with regularizer $\Psi_{\boldsymbol{\alpha}}^{\mathrm{dil}}$ the Dilated Optimistic Multiplicative Weights Update algorithm (DOMWU). Similar to DOGDA, there exists an $\boldsymbol{\alpha}$ such that $\Psi_{\boldsymbol{\alpha}}^{\mathrm{dil}}$ is 1-strongly convex with respect to the 2-norm [Kroer et al., 2020].[1] Moreover, in contrast to all the three algorithms mentioned above, the update of DOMWU has a closed-form solution:

**Lemma 2** (Hoda et al. [2010]). *Suppose* $\boldsymbol{z}' = \operatorname{argmin}_{\boldsymbol{z} \in \mathcal{Z}}\{\eta\langle \boldsymbol{z}, \boldsymbol{f}\rangle + D_{\Psi_{\boldsymbol{\alpha}}^{dil}}(\boldsymbol{z}, \widehat{\boldsymbol{z}})\}$. *Similarly to the notation* $q_i$, *define* $q'_i = z'_i/z'_{p_i}$ *and* $\widehat{q}_i = \widehat{z}_i/\widehat{z}_{p_i}$. *Then we have*

$$q'_i \propto \widehat{q}_i \exp\left(-\eta L_i/\alpha_{h(i)}\right), \text{ where } L_i = f_i - \sum_{h \in \mathcal{H}_i}\frac{\alpha_h}{\eta}\ln\left(\sum_{j \in \Omega_h}\widehat{q}_j \exp\left(-\eta L_j/\alpha_h\right)\right).$$

---

[1]Kroer et al. [2020] also show a better strong convexity result with respect to the 1-norm. We focus on the 2-norm here for consistency with our other results, but our analysis can be applied to the 1-norm case as well.

This lemma again implies that we can compute $q_i'$ bottom-up, and then $z'$ can be computed top-down. This is similar to DOGDA, except that all updates have a closed-form.

# 5  Last-iterate Convergence Results

In this section, we present our main last-iterate convergence results for the global regret-minimization algorithms discussed in Section 4. Before doing so, we point out again that the sequence produced by the well-known CFR algorithm may diverge (even if the average converges to a Nash equilibrium). Indeed, this can happen even for a simple normal-form game, as formally shown below.

**Theorem 3.** *In the rock-paper-scissors game, CFR (with some particular initialization) produces a diverging sequence.*

In fact, we empirically observe that all of CFR, CFR+ [Tammelin, 2014] (with simultaneous updates), and their optimistic versions [Farina et al., 2021a] may diverge in the rock-paper-scissors game. We introduce the algorithms and show the results in Appendix D.

On the contrary, every algorithm from the OOMD family given by Eq. (2) ensures last-iterate convergence, as long as the regularizer is strongly convex and continuously differentiable.

**Theorem 4.** *Consider the update rules in Eq. (2). Suppose that $\psi$ is $1$-strongly convex with respect to the $2$-norm and continuously differentiable on the entire domain, and $\eta \leq \frac{1}{8P}$. Then $z_t$ converges to a Nash equilibrium as $t \to \infty$.*

As mentioned, $\Phi^{\mathrm{van}}$ and $\Psi^{\mathrm{van}}$ are both $1$-strongly convex with respect to $2$-norm, so are $\Phi_{\boldsymbol{\alpha}}^{\mathrm{dil}}$ and $\Psi_{\boldsymbol{\alpha}}^{\mathrm{dil}}$ under some specific choice of $\boldsymbol{\alpha}$ (in the rest of the paper, we fix this choice of $\boldsymbol{\alpha}$). However, only $\Phi^{\mathrm{van}}$ and $\Phi_{\boldsymbol{\alpha}}^{\mathrm{dil}}$ are continuously differentiable in the entire domain. Therefore, Theorem 4 provides an asymptotic convergence result only for VOGDA and DOGDA, but not VOMWU and DOMWU. Nevertheless, below, we resort to different analyses to show a concrete last-iterate convergence rate for three of our algorithms, which is a much more challenging task.

First of all, note that [Wei et al., 2021, Theorem 5, Theorem 8] already provide a general last-iterate convergence rate for VOGDA over polytopes. Since treeplexes are polytopes, we can directly apply their results and obtain the following corollary.

**Corollary 5.** *Define $\mathrm{dist}^2(\boldsymbol{z}, \mathcal{Z}^*) = \min_{\boldsymbol{z}^* \in \mathcal{Z}^*} \|\boldsymbol{z} - \boldsymbol{z}^*\|^2$. For $\eta \leq \frac{1}{8P}$,* VOGDA *guarantees*

$$\mathrm{dist}^2(\boldsymbol{z}_t, \mathcal{Z}^*) \leq 64\mathrm{dist}^2(\widehat{\boldsymbol{z}}_1, \mathcal{Z}^*)(1 + C_1)^{-t},$$

*where $C_1 > 0$ is some constant that depends on the game and $\eta$.*

However, the results for VOMWU in [Wei et al., 2021, Theorem 3] is very specific to normal-form game (that is, when $\mathcal{X}$ and $\mathcal{Y}$ are simplexes) and thus cannot be applied here. Nevertheless, we are able to extend their analysis to get the following result.

**Theorem 6.** *If the EFG has a unique Nash equilibrium $\boldsymbol{z}^*$, then* VOMWU *with step size $\eta \leq \frac{1}{8P}$ guarantees $\frac{1}{2}\|\widehat{\boldsymbol{z}}_t - \boldsymbol{z}^*\|^2 \leq D_{\Psi^{\mathrm{van}}}(\boldsymbol{z}^*, \widehat{\boldsymbol{z}}_t) \leq \frac{C_2}{t}$, where $C_2 > 0$ is some constant depending on the game, $\widehat{\boldsymbol{z}}_1$, and $\eta$.*

We note that the uniqueness assumption is often required in the analysis of OMWU even for normal-form games [Daskalakis and Panageas, 2019, Wei et al., 2021] (although [Wei et al., 2021, Appendix A.5] provides empirical evidence to show that this may be an artifact of the analysis). Also note that for normal-form games, [Wei et al., 2021, Theorem 3] show a linear convergence rate, whereas here we only show a slower sub-linear rate, due to additional complications introduced by treeplexes (see more discussions in the next section). Whether this can be improved is left as a future direction.

On the other hand, thanks to the closed-form updates of DOMWU, we are able to show the following linear convergence rate for this algorithm.

**Theorem 7.** *If the EFG has a unique Nash equilibrium $\boldsymbol{z}^*$, then* DOMWU *with step size $\eta \leq \frac{1}{8P}$ guarantees $\frac{1}{2}\|\boldsymbol{z}_t - \boldsymbol{z}^*\|^2 \leq D_{\Psi_{\boldsymbol{\alpha}}^{\mathrm{dil}}}(\boldsymbol{z}^*, \boldsymbol{z}_t) \leq C_3(1 + C_4)^{-t}$, where $C_3, C_4 > 0$ are constants that depend on the game, $\widehat{\boldsymbol{z}}_1$, and $\eta$.*

To the best of our knowledge, this is the first last-iterate convergence result for algorithms with dilated regularizers. Unfortunately, due to technical difficulties, we were unable to prove similar results for DOGDA (see Appendix E for more discussion). We leave that as an important future direction.

## 6 Analysis Overview

In this section, we provide an overview of our analysis. It starts from the following standard one-step regret analysis of OOMD (see, for example, [Wei et al., 2021, Lemma 1]):

**Lemma 8.** *Consider the update rules in Eq. (2). Suppose that $\psi$ is 1-strongly convex with respect to the 2-norm, $\|F(z_1) - F(z_2)\| \leq L\|z_1 - z_2\|$ for all $z_1, z_2 \in \mathcal{Z}$ and some $L > 0$, and $\eta \leq \frac{1}{8L}$. Then for any $z \in \mathcal{Z}$ and any $t \geq 1$, we have*

$$\eta F(z_t)^\top (z_t - z) \leq D_\psi(z, \widehat{z}_t) - D_\psi(z, \widehat{z}_{t+1}) - D_\psi(\widehat{z}_{t+1}, z_t) - \tfrac{15}{16} D_\psi(z_t, \widehat{z}_t) + \tfrac{1}{16} D_\psi(\widehat{z}_t, z_{t-1}).$$

Note that the Lipschitz condition on $F$ holds in our case with $L = P$ since

$$\|F(z_1) - F(z_2)\| = \sqrt{\|G(y_1 - y_2)\|^2 + \|G^\top(x_1 - x_2)\|^2} \leq \sqrt{P\|z_1 - z_2\|_1^2} \leq P\|z_1 - z_2\|,$$

which is also why the step size is chosen to be $\eta \leq \frac{1}{8P}$ in all our results. In the following, we first prove Theorem 4. Then, we review the convergence analysis of [Wei et al., 2021] for OMWU in normal-form games, and finally demonstrate how to prove Theorem 6 and Theorem 7 by building upon this previous work and addressing the additional complications from EFGs.

### 6.1 Proof of Theorem 4

For any $z^* \in \mathcal{Z}^*$, by optimality of $z^*$ we have:

$$F(z_t)^\top (z_t - z^*) = x_t^\top G y_t - x_t^\top G y_t + x_t^\top G y^* - x^{*\top} G y_t \geq x^{*\top} G y^* - x^{*\top} G y^* = 0.$$

Thus, taking $z = z^*$ in Lemma 8 and rearranging, we arrive at

$$D_\psi(z^*, \widehat{z}_{t+1}) \leq D_\psi(z^*, \widehat{z}_t) - D_\psi(\widehat{z}_{t+1}, z_t) - \tfrac{15}{16} D_\psi(z_t, \widehat{z}_t) + \tfrac{1}{16} D_\psi(\widehat{z}_t, z_{t-1}).$$

Defining $\Theta_t = D_\psi(z^*, \widehat{z}_t) + \frac{1}{16} D_\psi(\widehat{z}_t, z_{t-1})$ and $\zeta_t = D_\psi(\widehat{z}_{t+1}, z_t) + D_\psi(z_t, \widehat{z}_t)$, we rewrite the inequality above as

$$\Theta_{t+1} \leq \Theta_t - \tfrac{15}{16}\zeta_t. \tag{5}$$

We remark that similar inequalities appear in [Wei et al., 2021, Eq. (3) and Eq. (4)], but here we pick $z^* \in \mathcal{Z}^*$ arbitrarily while they have to pick a particular $z^* \in \mathcal{Z}^*$ (such as the projection of $\widehat{z}_t$ onto $\mathcal{Z}^*$). Summing Eq. (5) over $t$, telescoping, and applying the strong convexity of $\psi$, we have

$$\Theta_1 \geq \Theta_1 - \Theta_T \geq \frac{15}{16} \sum_{t=1}^{T-1} \zeta_t \geq \frac{15}{32} \sum_{t=1}^{T-1} \|\widehat{z}_{t+1} - z_t\|^2 + \|z_t - \widehat{z}_t\|^2 \geq \frac{15}{64} \sum_{t=2}^{T-1} \|z_t - z_{t-1}\|^2.$$

Similar to the last inequality, we also have $\Theta_1 \geq \frac{15}{64} \sum_{t=1}^{T-1} \|\widehat{z}_{t+1} - \widehat{z}_t\|^2$ since $2\|\widehat{z}_{t+1} - z_t\|^2 + 2\|z_t - \widehat{z}_t\|^2 \geq \|\widehat{z}_{t+1} - \widehat{z}_t\|^2$. Therefore, we conclude that $\|z_t - \widehat{z}_t\|$, $\|z_{t+1} - z_t\|$, and $\|\widehat{z}_{t+1} - \widehat{z}_t\|$ all converge to 0 as $t \to \infty$. On the other hand, since the sequence $\{z_1, z_2, \dots,\}$ is bounded, by the Bolzano-Weierstrass theorem, there exists a convergent subsequence, which we denote by $\{z_{i_1}, z_{i_2}, \dots, \}$. Let $z_\infty = \lim_{\tau \to \infty} z_{i_\tau}$. By $\|\widehat{z}_t - z_t\| \to 0$ we also have $z_\infty = \lim_{\tau \to \infty} \widehat{z}_{i_\tau}$. Now, using the first-order optimality condition of $\widehat{z}_{t+1}$, we have for every $z' \in \mathcal{Z}$,

$$(\nabla\psi(\widehat{z}_{t+1}) - \nabla\psi(\widehat{z}_t) + \eta F(z_t))^\top (z' - \widehat{z}_{t+1}) \geq 0.$$

Apply this with $t = i_\tau$ for every $\tau$ and let $\tau \to \infty$, we obtain

$$\begin{aligned}
0 &\leq \lim_{\tau \to \infty} (\nabla\psi(\widehat{z}_{i_\tau+1}) - \nabla\psi(\widehat{z}_{i_\tau}) + \eta F(z_{i_\tau}))^\top (z' - \widehat{z}_{i_\tau+1}) && \text{(by the first-order optimality)} \\
&= \lim_{\tau \to \infty} \eta F(z_{i_\tau})^\top (z' - \widehat{z}_{i_\tau+1}) && \text{(by } \|\widehat{z}_{t+1} - \widehat{z}_t\| \to 0 \text{ and the continuity of } \nabla\psi) \\
&= \eta F(z_\infty)^\top (z' - z_\infty) && \text{(by } z_\infty = \lim_{\tau\to\infty} z_{i_\tau} = \lim_{\tau\to\infty} \widehat{z}_{i_\tau})
\end{aligned}$$

This implies that $z_\infty$ is a Nash equilibrium. Finally, choosing $z^* = z_\infty$ in the definition of $\Theta_t$, we have $\lim_{\tau \to \infty} \Theta_{i_\tau} = 0$ because $\lim_{\tau \to \infty} D_\psi(z_\infty, \widehat{z}_{i_\tau}) = 0$ and $\lim_{\tau \to \infty} \|\widehat{z}_{i_\tau} - z_{i_\tau - 1}\| = 0$. Additionally, by Eq. (5) we also have that $\lim_{t \to \infty} \Theta_t = 0$ as $\Theta_t$ is non-increasing. Therefore, we conclude that the entire sequence $\{z_1, z_2, \ldots\}$ converges to $z_\infty$. On the other hand, since OOMD is a regret-minimization algorithm, it is well known that the average iterate converges to a Nash equilibrium [Freund and Schapire, 1999]. Consequently, combining the two facts above implies that $z_t$ has to converge to a Nash equilibrium, which proves Theorem 4.

We remark that Lemma 8 holds for general closed convex domains as shown in [Wei et al., 2021]. Consequently, with the same argument, Theorem 4 holds more generally as long as $\mathcal{X}$ and $\mathcal{Y}$ are closed convex sets. While the argument is straightforward, we are not aware of similar results in prior works. Also note that unlike Theorem 6 and Theorem 7, Theorem 4 holds without the uniqueness assumption for VOMWU and DOMWU.

## 6.2 Review for normal-form games

To better explain our analysis and highlight its novelty, we first review the two-stage analysis of [Wei et al., 2021] for OMWU in normal-form games, a special case of our setting when $\mathcal{X}$ and $\mathcal{Y}$ are simplexes. Note that both VOMWU and DOMWU reduce to OMWU in this case. As with Theorem 6 and Theorem 7, the normal-form OMWU results assume a unique Nash equilibrium $z^*$. With this uniqueness assumption and [Mertikopoulos et al., 2018, Lemma C.4], Wei et al. [2021] show the following inequality

$$\zeta_t = D_\psi(\widehat{z}_{t+1}, z_t) + D_\psi(z_t, \widehat{z}_t) \geq C_5 \|z^* - \widehat{z}_{t+1}\|^2 \tag{6}$$

for some problem-dependent constant $C_5 > 0$, which, when combined with Eq. (5), implies that if the algorithm's current iterate is far from $z^*$, then the decrease in $\Theta_t$ is more substantial, that is, the algorithm makes more progress on approaching $z^*$. To establish a recursion, however, we need to connect the 2-norm back to the Bregman divergence (a reverse direction of strong convexity). To do so, Wei et al. [2021] argue that $\widehat{z}_{t+1,i}$ can be lower bounded by another problem-dependent constant for $i \in \text{supp}(z^*)$ [Wei et al., 2021, Lemma 19], where $\text{supp}(z^*)$ denotes the support of $z^*$. This further allows them to lower bound $\|z^* - \widehat{z}_{t+1}\|$ in terms of $D_\psi(z^*, \widehat{z}_{t+1})$ (which is just $\text{KL}(z^*, \widehat{z}_{t+1})$), leading to

$$\zeta_t = D_\psi(\widehat{z}_{t+1}, z_t) + D_\psi(z_t, \widehat{z}_t) \geq C_6 D_\psi(z^*, \widehat{z}_{t+1})^2, \tag{7}$$

for some $C_6 > 0$. On the other hand, ignoring the nonnegative term $D_\psi(z_t, \widehat{z}_t)$, we also have:

$$\zeta_t = D_\psi(\widehat{z}_{t+1}, z_t) + D_\psi(z_t, \widehat{z}_t) \geq D_\psi(\widehat{z}_{t+1}, z_t) \geq \frac{1}{4} D_\psi(\widehat{z}_{t+1}, z_t)^2, \tag{8}$$

where the last step uses the fact that $\widehat{z}_{t+1}$ and $z_t$ are close [Wei et al., 2021, Lemma 17 and Lemma 18]. Now, Eq. (7) and Eq. (8) together imply $6\zeta_t \geq 2C_6 D_\psi(z^*, \widehat{z}_{t+1})^2 + D_\psi(\widehat{z}_{t+1}, z_t)^2 \geq \min\{C_6, \frac{1}{2}\} \Theta_{t+1}^2$. Plugging this back into Eq. (5), we obtain a recursion

$$\Theta_{t+1} \leq \Theta_t - C_7 \Theta_{t+1}^2 \tag{9}$$

for some $C_7 > 0$, which then implies $\Theta_t = O(1/t)$ [Wei et al., 2021, Lemma 12]. This can be seen as the first and slower stage of the convergence behavior of the algorithm.

To further show a linear convergence rate, they argue that there exists a constant $C_8 > 0$ such that when the algorithm's iterate is reasonably close to $z^*$ in the following sense:

$$\max\{\|z^* - \widehat{z}_t\|_1, \|z^* - z_t\|_1\} \leq C_8, \tag{10}$$

the following improved version of Eq. (7) holds (note the lack of square on the right-hand side):

$$\zeta_t = D_\psi(\widehat{z}_{t+1}, z_t) + D_\psi(z_t, \widehat{z}_t) \geq C_9 D_\psi(z^*, \widehat{z}_{t+1}) \tag{11}$$

for some constant $0 < C_9 < 1$. Therefore, using the $1/t$ convergence rate derived in the first stage, there exists a $T_0$ such that when $t \geq T_0$, Eq. (10) holds and the algorithm enters the second stage. In this stage, combining Eq. (11) and the fact $\zeta_t \geq D_\psi(\widehat{z}_{t+1}, z_t)$ gives $\zeta_t \geq \frac{C_9}{2} \Theta_{t+1}$, which, together with Eq. (5) again, implies an improved recursion $\Theta_{t+1} \leq \Theta_t - \frac{15}{32} C_9 \Theta_{t+1}$. This finally shows a linear convergence rate $\Theta_t = O((1 + \rho)^{-t})$ for some problem-dependent constant $\rho > 0$.

## 6.3 Analysis of Theorem 6 and Theorem 7

While we mainly follow the steps of the analysis of [Wei et al., 2021] discussed above to prove Theorem 6 and Theorem 7, we remark that the generalization is highly non-trivial. First of all, we have to prove Eq. (6) for $\mathcal{Z}$ being a general treeplex, which does not follow [Mertikopoulos et al., 2018, Lemma C.4] since its proof is very specific to simplexes. Instead, we prove it by writing down the primal-dual linear program of Eq. (1) and applying the strict complementary slackness; see Appendix E.1 for details.

Next, to connect the 2-norm back to the Bregman divergence (which is not the simple KL divergence anymore, especially for DOMWU), we prove the following for VOMWU:

$$D_\psi(\boldsymbol{z}^*, \widehat{\boldsymbol{z}}_{t+1}) \leq \sum_{i \in \mathrm{supp}(\boldsymbol{z}^*)} \frac{(z_i^* - \widehat{z}_{t+1,i})^2}{\widehat{z}_{t+1,i}} + \sum_{i \notin \mathrm{supp}(\boldsymbol{z}^*)} \widehat{z}_{t+1,i} \leq \frac{3P\|\boldsymbol{z}^* - \widehat{\boldsymbol{z}}_{t+1}\|}{\min_{i \in \mathrm{supp}(\boldsymbol{z}^*)} \widehat{z}_{t+1,i}}, \qquad (12)$$

and the following for DOMWU:

$$\frac{D_\psi(\boldsymbol{z}^*, \widehat{\boldsymbol{z}}_{t+1})}{C'} \leq \sum_{i \in \mathrm{supp}(\boldsymbol{z}^*)} \frac{(z_i^* - \widehat{z}_{t+1,i})^2}{z_i^* \widehat{q}_{t+1,i}} + \sum_{i \notin \mathrm{supp}(\boldsymbol{z}^*)} z_{p_i}^* \widehat{q}_{t+1,i} \leq \frac{\|\boldsymbol{z}^* - \widehat{\boldsymbol{z}}_{t+1}\|_1}{\min_{i \in \mathrm{supp}(\boldsymbol{z}^*)} z_i^* \widehat{z}_{t+1,i}}, \quad (13)$$

where $C' = 4P\|\boldsymbol{\alpha}\|_\infty$ (see Appendix E.2). We then show a lower bound on $z_{t+1,i}$ and $\widehat{z}_{t+1,i}$ for all $i \in \mathrm{supp}(z^*)$, using similar arguments of [Wei et al., 2021] (see Appendix E.3). Combining Eq. (12) and Eq. (13) with Eq. (6), we have the counterpart of Eq. (7) for both VOMWU and DOMWU.

Showing Eq. (8) also involves extra complication if we follow their analysis, especially for VOMWU which does not admit a closed-form update. Instead, we find a simple workaround: by applying Eq. (5) repeatedly, we get $D_\psi(\boldsymbol{z}^*, \widehat{\boldsymbol{z}}_1) = \Theta_1 \geq \cdots \geq \Theta_{t+1} \geq \frac{1}{16} D_\psi(\widehat{\boldsymbol{z}}_{t+1}, \boldsymbol{z}_t)$, thus, $\zeta_t \geq D_\psi(\widehat{\boldsymbol{z}}_{t+1}, \boldsymbol{z}_t) \geq C_{10} D_\psi(\widehat{\boldsymbol{z}}_{t+1}, \boldsymbol{z}_t)^2$ for some $C_{10} > 0$ depending on $D_\psi(\boldsymbol{z}^*, \widehat{\boldsymbol{z}}_1)$. Combining this with Eq. (7), and applying them to Eq. (5), we obtain the recursion $\Theta_{t+1} \leq \Theta_t - C_{11}\Theta_{t+1}^2$ for some $C_{11} > 0$ similar to Eq. (9), which implies $\Theta_t = O(1/t)$ for both VOMWU and DOMWU and proves Theorem 6.

Finally, to show a linear convergence rate, we need to show the counterpart of Eq. (11), which is again more involved compared to the normal-form game case. Indeed, we are only able to do so for DOMWU by making use of its closed-form update described in Lemma 2. Specifically, observe that in Eq. (13), the term $\sum_{i \notin \mathrm{supp}(\boldsymbol{z}^*)} z_{p_i}^* \widehat{q}_{t+1,i}$ is the one that prevents us from bounding $D_\psi(\boldsymbol{z}^*, \widehat{\boldsymbol{z}}_{t+1})$ by $\mathcal{O}(\|\boldsymbol{z}^* - \widehat{\boldsymbol{z}}_{t+1}\|^2)$. Thus, our high-level idea is to argue that $\sum_{i \notin \mathrm{supp}(\boldsymbol{z}^*)} z_{p_i}^* \widehat{q}_{t+1,i}$ decreases significantly as $\widehat{\boldsymbol{z}}_t$ gets close enough to $\boldsymbol{z}^*$. To do so, we use a bottom-up induction to prove that, for any information set $h \in \mathcal{H}^{\mathcal{Z}}$, indices $i, j \in \Omega_h$ such that $i \notin \mathrm{supp}(\boldsymbol{z}^*)$ and $j \in \mathrm{supp}(\boldsymbol{z}^*)$, $\widehat{L}_{t,i}$ is significantly larger than $\widehat{L}_{t,j}$ when $\widehat{\boldsymbol{z}}_t$ is close to $\boldsymbol{z}^*$, where $\widehat{\boldsymbol{L}}_t$ is the counterpart of $\boldsymbol{L}$ in Lemma 2 when computing of $\widehat{\boldsymbol{q}}_{t+1}$. This makes sure that the term $\sum_{i \notin \mathrm{supp}(\boldsymbol{z}^*)} z_{p_i}^* \widehat{q}_{t+1,i}$ is dominated by the other term involving $i \in \mathrm{supp}(\boldsymbol{z}^*)$ in Eq. (13), which eventually helps us show Eq. (11) and the final linear convergence rate in Theorem 7. See Appendix E.5 for details.

## 7 Experiments

In this section, we experimentally evaluate the algorithms on three standard EFG benchmarks: Kuhn poker [Kuhn, 1950], Pursuit-evasion [Kroer et al., 2018], and Leduc poker [Southey et al., 2005]. The results are shown in Figure 1. Besides the optimistic algorithms, we also show two CFR-based algorithms as reference points. "CFR+" refers to CFR with alternating updates, linear averaging [Tammelin, 2014], and regret matching+ as the regret minimizer. "CFR w/ RM+" is CFR with regret matching+ and linear averaging. We provide the formal descriptions of these two algorithms in Appendix D for completeness. For the optimistic algorithms, we plot the last iterate performance. For the CFR-based algorithms, we plot the performance of the linear average of iterates (recall that the last iterate of CFR-based algorithms is not guaranteed to converge to a Nash equilibrium).

For Kuhn poker and Pursuit-evasion (on the left and in the middle of Figure 1), all of the optimistic algorithms perform much better than CFR+, and their curves are nearly straight, showing their linear last-iterate convergence on these games.

For Leduc poker, although CFR+ performs the best, we can still observe the last-iterate convergence trends of the optimistic algorithms. We remark that although VOGDA and DOMWU have linear

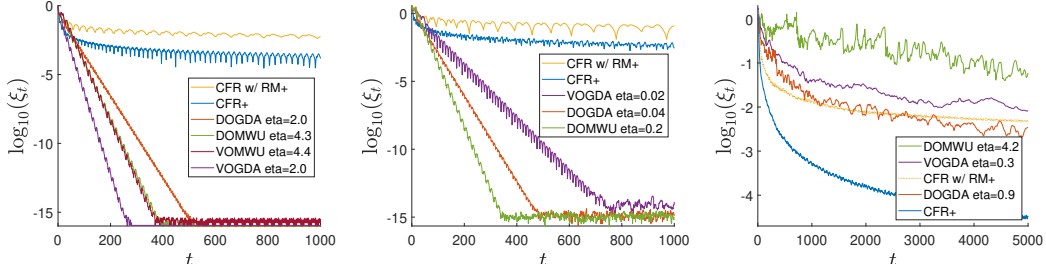

Figure 1: Experiments on Kuhn poker (left), Pursuit-evasion (middle), and Leduc poker (right). A description of each game is given in Appendix B. $\xi_t = \max_{\boldsymbol{y}} \overline{\boldsymbol{x}}_t^\top \boldsymbol{G} \boldsymbol{y} - \min_{\boldsymbol{x}} \boldsymbol{x}^\top \boldsymbol{G} \overline{\boldsymbol{y}}_t$ is the duality gap at time step $t$, where $(\overline{\boldsymbol{x}}_t, \overline{\boldsymbol{y}}_t)$ is the approximate Nash equilibrium computed by the algorithm at time $t$ (for the optimistic algorithms, $(\overline{\boldsymbol{x}}_t, \overline{\boldsymbol{y}}_t)$ is $(\boldsymbol{x}_t, \boldsymbol{y}_t)$ while for the CFR-based algorithms, $(\overline{\boldsymbol{x}}_t, \overline{\boldsymbol{y}}_t)$ is the linear average). The legend order reflects the curve order at the right-most point. Due to much higher computation overhead than all the other algorithms, we only run VOMWU on Kuhn poker, the game with the smallest size among the three games. For each optimistic algorithm, we fine-tune step size $\eta$ to get better convergence results and show its value in the legends. There is no hyperparameter for the CFR-based algorithms. All the experiments are run on CPU in a personal computer and the total computation time is less than an hour. There is no random seed and the results are all deterministic.

convergence rate in theory, the experiment on Leduc uses a step size $\eta$ which is much larger than Corollary 5 and Theorem 7 suggest, which may void the linear convergence guarantee. This is done because the theoretical step size takes too many iterations before it starts to show improvement. It is worth noting that CFR+ improves significantly when changing simultaneous updates (that is, CFR w/ RM+) to alternating updates. Analyzing alternation and combining it with optimistic algorithms is a promising direction. We provide a description of each game, more discussions, and details of the experiments in Appendix B.

## 8   Conclusions

In this work, we developed the first general last-iterate convergence results for solving EFGs.  Our paper opens up many potential future directions. The recent dilatable global entropy regularizer of Farina et al. [2021b] can likely be analyzed using techniques similar to our analysis of VOMWU and DOMWU, and it would likely lead to a linear rate as with DOMWU, due to its closed-form DOMWU-style update. Other natural questions include whether it is possible to obtain better convergence rates for VOMWU and DOGDA, whether one can remove the uniqueness assumption for VOMWU and DOMWU,  and finally whether it is possible to obtain last-iterate convergence rates for CFR-like optimistic algorithms such as those in [Farina et al., 2021a]. On the practical side, optimistic algorithms with last-iterate convergence guarantees may allow more efficient computation and better incorporation with deep learning-based game-solving approaches.

## Acknowledgments and Disclosure of Funding

CWL and HL are supported by NSF Award IIS-1943607.

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
