$$\boldsymbol{x}_t = \operatorname*{argmin}_{\boldsymbol{x} \in \mathcal{X}} \Big\{\eta \langle \boldsymbol{x}, \boldsymbol{G}\boldsymbol{y}_{t-1}\rangle + D_\psi(\boldsymbol{x}, \widehat{\boldsymbol{x}}_t)\Big\}, \qquad \widehat{\boldsymbol{x}}_{t+1} = \operatorname*{argmin}_{\boldsymbol{x} \in \mathcal{X}} \Big\{\eta \langle \boldsymbol{x}, \boldsymbol{G}\boldsymbol{y}_t\rangle + D_\psi(\boldsymbol{x}, \widehat{\boldsymbol{x}}_t)\Big\},$$

$$\boldsymbol{y}_t = \operatorname*{argmin}_{\boldsymbol{y} \in \mathcal{Y}} \Big\{\eta \langle \boldsymbol{y}, -\boldsymbol{G}^\top \boldsymbol{x}_{t-1}\rangle + D_\psi(\boldsymbol{y}, \widehat{\boldsymbol{y}}_t)\Big\}, \quad \widehat{\boldsymbol{y}}_{t+1} = \operatorname*{argmin}_{\boldsymbol{y} \in \mathcal{Y}} \Big\{\eta \langle \boldsymbol{y}, -\boldsymbol{G}^\top \boldsymbol{x}_t\rangle + D_\psi(\boldsymbol{y}, \widehat{\boldsymbol{y}}_t)\Big\},$$

with $(\widehat{\boldsymbol{x}}_1, \widehat{\boldsymbol{y}}_1) = (\boldsymbol{x}_0, \boldsymbol{y}_0) \in \mathcal{Z}$ being arbitrary. Using shorthands $\boldsymbol{z}_t = (\boldsymbol{x}_t, \boldsymbol{y}_t)$, $\widehat{\boldsymbol{z}}_t = (\widehat{\boldsymbol{x}}_t, \widehat{\boldsymbol{y}}_t)$, $\psi(\boldsymbol{z}) = \psi(\boldsymbol{x}) + \psi(\boldsymbol{y})$ and recalling the notation $F(\boldsymbol{z}) = (\boldsymbol{G}\boldsymbol{y}, -\boldsymbol{G}^\top \boldsymbol{x})$, the updates above can be compactly written as OOMD with regularizer $\psi$ over treeplex $\mathcal{Z}$:

$$\boldsymbol{z}_t = \operatorname*{argmin}_{\boldsymbol{z} \in \mathcal{Z}} \Big\{\eta \langle \boldsymbol{z}, F(\boldsymbol{z}_{t-1})\rangle + D_\psi(\boldsymbol{z}, \widehat{\boldsymbol{z}}_t)\Big\}, \ \ \widehat{\boldsymbol{z}}_{t+1} = \operatorname*{argmin}_{\boldsymbol{z} \in \mathcal{Z}} \Big\{\eta \langle \boldsymbol{z}, F(\boldsymbol{z}_t)\rangle +

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

# A   Examples of EFG and Treeplexes

In this section, we introduce Kuhn poker [Kuhn, 1950], a simple EFG as an example to introduce treeplexes and the corresponding definitions. In this game, there are three cards in the deck: King, Queen, and Jack. Both player $x$ and player $y$ are dealt one card, while the third card is put aside unseen. In the first round, player $x$ can bet or check. Then, if player $x$ bets player $y$ can choose to call or fold. If player $x$ checks then player $y$ can bet or check. Finally, if player $x$ checks and player $y$ bets, then player $x$ has a final round where they can call or fold. If neither player folded, then the player with the higher card wins the pot. If a player folded then the other player wins the pot.

We show a game tree of Kuhn poker in Figure 2. Players' imperfect information is modeled by information sets. In Figure 2, nodes with the same color belong to the same information set. A player cannot distinguish among nodes in a given information set (that belongs to this player). For example, player $x$ cannot distinguish among the blue nodes, since in both nodes, player $x$ was dealt Queen but does not know whether player $y$ was dealt Jack or King.

We can further separate the decision spaces and consider them individually on a per-player basis, which is where the concept of a treeplex arises. We show player $x$'s decision space in Figure 3. Here each circular node is an information set, which is a decision node for player $x$ where they choose an action. For example, the blue node $h_3$ corresponds to the initial state where player $x$ is dealt Queen, and player $x$ can choose to bet or check at this node. Each square node is an observation node, where player $x$ does not make a decision, but the environment or player $y$ makes decisions which determine the next decision node for player $x$. Each triangular node is a terminal node, where the game ends.

Each index of treeplex $\mathcal{X}$ corresponds to a solid, directed, edge in the figure. In other words, each index corresponds to an action in finite action set $\Omega_h$ for every $h$ in $\mathcal{H}^{\mathcal{X}}$, the set of information sets that belongs to player $x$. Indices (solid edges) are labeled from $x_1$ to $x_{12}$. More specifically, $\boldsymbol{x} \in \mathcal{X}$ if $\boldsymbol{x}$ satisfies $x_i \geq 0$ for every index $i$ and for every $h \in \mathcal{H}^{\mathcal{X}}$,

$$\sum_{i \in \Omega_h} x_i = x_{\sigma(h)},$$

where index $\sigma(h) \in \Omega_{h'}$ is the unique action such that $h$ can be reached immediately by taking $\sigma(h)$ when player $x$ is in information set $h'$. When no such action exists, that is, $h$ can be reached immediately in the beginning, we set $\sigma(h) = 0$ and $x_0 = 1$. For example, we must have $x_6 = x_7 + x_8$ and $x_5 + x_6 = x_0 = 1$ if $\boldsymbol{x} \in \mathcal{X}$. Intuitively, $x_i$ is the probability taking action $i$, given that the sequential decisions from the environment and player $y$ can lead to the information set where action $i$ is. Similarly, we show player $y$'s decision space in Figure 4, which illustrates treeplex $\mathcal{Y}$.

# B   Omitted Details of Section 7

In this section, we provide more details about the experiments.

## B.1   Additional Experiments

As we mentioned in Section 7, although VOGDA and DOMWU have linear convergence rate in theory, we use much larger step sizes $\eta$ in the Leduc poker experiment than what Corollary 5 and Theorem 7 suggest, which explains why we were not able to observe the linear convergence. Here, we rerun this experiment with a smaller step size for VOGDA and DOMWU. With more iterations, on the order of $10^5$, we observe again that they exhibit fast convergence, as shown in Figure 5.

## B.2   Description of the Games

We briefly introduce the games in the experiments. Beside the rules of the games, we show the game size by providing $M, N, |\mathcal{H}^{\mathcal{Y}}|$, and $|\mathcal{H}^{\mathcal{Y}}|$ (recall $\boldsymbol{G} = \mathbb{R}^{M \times N}$).

**Kuhn poker**   Introduced in [Kuhn, 1950], the deck for Kuhn poker contains three playing cards: King, Queen, and Jack. Each player is dealt one card, while the third card is unseen. Then a betting process proceeds. Player $x$ can check or raise, and then player $y$ can also check or raise. Player $x$ has a final round to call or fold if player $x$ checks but player $y$ raises in the previous round. The player with the higher card wins the pot. In this game, $M = N = 13$, $|\mathcal{H}^{\mathcal{X}}| = |\mathcal{H}^{\mathcal{Y}}| = 6$.

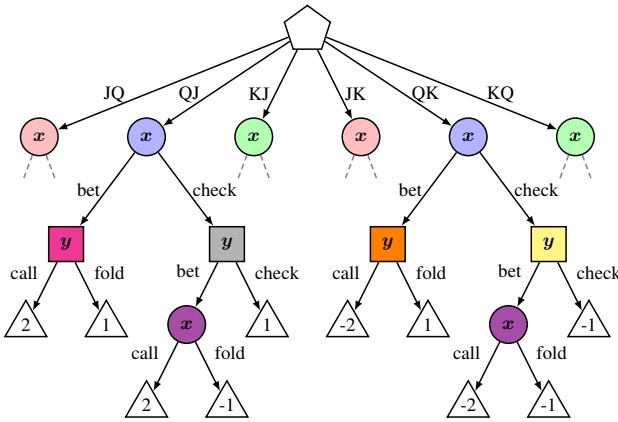

Figure 2: A game tree for Kuhn poker. The edge labeled with JQ means that player $x$ is dealt Jack and player $y$ is dealt Queen. The cases are similar at other edges. We omit branches stemming from the green and red nodes, which are similar to what we present for the blue nodes. Circular nodes are player $x$'s decision nodes, while square nodes are player $y$'s decision nodes. Triangular nodes are terminal nodes, where the values denote the utility for player $x$ (and thus the loss for player $y$). Nodes with the same color belong to the same information set, and a player cannot distinguish among nodes within the same information set, that is, they only know they are at one of these nodes but do not know which node they are at exactly.

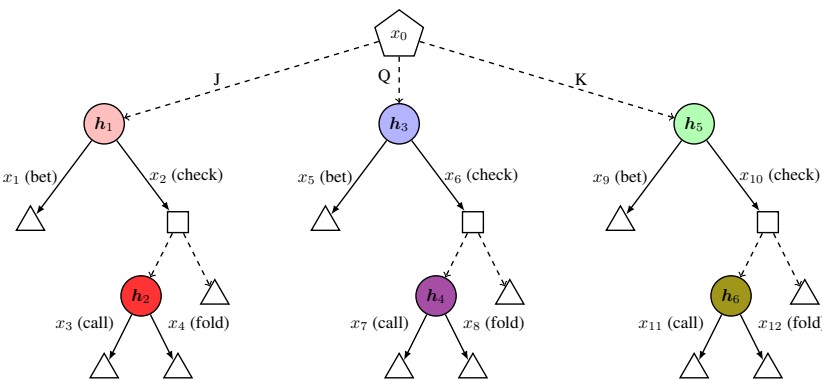

Figure 3: The decision space for player $x$ and treeplex $\mathcal{X}$. Each circular node is an information set, each square node is an observation node, and each triangular node is a termination node, where the decision process ends. Each solid edge corresponds an action and one of the indexes in treeplex $\mathcal{X}$. More specifically, we have $M = 13$, $\mathcal{H}^{\mathcal{X}} = \{\boldsymbol{h}_1, \ldots, \boldsymbol{h}_6\}$, $\Omega_{\boldsymbol{h}_i} = \{2i-1, 2i\}$ for $i = 1, \ldots, 6$, $\sigma(\boldsymbol{h}_1) = \sigma(\boldsymbol{h}_3) = \sigma(\boldsymbol{h}_5) = 0$, $\sigma(\boldsymbol{h}_2) = 2$, $\sigma(\boldsymbol{h}_4) = 6$, $\sigma(\boldsymbol{h}_6) = 10$.

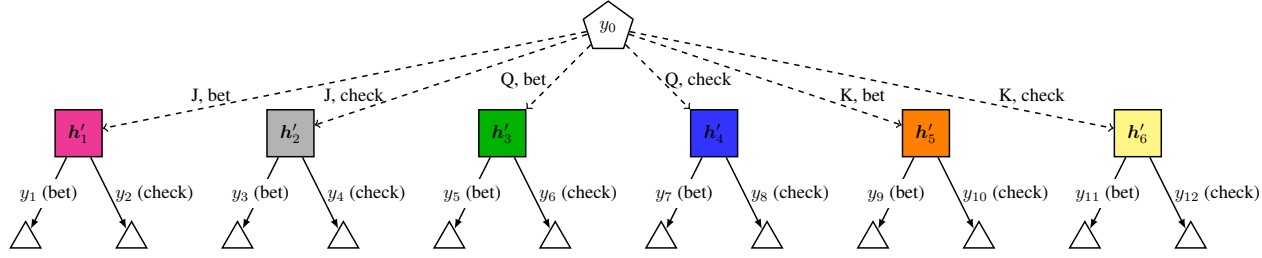

Figure 4: The decision space for player $y$ and treeplex $\mathcal{Y}$. Each square node is an information set and each triangular node is a termination node. More specifically, we have $N = 13$, $\mathcal{H}^{\mathcal{Y}} = \{\boldsymbol{h}_1', \ldots, \boldsymbol{h}_6'\}$, $\Omega_{\boldsymbol{h}_i'} = \{2i-1, 2i\}$ and $\sigma(\boldsymbol{h}_i') = 0$ for $i = 1, \ldots, 6$.

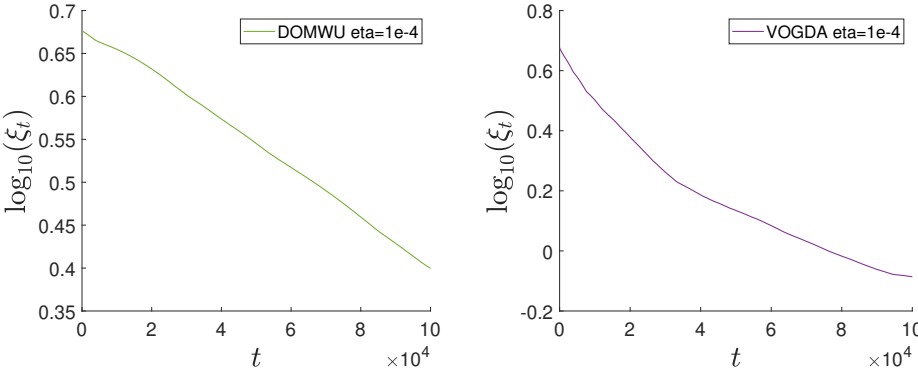

Figure 5: Experiments on Kuhn Leduc poker for DOMWU (left) and VOGDA (right) with small step sizes and more time steps.

**Pursuit-evasion** This is a search game considered in [Kroer et al., 2018]. Given a direct graph, player $x$ controls an attacker to move in the graph, while player $y$ controls two patrols, who are only allowed to move in their patrol areas. The game has 4 rounds. In each round, the attacker and the patrols act simultaneously. The attacker can move to any adjacent node or choose to wait in each round, with the final goal of going across the patrol areas and reach the goal nodes without being caught by the patrols. On the other hand, player $y$'s goal is to let one of the patrols to reach the same node as the attacker in any round. A patrol who visits a node that was previously visited by the attacker will know that the attacker was there if the attacker did not wait at that node (in order to clean up their traces). In this game, $M = 52$, $N = 2029$, $|\mathcal{H}^{\mathcal{X}}| = 34$, $|\mathcal{H}^{\mathcal{Y}}| = 348$.

**Leduc poker** Introduced in [Southey et al., 2005], the game is similar to Kuhn poker. There are total 6 cards in the deck with two Kings, two Queens, and two Jacks. Each player is dealt a private card while there is another unrevealed public card. In the first round, player $y$ bets after player $x$ bets. After that the public card will be revealed and there is another betting stage. In a showdown stage, a player who has the same rank with the public card wins. Otherwise, the player with the higher card wins. In this game, $M = N = 337$, $|\mathcal{H}^{\mathcal{X}}| = |\mathcal{H}^{\mathcal{Y}}| = 144$.

### B.3 Parameter Selection in the Dilated Regularizers

We note that for the dilated regularizers, we use the unweighted version in the experiments. This is sufficient as Lemma 9 shows there always exists an assignment $\boldsymbol{\alpha}$ such that $\psi_{\boldsymbol{\alpha}}^{\text{dil}}$ is 1-strongly convex and $\boldsymbol{\alpha} = \beta \cdot \mathbf{1}$ for some $\beta > 0$, where $\mathbf{1}$ is the all-one vector over $\mathcal{H}^{\mathcal{Z}}$. In this way, the value of $\eta$ we refer to in Figure 1 is actually $\eta/\beta$. However, as we mentioned, the final $\eta$ used in the experiments may still be larger than what the theorems suggest. Showing similar results while allowing larger $\eta$ is an interesting future direction.

**Lemma 9.** *For an assignment $\boldsymbol{\alpha}$ such that $\psi_{\boldsymbol{\alpha}}^{dil}$ is 1-strongly convex, $\psi_{\boldsymbol{\alpha}'}^{dil}$ is also 1-strongly convex where $\boldsymbol{\alpha}' = \|\boldsymbol{\alpha}\|_\infty \mathbf{1}$.*

*Proof.* Recall the definition of regularizer $\psi_{\boldsymbol{\alpha}}^{\text{dil}}(\boldsymbol{z})$ in Eq. (3). Since each term $z_{\sigma(h)} \cdot \psi\left(\frac{\boldsymbol{z}_h}{z_{\sigma(h)}}\right)$ is convex in $\boldsymbol{z}$ (thus with a non-negative Bregman divergence), $D_{\psi_{\boldsymbol{\alpha}}^{\text{dil}}}$ is an increasing function in any variable $\alpha_h$ ($h \in \mathcal{H}^{\mathcal{Z}}$). Therefore, we have

$$\frac{\|\boldsymbol{z} - \boldsymbol{z}'\|^2}{2} \le D_{\psi_{\boldsymbol{\alpha}}^{\text{dil}}}(\boldsymbol{z}, \boldsymbol{z}') \le D_{\psi_{\boldsymbol{\alpha}'}^{\text{dil}}}(\boldsymbol{z}, \boldsymbol{z}'),$$

which completes the proof. $\qquad\square$

# C Omitted Details of Section 4

When introducing VOMWU in Section 4, we mention that $\Psi^{\mathrm{van}}$ is 1-strongly convex with respect to the 2-norm. In the following, we formally show this result. Before that, we first show a technical lemma.

**Lemma 10.** *For $u, v \in [0, 1]$, the following inequalities hold:*

$$\frac{(u-v)^2}{2} \leq u \ln\left(\frac{u}{v}\right) - u + v \leq \frac{(u-v)^2}{v}. \tag{14}$$

*Proof.* Define $f(u,v) = u \ln\left(\frac{u}{v}\right) - u + v - \frac{(u-v)^2}{2}$ and $g(u,v) = \frac{(u-v)^2}{v} - u \ln\left(\frac{u}{v}\right) + u - v$. To prove the claim, it is sufficient to show that the minimum of each of the two functions is zero. Since both functions have the only critical point $(0,0)$, there is no extreme point in the interior. Also, it is straightforward to find a $(u,v)$ such that $f(u,v), g(u,v) > 0$ in the interior. Thus, it remains to check if the boundary of the domain satisfies $f(u,v), g(u,v) \geq 0$. For $u = 0$, we have

$$\frac{(0-v)^2}{2} = \frac{v^2}{2} \leq v = \frac{(0-v)^2}{v}.$$

The case for $v = 0$ is trivial. For $u = 1$, note that

$$\frac{(v-1)^2}{2} \leq v - 1 - \ln(v) \leq \frac{(v-1)^2}{v}$$

when $0 \leq v \leq 1$. For $v = 1$, we have

$$\frac{(u-1)^2}{2} \leq u \ln u - u + 1 \leq (u-1)^2$$

when $0 \leq u \leq 1$. Therefore, we conclude $f(u,v), g(u,v) \geq 0$ and this finishes the proof. $\qquad \square$

Now we are ready to give the result.

**Lemma 11.** $\Psi^{van}$ *is 1-strongly convex with respect to the 2-norm.*

*Proof.* The result follows by the first inequality of Eq. (14) and $0 \leq z_i \leq 1$ for all $\boldsymbol{z} \in \mathcal{Z}$. $\qquad \square$

# D CFR-based Algorithms and Proof of Theorem 3

In this section, we first introduce CFR, CFR+, and their optimistic versions. Then we show Theorem 3 in Appendix D.3 and their empirical last-iterate divergence in Appendix D.4.

## D.1 CFR and Its Optimistic Version

Given $P$-dimensional treeplex $\mathcal{Z}$, loss vector $\boldsymbol{\ell} \in \mathbb{R}^P$, and $\boldsymbol{z} \in \mathcal{Z}$, we recursively define the value vector $\boldsymbol{L} \in \mathbb{R}^P$, to be

$$L_i = \ell_i + \sum_{g \in \mathcal{H}_i} \langle \boldsymbol{q}_g, \boldsymbol{L}_g \rangle$$

for every index $i$ (recall that $q_i = z_i / z_{p_i}$). At time $t$, given $\boldsymbol{q}_t$, loss vector $\boldsymbol{\ell}_t$, and its value vector $\boldsymbol{L}_t$, denote $\mathrm{reg}_{t,j}^g = \langle \boldsymbol{q}_{t,g}, \boldsymbol{L}_{t,g} \rangle - L_{t,j}$, $\mathrm{Reg}_{0,j}^g = 0$, and $\mathrm{Reg}_{t,j}^g = \mathrm{Reg}_{t-1,j}^g + \mathrm{reg}_{t,j}^g$ for every simplex $g \in \mathcal{H}^{\mathcal{Z}}$ and index $j \in \Omega_g$. In the literature, *Counterfactual Regret Minimization* (CFR) Zinkevich et al. [2007] refers to the algorithm running *regret matching* on every simplex in a treeplex. Specifically, on simplex $g$ at time $t+1$, regret matching plays arbitrarily when $\sum_{j \in \Omega_g} \left[\mathrm{Reg}_{t,j}^g\right]_+ = 0$, where $[x]_+ = \max(0, x)$; otherwise, it plays

$$q_{t+1,i} = \frac{\left[\mathrm{Reg}_{t,i}^g\right]_+}{\sum_{j \in \Omega_g} \left[\mathrm{Reg}_{t,j}^g\right]_+},$$

for all $i$ in $\Omega_g$. In the two-player zero-sum setting, player $\boldsymbol{x}$ runs CFR on every simplex in treeplex $\mathcal{X}$ along with point $\boldsymbol{x}_t \in \mathcal{X}$ and loss vector $\boldsymbol{L}_t^{\boldsymbol{x}} = \boldsymbol{G}\boldsymbol{y}_t$, and player $\boldsymbol{y}$ runs CFR on every simplex in $\mathcal{Y}$ along with point $\boldsymbol{y}_t \in \mathcal{Y}$ and loss vector $\boldsymbol{L}_t^{\boldsymbol{y}} = -\boldsymbol{G}^\top \boldsymbol{x}_t$ for every time $t$.

The optimistic version of CFR Farina et al. [2021a] is running the optimistic version of regret matching on every simplex in a treeplex. Specifically, the algorithm plays arbitrarily when $\sum_{j \in \Omega_g} \left[\mathrm{Reg}_{t,j}^g + \mathrm{reg}_{t,j}^g\right]_+ = 0$; otherwise,

$$q_{t+1,i} = \frac{\left[\mathrm{Reg}_{t,i}^g + \mathrm{reg}_{t,i}^g\right]_+}{\sum_{j \in \Omega_g} \left[\mathrm{Reg}_{t,j}^g + \mathrm{reg}_{t,j}^g\right]_+}.$$

To get an approximate Nash equilibrium at time $t$, CFR and its optimistic version consider the average iterate, that is, they return

$$(\overline{\boldsymbol{x}}_t, \overline{\boldsymbol{y}}_t) = \left(\frac{1}{t}\sum_{\tau=1}^t \boldsymbol{x}_\tau, \frac{1}{t}\sum_{\tau=1}^t \boldsymbol{y}_\tau\right).$$

## D.2 CFR+ and Its Optimistic Version

To introduce CFR+, we first introduce another regret-minimization algorithm on simplex, *regret matching+*. Similar to $\mathrm{Reg}_{t,j}^g$, we define $\widehat{\mathrm{Reg}}_{0,j}^g = 0$ and

$$\widehat{\mathrm{Reg}}_{t,j}^g = \left[\widehat{\mathrm{Reg}}_{t-1,j}^g + \mathrm{reg}_{t,j}^g\right]_+,$$

for every simplex $g \in \mathcal{H}^{\mathcal{Z}}$ and index $j \in \Omega_g$. On simplex $g$ at time $t+1$, regret matching+ plays arbitrarily when $\sum_{j \in \Omega_g} \left[\widehat{\mathrm{Reg}}_{t,j}^g\right]_+ = 0$; otherwise, it plays

$$q_{t+1,i} = \frac{\left[\widehat{\mathrm{Reg}}_{t,i}^g\right]_+}{\sum_{j \in \Omega_g} \left[\widehat{\mathrm{Reg}}_{t,j}^g\right]_+}.$$

CFR+ Tammelin [2014] refers to running *regret matching+* on every simplex in a treeplex. In the two-player zero-sum setting, CFR+ usually refers to the one with *alternating updates*. Specifically, player $\boldsymbol{x}$ runs CFR+ on every simplex in $\mathcal{H}^{\mathcal{X}}$ with loss vector $\boldsymbol{L}_t^{\boldsymbol{x}} = \boldsymbol{G}\boldsymbol{y}_t$ and player $\boldsymbol{y}$ runs CFR on every simplex in $\mathcal{H}^{\mathcal{Y}}$ with loss vector $\boldsymbol{L}_t^{\boldsymbol{y}} = -\boldsymbol{G}^\top \boldsymbol{x}_{t+1}$ (note that in the case with simultaneous updates, $\boldsymbol{L}_t^{\boldsymbol{y}} = -\boldsymbol{G}^\top \boldsymbol{x}_t$).

The optimistic version of CFR+ Farina et al. [2021a] is running the optimistic version of regret matching+ on every simplex in a treeplex. Specifically, the algorithm plays arbitrarily when $\sum_{j \in \Omega_g} \left[\widehat{\mathrm{Reg}}_{t,j}^g + \mathrm{reg}_{t,j}^g\right]_+ = 0$; otherwise, it plays

$$q_{t+1,i} = \frac{\left[\widehat{\mathrm{Reg}}_{t,i}^g + \mathrm{reg}_{t,i}^g\right]_+}{\sum_{j \in \Omega_g} \left[\widehat{\mathrm{Reg}}_{t,j}^g + \mathrm{reg}_{t,j}^g\right]_+}.$$

Regarding the averaging scheme, CFR+ usually refers to the version with *linear averaging* to get an approximate Nash equilibrium at time $t$. Specifically, it returns

$$(\overline{\boldsymbol{x}}_t, \overline{\boldsymbol{y}}_t) = \left(\frac{2}{t(t+1)}\sum_{\tau=1}^t \tau \cdot \boldsymbol{x}_\tau, \frac{2}{t(t+1)}\sum_{\tau=1}^t \tau \cdot \boldsymbol{y}_\tau\right).$$

In summary, CFR+ and its optimistic version refer to running regret matching+ and optimistic regret matching+ on every simplex with alternating updates and linear averaging.

## D.3 Proof of Theorem 3

*Proof of Theorem 3.* Note that in this instance, there is only one simplex $g^{\boldsymbol{x}}$ for player $\boldsymbol{x}$ and one simplex $g^{\boldsymbol{y}}$ for player $\boldsymbol{y}$. The game matrix $\boldsymbol{G}$ of the rock-paper-scissors is

$$\boldsymbol{G} = \begin{bmatrix} 0 & -1 & 1 \\ 1 & 0 & -1 \\ -1 & 1 & 0 \end{bmatrix} = -\boldsymbol{G}^\top.$$

We consider the case when $\boldsymbol{x}_1 = \boldsymbol{y}_1$. Recall that we have $\boldsymbol{L}_t^{\boldsymbol{x}} = \boldsymbol{G}\boldsymbol{y}_t$ and $\boldsymbol{L}_t^{\boldsymbol{y}} = -\boldsymbol{G}^\top\boldsymbol{x}_t$. Therefore, we know that

$$\boldsymbol{L}_1^{\boldsymbol{x}} = \boldsymbol{G}^\top\boldsymbol{y}_1 = \boldsymbol{G}^\top\boldsymbol{x}_1 = -\boldsymbol{G}^\top\boldsymbol{x}_1 = \boldsymbol{L}_1^{\boldsymbol{y}},$$

and

$$\mathrm{reg}_{1,j}^{g^{\boldsymbol{x}}} = \boldsymbol{x}_1^\top\boldsymbol{L}_1^{\boldsymbol{x}} - L_{1,j}^{\boldsymbol{x}} = \boldsymbol{x}_1^\top\boldsymbol{G}\boldsymbol{x}_1 - L_{1,j}^{\boldsymbol{x}} = -L_{1,j}^{\boldsymbol{x}} = -L_{1,j}^{\boldsymbol{y}} = \mathrm{reg}_{1,j}^{g^{\boldsymbol{y}}}$$

for $j = 1, 2, 3$. It is not hard to see that in this case, we have $\boldsymbol{x}_2 = \boldsymbol{y}_2$, and thus $\boldsymbol{x}_t = \boldsymbol{y}_t$ for every $t$. Consequently, it is sufficient to focus on the updates of $\boldsymbol{x}_t$. For notional convenience, we write

$$\mathrm{reg}_t = \mathrm{reg}_t^{g^{\boldsymbol{x}}} = \boldsymbol{G}^\top\boldsymbol{x}_t = (x_{t,2} - x_{t,3}, x_{t,3} - x_{t,1}, x_{t,1} - x_{t,2})^\top, \tag{15}$$

$$\mathrm{Reg}_t = \mathrm{Reg}_t^{g^{\boldsymbol{x}}} = \mathrm{Reg}_{t-1}^{g^{\boldsymbol{x}}} + \mathrm{reg}_t = \sum_{\tau=1}^t \mathrm{reg}_\tau, \tag{16}$$

and thus

$$x_{t+1,j} = \frac{\left[\mathrm{Reg}_{t,j}\right]_+}{\left[\mathrm{Reg}_{t,1}\right]_+ + \left[\mathrm{reg}_{t,2}\right]_+ + \left[\mathrm{Reg}_{t,3}\right]_+} \tag{17}$$

for $j = 1, 2, 3$. We call distribution $\boldsymbol{x}_t$ *imbalanced* if there exists a permutation $\lambda$ of $\{1, 2, 3\}$ such that

$$x_{t,\lambda(1)} \geq x_{t,\lambda(2)} \geq 0 = x_{t,\lambda(3)}.$$

We prove that if $\boldsymbol{x}_1$ is imbalanced, then every $\boldsymbol{x}_t$ is imbalanced. Suppose at some time $t$, $\boldsymbol{x}_t$ is imbalanced and the corresponding $\lambda$ is the identity without loss of generality, that is,

$$x_{t,1} \geq x_{t,2} \geq 0 = x_{t,3}. \tag{18}$$

In this case, we know that $\mathrm{Reg}_{t-1,1} > 0$. By Eq. (18) and Eq. (15), we also know that $\mathrm{reg}_{t,1} \geq 0$, $\mathrm{reg}_{t,3} \geq 0$, $\mathrm{reg}_{t,2} < 0$, and $\mathrm{Reg}_{t,1} > 0$. Moreover, by Eq. (15) and Eq. (16), we can get

$$\mathrm{reg}_{t,1} + \mathrm{reg}_{t,2} + \mathrm{reg}_{t,3} = 0, \ \mathrm{Reg}_{t,1} + \mathrm{Reg}_{t,2} + \mathrm{Reg}_{t,3} = 0.$$

Therefore, we have $\mathrm{Reg}_{t,2} + \mathrm{Reg}_{t,3} < 0$, which means that at least one of $x_{t+1,2}$ and $x_{t+1,3}$ is zero. Moreover, we have

$$\mathrm{Reg}_{t,1} = \mathrm{Reg}_{t-1,1} + \mathrm{reg}_{t,1} \geq \mathrm{Reg}_{t-1,1} \geq \mathrm{Reg}_{t-1,2} > \mathrm{Reg}_{t-1,2} + \mathrm{reg}_{t,2} = \mathrm{Reg}_{t,2},$$

where the second inequality follows from $x_{t,1} \geq x_{t,2}$ and Eq. (17). The inequalities above imply that $x_{t+1,1} > x_{t+1,2}$. Thus, we get one of the following three situations continues to hold at time $t + 1$:

$$x_{t+1,1} \geq x_{t+1,2} \geq 0 = x_{t+1,3}, \tag{19}$$
$$x_{t+1,1} \geq x_{t+1,3} \geq 0 = x_{t+1,2}, \tag{20}$$
$$x_{t+1,3} \geq x_{t+1,1} \geq 0 = x_{t+1,2}. \tag{21}$$

If Eq. (19) holds at time $t + 1$, the same argument implies one of the three arguments above continues to hold; otherwise, if at some time step $\tau > t$, Eq. (20) holds, that is,

$$x_{\tau,1} \geq x_{\tau,3} \geq 0 = x_{\tau,2}. \tag{22}$$

Similarly, we know that $\mathrm{reg}_{\tau,1} \leq 0, \mathrm{reg}_{\tau,2} \leq 0$ and $\mathrm{reg}_{\tau,3} \geq 0$, and $x_{\tau+1,2} = 0$. Thus, we get either Eq. (22) continues to hold at time $\tau + 1$ or

$$x_{\tau+1,3} \geq x_{\tau+1,1} \geq 0 = x_{\tau+1,2}$$

holds, which is exactly the same permutation in Eq. (21). With similar arguments, we know that for every imbalanced distribution, either the same permutation holds in the next round, or it transits to another imbalanced distribution with another permutation. Note that the average iterate of the sequence $\{\boldsymbol{x}_t\}_t$ converges to the uniform distribution as CFR is a no-regret algorithm, so $\boldsymbol{x}_t$ never converges to any imbalanced distribution. Therefore, we conclude that $\boldsymbol{x}_t$ diverges if the algorithm starts from $x_1 = y_1$ being an arbitrary imbalanced distribution. □

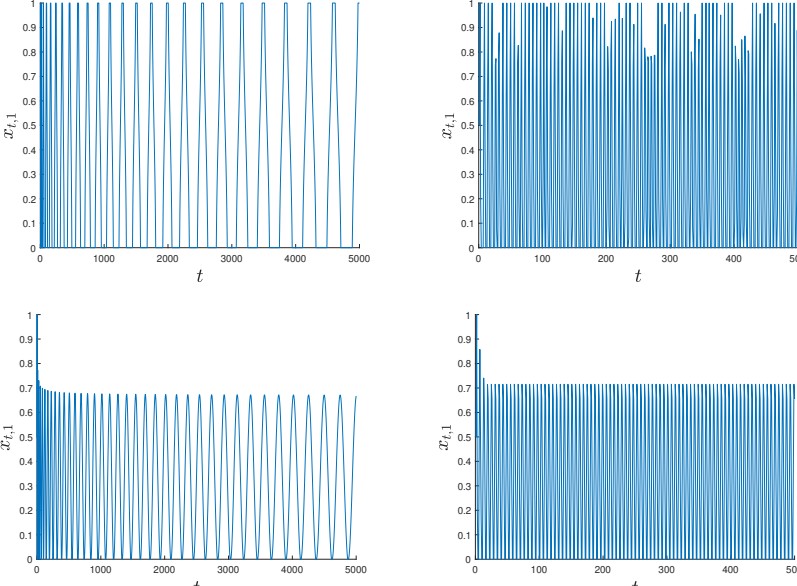

Figure 6: Last-iterate divergence of CFR, CFR+ with simultaneous updates, and their optimistic versions in the rock-paper-scissors game. The first row is the CFR algorithms while the second row is the CFR+ algorithms. For both rows, we put the vanilla version on the left and the optimistic version on the right. All algorithms start with $\boldsymbol{x}_1 = \boldsymbol{y}_1 = (1, 0, 0)^\top$.

### D.4  Experiments

Besides Theorem 3, we empirically observe divergence of CFR, CFR+ with simultaneous updates, and their optimistic versions in the rock-paper-scissors game. The results are shown in Figure 6. We remark that in these experiments, we consider CFR+ with simultaneous updates instead of the more commonly used ones (alternating updates). In fact, we observe that with alternating updates, the optimistic CFR+ empirically has last-iterate convergence in some matrix games. As all of our theoretical results are for simultaneous updates, the theoretical justification of this observation is beyond the scope of this paper, but it is an interesting direction for future works.

### E  Proofs of Theorem 6 and Theorem 7

In this section, we show the proof of Theorem 6 in Appendix E.4 and the proof of Theorem 7 in Appendix E.5. We generally follow the outline in Section 6.3. We discuss the technical difficulty to get a convergence rate for DOGDA in Appendix E.6. Throughout the section, we assume that $\boldsymbol{G}$ has a unique Nash equilibrium $\boldsymbol{z}^* = (\boldsymbol{x}^*, \boldsymbol{y}^*)$ (call it the uniqueness assumption). For the sake of analysis, we equivalently define $\Psi_{\boldsymbol{\alpha}}^{\mathrm{dil}}$ as

$$\Psi_{\boldsymbol{\alpha}}^{\mathrm{dil}}(\boldsymbol{z}) = \sum_i \alpha_{h(i)} z_i \ln \frac{z_i}{\sum_{j \in \Omega_{h(i)}} z_j}.$$

Note that under this definition, we have

$$\frac{\partial \Psi_{\boldsymbol{\alpha}}^{\mathrm{dil}}(\boldsymbol{z})}{\partial z_i} = \alpha_{h(i)} \left[ \ln \left( \frac{z_i}{\sum_{j \in \Omega_{h(i)}} z_j} \right) + 1 - \sum_{k \in \Omega_{h(i)}} \frac{z_k}{\sum_{j \in \Omega_{h(i)}} z_j} \right] = \alpha_{h(i)} \ln q_i, \qquad (23)$$

and

$$D_{\Psi_{\boldsymbol{\alpha}}^{\mathrm{dil}}}(\boldsymbol{z}, \boldsymbol{z}') = \sum_i \alpha_{h(i)} \left[ z_i \ln(q_i) - z_i' \ln(q_i') - (z_i - z_i') \ln(q_i') \right] = \sum_i \alpha_{h(i)} z_i \ln \frac{q_i}{q_i'}. \qquad (24)$$

### E.1 Strict Complementary Slackness and Proof of Eq. (6) in EFGs

In this subsection, we prove Eq. (6), which is restated in Lemma 12.

**Lemma 12.** *Under the uniqueness assumption, for both* VOMWU *and* DOMWU*, there exists some constant $C_{12} > 0$ that depends on the game, $\eta$, and $\widehat{z}_1$ such that for any $t$,*

$$\zeta_t = D_\psi(\widehat{z}_{t+1}, z_t) + D_\psi(z_t, \widehat{z}_t) \geq C_{12} \|z^* - \widehat{z}_{t+1}\|^2. \tag{25}$$

Before proving this lemma, we show some useful properties of EFGs. The first one is the strict complementary slackness.

#### E.1.1 Strict Complementary Slackness

We formulate the minimax problem in Eq. (1) as a linear program. This is a standard procedure in the literature (see, for example, [Nisan et al., 2007, Section 3.11]), but we show its derivation here for completeness. Based on Definition 1, we have $x \in \mathcal{X}$ if $x$ satisfies $x_i \geq 0$ for every $i$ and

$$x_0 = 1, \qquad \sum_{i \in \Omega_h} x_i = x_{\sigma(h)}, \ \forall h \in \mathcal{H}^{\mathcal{X}}. \tag{26}$$

We can write the constraints in Eq. (26) using a matrix $A \in \mathbb{R}^{(|\mathcal{H}^{\mathcal{X}}|+1) \times M}$ such that $x \in \mathcal{X}$ if and only if $Ax = e$, $x \geq 0$, where all entries in $e$ are zero except for $e_0 = 1$, which corresponds to the constraint $x_0 = e_0 = 1$. Similarly, for player $y$, we have the constraint matrix $B$ and vector $f$. Consequently, we write the best response of $x$ to a fixed $y$ to as the following linear program:

$$\min_x x^\top (Gy), \text{ subject to } Ax = e, \ x \geq 0.$$

The dual of this linear program is

$$\max_V e^\top V, \text{ subject to } A^\top V \leq Gy,$$

where $V \in \mathbb{R}^{|\mathcal{H}^{\mathcal{X}}|+1}$. Recall that all entries in $e$ are zero except for $e_0 = 1$ so the objective $e^\top V = V_0$. On the other hand, player $y$ tries to maximize $x^\top Gy$, that is, maximize $V_0$ by the strong duality. Therefore, every $y^* \in \mathcal{Y}^*$ is a solution to the following maximin problem

$$\max_{V,y} V_0, \text{ subject to } A^\top V \leq Gy, \ By = f, \ y \geq 0, \tag{27}$$

which is also a linear program. The dual of this linear problem is

$$\min_{U,x} U_0, \text{ subject to } B^\top U \geq G^\top x, \ Ax = e, \ x \geq 0. \tag{28}$$

It is also not hard to see that every $x^* \in \mathcal{X}$ is a solution to this dual. We conclude that Eq. (27) and Eq. (28) are primal-dual linear programs of the minmax problem in Eq. (1). Given an optimal solution pair $x^*, y^*$ along with $V^*, U^*$, by the complementary slackness, we have some slackness variables $w^* \in \mathbb{R}^M, s^* \in \mathbb{R}^N$ such that $x^* \odot w^* = 0$, $y^* \odot s^* = 0$ ($\odot$ denotes the element-wise product), and

$$A^\top V^* - Gy^* + w^* = 0, \quad B^\top U^* - G^\top x^* - s^* = 0, \quad w^*, s^* \geq 0. \tag{29}$$

Thus, Eq. (29) implies that for every index $i$ of player $x$,

$$V^*_{h(i)} - \sum_{g \in \mathcal{H}_i} V^*_g - (Gy^*)_i = (A^\top V^*)_i - (Gy^*)_i = (A^\top V^* - Gy^*)_i = -w^*_i. \tag{30}$$

Additionally, the *strict complementary slackness* (see, for example, [Vanderbei et al., 2015, Theorem 10.7]) ensures that there exists an optimal solution $(x^*, y^*)$ such that

$$x^* + w^* > 0, \quad y^* + s^* > 0. \tag{31}$$

Under the uniqueness assumption, the strict complementary slackness must hold for the unique optimal solution $(x^*, y^*)$. Therefore, when $x^*_i > 0$, we have $w^*_i = 0$, which means that Eq. (30) implies

$$V^*_h = (Gy^*)_i + \sum_{g \in \mathcal{H}_i} V^*_g, \quad \forall h \in \mathcal{H}^{\mathcal{X}}, \ i \in \Omega_h, \ x^*_i > 0; \tag{32}$$

otherwise, if $x_i^* = 0$ and $w_i^* > 0$, by Eq. (30), we have

$$V_h^* < (\boldsymbol{G}\boldsymbol{y}^*)_i + \sum_{g \in \mathcal{H}_i} V_g^*, \quad \forall h \in \mathcal{H}^{\mathcal{X}}, \ i \in \Omega_h, \ x_i^* = 0. \tag{33}$$

Note that for every terminal index $i$, $\mathcal{H}_i$ is empty and we set the term $\sum_{g \in \mathcal{H}_i} V_g^*$ zero correspondingly. The case is similar for $\boldsymbol{U}^*$ and $\boldsymbol{G}^\top \boldsymbol{x}^*$. We summarize this result as the following lemma.

**Lemma 13.** *Under the uniqueness assumption, we have*

$$\sum_{g \in \mathcal{H}_i} V_g^* + (\boldsymbol{G}\boldsymbol{y}^*)_i = V_{h(i)}^* \qquad\qquad \forall i \in supp(\boldsymbol{x}^*),$$

$$\sum_{g \in \mathcal{H}_i} V_g^* + (\boldsymbol{G}\boldsymbol{y}^*)_i > V_{h(i)}^* \qquad\qquad \forall i \notin supp(\boldsymbol{x}^*),$$

$$\sum_{g \in \mathcal{H}_j} U_g^* + (\boldsymbol{G}^\top \boldsymbol{x}^*)_j = U_{h(j)}^* \qquad\qquad \forall j \in supp(\boldsymbol{y}^*),$$

$$\sum_{g \in \mathcal{H}_j} U_g^* + (\boldsymbol{G}^\top \boldsymbol{x}^*)_j < U_{h(j)}^* \qquad\qquad \forall j \notin supp(\boldsymbol{y}^*).$$

### E.1.2 Some Problem-dependent Constants

After introducing the strict complementary slackness, we are ready to introduce some problem-dependent constants. Note that by Lemma 13, we have the following constant $\xi > 0$.

**Definition 2.** *Under the uniqueness assumption, we define*

$$\xi \triangleq \min \left\{ \min_{i \notin supp(\boldsymbol{x}^*)} \sum_{g \in \mathcal{H}_i} V_g^* + (\boldsymbol{G}\boldsymbol{y}^*)_i - V_{h(i)}^*, \ \min_{j \notin supp(\boldsymbol{y}^*)} U_{h(j)}^* - (\boldsymbol{G}^\top \boldsymbol{x}^*)_j - \sum_{g \in \mathcal{H}_j} U_g^* \right\} \in (0, 2].$$

Note that $\xi \leq 2$ follows from the fact that for any information set $h \in \mathcal{H}^{\mathcal{X}}$, indices $i, j \in \Omega_h$ such that $i \notin supp(\boldsymbol{x}^*)$ and $j \in supp(\boldsymbol{x}^*)$, by Lemma 13, we have

$$\xi \leq \sum_{g \in \mathcal{H}_i} V_g^* + (\boldsymbol{G}\boldsymbol{y}^*)_i - V_h^* = (\boldsymbol{G}\boldsymbol{y}^*)_i - (\boldsymbol{G}\boldsymbol{y}^*)_j \leq 2.$$

Below, we define $\mathcal{V}^*(\mathcal{Z}) = \mathcal{V}^*(\mathcal{X}) \times \mathcal{V}^*(\mathcal{Y})$, where

$$\mathcal{V}^*(\mathcal{X}) \triangleq \{\boldsymbol{x} : \boldsymbol{x} \in \mathcal{X}, \ supp(\boldsymbol{x}) \subseteq supp(\boldsymbol{x}^*)\}$$

and

$$\mathcal{V}^*(\mathcal{Y}) \triangleq \{\boldsymbol{y} : \boldsymbol{y} \in \mathcal{Y}, \ supp(\boldsymbol{y}) \subseteq supp(\boldsymbol{y}^*)\}.$$

**Definition 3.**

$$c_x \triangleq \min_{\boldsymbol{x} \in \mathcal{X} \setminus \{\boldsymbol{x}^*\}} \max_{\boldsymbol{y} \in \mathcal{V}^*(\mathcal{Y})} \frac{(\boldsymbol{x} - \boldsymbol{x}^*)^\top \boldsymbol{G}\boldsymbol{y}}{\|\boldsymbol{x} - \boldsymbol{x}^*\|_1}, \qquad c_y \triangleq \min_{\boldsymbol{y} \in \mathcal{Y} \setminus \{\boldsymbol{y}^*\}} \max_{\boldsymbol{x} \in \mathcal{V}^*(\mathcal{X})} \frac{\boldsymbol{x}^\top \boldsymbol{G}(\boldsymbol{y}^* - \boldsymbol{y})}{\|\boldsymbol{y}^* - \boldsymbol{y}\|_1}.$$

The following lemma shows that $c_x$ and $c_y$ are well-defined even though the outer minimization is over an open set. The proof generally follows [Wei et al., 2021, Lemma 14] but requires the results derived in Appendix E.1.1.

**Lemma 14.** $c_x$ *and* $c_y$ *are well-defined, and* $0 < c_x, c_y \leq 1$.

*Proof.* We first show $c_x$ and $c_y$ are well-defined. To simplify the notations, we define $x_{\min}^* \triangleq \min_{i \in supp(\boldsymbol{x}^*)} x_i^*$ and $\mathcal{X}' \triangleq \{\boldsymbol{x} : \boldsymbol{x} \in \mathcal{X}, \ \|\boldsymbol{x} - \boldsymbol{x}^*\|_1 \geq x_{\min}^*\}$, and define $y_{\min}^*$ and $\mathcal{Y}'$ similarly. We will show that

$$c_x = \min_{\boldsymbol{x} \in \mathcal{X}'} \max_{\boldsymbol{y} \in \mathcal{V}^*(\mathcal{Y})} \frac{(\boldsymbol{x} - \boldsymbol{x}^*)^\top \boldsymbol{G}\boldsymbol{y}}{\|\boldsymbol{x} - \boldsymbol{x}^*\|_1}, \quad c_y = \min_{\boldsymbol{y} \in \mathcal{Y}'} \max_{\boldsymbol{x} \in \mathcal{V}^*(\mathcal{X})} \frac{\boldsymbol{x}^\top \boldsymbol{G}(\boldsymbol{y}^* - \boldsymbol{y})}{\|\boldsymbol{y}^* - \boldsymbol{y}\|_1},$$

which are well-defined as the outer minimization is now over a closed set. To prove the equality for $c_x$, it suffices to show that for any $\boldsymbol{x} \in \mathcal{X}$ such that $\boldsymbol{x} \neq \boldsymbol{x}^*$ and $\|\boldsymbol{x} - \boldsymbol{x}^*\|_1 < x_{\min}^*$, there exists $\boldsymbol{x}' \in \mathcal{X}$ such that $\|\boldsymbol{x}' - \boldsymbol{x}^*\|_1 = x_{\min}^*$ and

$$\frac{(\boldsymbol{x} - \boldsymbol{x}^*)^\top \boldsymbol{G} \boldsymbol{y}}{\|\boldsymbol{x} - \boldsymbol{x}^*\|_1} = \frac{(\boldsymbol{x}' - \boldsymbol{x}^*)^\top \boldsymbol{G} \boldsymbol{y}}{\|\boldsymbol{x}' - \boldsymbol{x}^*\|_1}, \ \forall \boldsymbol{y}. \tag{34}$$

In fact, we can simply choose $\boldsymbol{x}' = \boldsymbol{x}^* + (\boldsymbol{x} - \boldsymbol{x}^*) \cdot \frac{x_{\min}^*}{\|\boldsymbol{x} - \boldsymbol{x}^*\|_1}$. We first argue that $\boldsymbol{x}'$ is still in $\mathcal{X}$. For each index $j$, if $x_j - x_j^* \geq 0$, we surely have $x_j' \geq x_j^* + 0 \geq 0$; otherwise, $x_j^* > x_j \geq 0$ and thus $j \in \operatorname{supp}(\boldsymbol{x}^*)$ and $x_j^* \geq x_{\min}^*$, which implies $x_j' \geq x_j^* - |x_j - x_j^*| \cdot \frac{x_{\min}^*}{\|\boldsymbol{x} - \boldsymbol{x}^*\|_1} \geq x_j^* - x_{\min}^* \geq 0$. In addition, for any $h \in \mathcal{H}^{\mathcal{X}}$,

$$\sum_{j \in \Omega_h} x_j' = \frac{x_{\min}^*}{\|\boldsymbol{x} - \boldsymbol{x}^*\|_1} \cdot \sum_{j \in \Omega_h} x_j + \left(1 - \frac{x_{\min}^*}{\|\boldsymbol{x} - \boldsymbol{x}^*\|_1}\right) \sum_{j \in \Omega_h} x_j^*$$

$$= \frac{x_{\min}^*}{\|\boldsymbol{x} - \boldsymbol{x}^*\|_1} \cdot x_{\sigma(h)} + \left(1 - \frac{x_{\min}^*}{\|\boldsymbol{x} - \boldsymbol{x}^*\|_1}\right) x_{\sigma(h)}^*$$

$$= x_{\sigma(h)}'.$$

Therefore, we conclude $\boldsymbol{x}' \in \mathcal{X}$. Moreover, according to the definition of $\boldsymbol{x}'$, $\|\boldsymbol{x}' - \boldsymbol{x}^*\|_1 = x_{\min}^*$ holds. Also, since $\boldsymbol{x}^* - \boldsymbol{x}$ and $\boldsymbol{x}^* - \boldsymbol{x}'$ are parallel vectors, Eq. (34) is satisfied. The arguments above show that the $c_x$ in Definition 3 is a well-defined real number. The case of $c_y$ is similar.

Now we show $0 < c_x, c_y \leq 1$. The fact that $c_x, c_y \leq 1$ is a direct consequence of the definitions. Below, we use contradiction to prove that $c_y > 0$. First, if $c_y < 0$, then there exists $\boldsymbol{y} \neq \boldsymbol{y}^*$ such that $\boldsymbol{x}^{*\top} \boldsymbol{G} \boldsymbol{y}^* < \boldsymbol{x}^{*\top} \boldsymbol{G} \boldsymbol{y}$. This contradicts with the fact that $(\boldsymbol{x}^*, \boldsymbol{y}^*)$ is the equilibrium.

On the other hand, if $c_y = 0$, then there is some $\boldsymbol{y} \neq \boldsymbol{y}^*$ such that

$$\max_{\boldsymbol{x} \in \mathcal{V}^*(\mathcal{X})} \boldsymbol{x}^\top \boldsymbol{G} (\boldsymbol{y}^* - \boldsymbol{y}) = 0. \tag{35}$$

Consider the point $\boldsymbol{y}' = \boldsymbol{y}^* + \frac{\xi}{2N}(\boldsymbol{y} - \boldsymbol{y}^*)$ (recall the definition of $\xi$ in Definition 2 and that $0 < \xi \leq 2$), which is a convex combination of $\boldsymbol{y}^*$ and $\boldsymbol{y}$, and hence $\boldsymbol{y}' \in \mathcal{Y}$. Then, for any $\boldsymbol{x} \in \mathcal{X}$,

$$\boldsymbol{x}^\top \boldsymbol{G} \boldsymbol{y}' = \sum_{i \notin \operatorname{supp}(\boldsymbol{x}^*)} x_i (\boldsymbol{G} \boldsymbol{y}')_i + \sum_{i \in \operatorname{supp}(\boldsymbol{x}^*)} x_i (\boldsymbol{G} \boldsymbol{y}')_i$$

$$\geq \sum_{i \notin \operatorname{supp}(\boldsymbol{x}^*)} \left(x_i (\boldsymbol{G} \boldsymbol{y}^*)_i - x_i \|\boldsymbol{y}' - \boldsymbol{y}^*\|_1\right) + \sum_{i \in \operatorname{supp}(\boldsymbol{x}^*)} \left(\frac{\xi}{2} \cdot x_i (\boldsymbol{G}(\boldsymbol{y} - \boldsymbol{y}^*))_i + x_i (\boldsymbol{G} \boldsymbol{y}^*)_i\right)$$

$$\text{(using } G_{ij} \in [-1, 1] \text{ for the first part and } \boldsymbol{y}' = \boldsymbol{y}^* + \frac{\xi}{2N}(\boldsymbol{y} - \boldsymbol{y}^*) \text{ for the second)}$$

$$\geq \sum_{i \notin \operatorname{supp}(\boldsymbol{x}^*)} \left(x_i (\boldsymbol{G} \boldsymbol{y}^*)_i - x_i \|\boldsymbol{y}' - \boldsymbol{y}^*\|_1\right) + \sum_{i \in \operatorname{supp}(\boldsymbol{x}^*)} x_i \left(V_{h(i)}^* - \sum_{g \in \mathcal{H}_i} V_g^*\right),$$

where the last inequality is due to Eq. (35) and Lemma 13. We continue to bound the terms above, which are bounded by

$$\geq \sum_{i \notin \operatorname{supp}(\boldsymbol{x}^*)} \left(x_i \left((\boldsymbol{G} \boldsymbol{y}^*)_i - \xi\right)\right) + \sum_{i \in \operatorname{supp}(\boldsymbol{x}^*)} x_i \left(V_{h(i)}^* - \sum_{g \in \mathcal{H}_i} V_g^*\right)$$

$$\text{(using } \boldsymbol{y}' - \boldsymbol{y}^* = \frac{\xi}{2N}(\boldsymbol{y} - \boldsymbol{y}^*) \text{ and } \|\boldsymbol{y} - \boldsymbol{y}^*\|_1 \leq 2N)$$

$$\geq \sum_{i \notin \operatorname{supp}(\boldsymbol{x}^*)} x_i \left(V_{h(i)}^* - \sum_{g \in \mathcal{H}_i} V_g^*\right) + \sum_{i \in \operatorname{supp}(\boldsymbol{x}^*)} x_i \left(V_{h(i)}^* - \sum_{g \in \mathcal{H}_i} V_g^*\right) \quad \text{(by the definition of } \xi\text{)}$$

$$= \sum_i x_i \left(V_{h(i)}^* - \sum_{g \in \mathcal{H}_i} V_g^*\right).$$

The last term can be writen as the matrix form $\boldsymbol{x}^\top \boldsymbol{A}^\top \boldsymbol{V}^* = \mathbf{e}^\top \boldsymbol{V}^* = V_0^*$. This shows that $\min_{\boldsymbol{x}\in\mathcal{X}} \boldsymbol{x}^\top \boldsymbol{G} \boldsymbol{y}' \geq V_0^*$, that is, $\boldsymbol{y}' \neq \boldsymbol{y}^*$ is also a maximin point, contradicting the uniqueness assumption. Therefore, $c_y > 0$ has to hold, and so does $c_x > 0$ by the same argument. $\qquad\square$

We continue to define more constants in the following.

**Definition 4.** *Define constants* $z_{\min} \triangleq \min_i \widehat{z}_{1,i} \in (0,1]$,

$$\epsilon_{van} \triangleq \min_{j \in supp(\boldsymbol{z}^*)} \exp\left(-\frac{P(1 + \ln(1/z_{\min}))}{z_j^*}\right) \in (0,1),$$

$$\epsilon_{dil} \triangleq \min_{j \in supp(\boldsymbol{z}^*)} \left\{ z_j^* \cdot \exp\left(-\frac{\|\alpha\|_\infty P^2 \ln(1/z_{\min})}{z_j^*}\right) \cdot \left(\frac{3}{4}\right)^P \right\} \in (0,1).$$

For all $i \in \mathrm{supp}(\boldsymbol{z}^*)$, we will show that $\epsilon_{\text{van}}$ is a lower bound of $\widehat{z}_{t,i}$ for VOMWU, while $\epsilon_{\text{dil}}$ is a lower bound of $\widehat{z}_{t,i}$ and $z_{t,i}$ for DOMWU. We show the results in Lemma 20 and defer the proof there.

### E.1.3 Proof of Lemma 12

We are almost ready to prove Lemma 12. Before that, we first show the following auxiliary lemma.

**Lemma 15.** *For any* $\boldsymbol{z} \in \mathcal{Z}$, *we have*

$$\max_{\boldsymbol{z}' \in \mathcal{V}^*(\mathcal{Z})} F(\boldsymbol{z})^\top (\boldsymbol{z} - \boldsymbol{z}') \geq C \|\boldsymbol{z}^* - \boldsymbol{z}\|_1,$$

*for* $C = \min\{c_x, c_y\} \in (0,1]$.

*Proof.* Recall that $V_0^* = \boldsymbol{x}^{*\top} \boldsymbol{G} \boldsymbol{y}^*$ is the game value and note that

$$
\begin{aligned}
\max_{\boldsymbol{z}' \in \mathcal{V}^*(\mathcal{Z})} F(\boldsymbol{z})^\top (\boldsymbol{z} - \boldsymbol{z}') &= \max_{\boldsymbol{z}' \in \mathcal{V}^*(\mathcal{Z})} (\boldsymbol{x} - \boldsymbol{x}')^\top \boldsymbol{G} \boldsymbol{y} + \boldsymbol{x}^\top \boldsymbol{G}(\boldsymbol{y}' - \boldsymbol{y}) = \max_{\boldsymbol{z}' \in \mathcal{V}^*(\mathcal{Z})} -\boldsymbol{x}'^\top \boldsymbol{G} \boldsymbol{y} + \boldsymbol{x}^\top \boldsymbol{G} \boldsymbol{y}' \\
&= \max_{\boldsymbol{x}' \in \mathcal{V}^*(\mathcal{X})} \left( V_0^* - \boldsymbol{x}'^\top \boldsymbol{G} \boldsymbol{y} \right) + \max_{\boldsymbol{y}' \in \mathcal{V}^*(\mathcal{Y})} \left( \boldsymbol{x}^\top \boldsymbol{G} \boldsymbol{y}' - V_0^* \right) \\
&= \max_{\boldsymbol{x}' \in \mathcal{V}^*(\mathcal{X})} \boldsymbol{x}'^\top \boldsymbol{G}(\boldsymbol{y}^* - \boldsymbol{y}) + \max_{\boldsymbol{y}' \in \mathcal{V}^*(\mathcal{Y})} (\boldsymbol{x} - \boldsymbol{x}^*)^\top \boldsymbol{G} \boldsymbol{y}' \\
&\geq c_y \|\boldsymbol{y}^* - \boldsymbol{y}\|_1 + c_x \|\boldsymbol{x}^* - \boldsymbol{x}\|_1 \qquad\qquad \text{(by Definition 3)} \\
&\geq \min\{c_x, c_y\} \|\boldsymbol{z}^* - \boldsymbol{z}\|_1,
\end{aligned}
$$

where the third equality is due to Eq. (29), $\boldsymbol{x}' \in \mathcal{V}^*(\mathcal{X})$, and

$$V_0^* = \mathbf{e}^\top \boldsymbol{V}^* = (\boldsymbol{x}'^\top \boldsymbol{A}^\top) \boldsymbol{V}^* = \boldsymbol{x}'^\top (\boldsymbol{A}^\top \boldsymbol{V}^*) = \boldsymbol{x}'^\top (\boldsymbol{G} \boldsymbol{y}^*);$$

the case for $\boldsymbol{y}'$ is similar. This completes the proof. $\qquad\square$

Now we show the proof of Lemma 12.

*Proof of Lemma 12.* Below we consider any $\boldsymbol{z}' \in \mathcal{Z}$ such that $\mathrm{supp}(\boldsymbol{z}') \subseteq \mathrm{supp}(\boldsymbol{z}^*)$, that is, $\boldsymbol{z}' \in \mathcal{V}^*(\mathcal{Z})$. Considering Eq. (2), and using the first-order optimality condition of $\widehat{\boldsymbol{z}}_{t+1}$, we have

$$(\nabla\psi(\widehat{\boldsymbol{z}}_{t+1}) - \nabla\psi(\widehat{\boldsymbol{z}}_t) + \eta F(\boldsymbol{z}_t))^\top (\boldsymbol{z}' - \widehat{\boldsymbol{z}}_{t+1}) \geq 0. \tag{36}$$

Rearranging the terms and we get

$$\eta F(\boldsymbol{z}_t)^\top (\widehat{\boldsymbol{z}}_{t+1} - \boldsymbol{z}') \leq (\nabla\psi(\widehat{\boldsymbol{z}}_{t+1}) - \nabla\psi(\widehat{\boldsymbol{z}}_t))^\top (\boldsymbol{z}' - \widehat{\boldsymbol{z}}_{t+1}). \tag{37}$$

The left hand side of Eq. (37) is lower bounded as

$$
\begin{aligned}
\eta F(\boldsymbol{z}_t)^\top (\widehat{\boldsymbol{z}}_{t+1} - \boldsymbol{z}') &= \eta F(\widehat{\boldsymbol{z}}_{t+1})^\top (\widehat{\boldsymbol{z}}_{t+1} - \boldsymbol{z}') + \eta (F(\boldsymbol{z}_t) - F(\widehat{\boldsymbol{z}}_{t+1}))^\top (\widehat{\boldsymbol{z}}_{t+1} - \boldsymbol{z}') \\
&\geq \eta F(\widehat{\boldsymbol{z}}_{t+1})^\top (\widehat{\boldsymbol{z}}_{t+1} - \boldsymbol{z}') - \eta \|F(\boldsymbol{z}_t) - F(\widehat{\boldsymbol{z}}_{t+1})\|_\infty \|\widehat{\boldsymbol{z}}_{t+1} - \boldsymbol{z}'\|_1 \\
&\geq \eta F(\widehat{\boldsymbol{z}}_{t+1})^\top (\widehat{\boldsymbol{z}}_{t+1} - \boldsymbol{z}') - 2P\eta \|\boldsymbol{z}_t - \widehat{\boldsymbol{z}}_{t+1}\|_1 \\
&\geq \eta F(\widehat{\boldsymbol{z}}_{t+1})^\top (\widehat{\boldsymbol{z}}_{t+1} - \boldsymbol{z}') - \frac{1}{4} \|\boldsymbol{z}_t - \widehat{\boldsymbol{z}}_{t+1}\|_1; \qquad (\eta \leq 1/(8P))
\end{aligned}
$$

When $\psi = \Psi^{\mathrm{van}}$, we have

$$(\nabla\Psi^{\mathrm{van}}(\widehat{z}_{t+1}) - \nabla\Psi^{\mathrm{van}}(\widehat{z}_t))_i = (1 + \ln\widehat{z}_{t+1,i}) - (1 + \ln\widehat{z}_{t,i}) = \ln\frac{\widehat{z}_{t+1,i}}{\widehat{z}_{t,i}}. \tag{38}$$

On the other hand, when $\psi = \Psi^{\mathrm{dil}}_{\boldsymbol{\alpha}}$, by Eq. (23), we have

$$\left(\nabla\Psi^{\mathrm{dil}}_{\boldsymbol{\alpha}}(\widehat{z}_{t+1}) - \nabla\Psi^{\mathrm{dil}}_{\boldsymbol{\alpha}}(\widehat{z}_t)\right)_i = \alpha_i \ln\frac{\widehat{q}_{t+1,i}}{\widehat{q}_{t,i}}. \tag{39}$$

Therefore, the right hand side of Eq. (37) for VOMWU is upper bounded by

$$\begin{aligned}
&(\nabla\Psi^{\mathrm{van}}(\widehat{z}_{t+1}) - \nabla\Psi^{\mathrm{van}}(\widehat{z}_t))^\top (z' - \widehat{z}_{t+1}) \\
&= \sum_i (z_i' - \widehat{z}_{t+1,i}) \ln\frac{\widehat{z}_{t+1,i}}{\widehat{z}_{t,i}} && \text{(Eq. (38))} \\
&\le \|\widehat{z}_{t+1} - \widehat{z}_t\|_1 - D_{\Psi^{\mathrm{van}}}(\widehat{z}_{t+1}, \widehat{z}_t) + \sum_{i\in\mathrm{supp}(z^*)} z_i' \ln\frac{\widehat{z}_{t+1,i}}{\widehat{z}_{t,i}} && (\mathrm{supp}(z') \subseteq \mathrm{supp}(z^*)) \\
&\le \|\widehat{z}_{t+1} - \widehat{z}_t\|_1 + \sum_{i\in\mathrm{supp}(z^*)} \left|\ln\frac{\widehat{z}_{t+1,i}}{\widehat{z}_{t,i}}\right| \\
&\le \|\widehat{z}_{t+1} - \widehat{z}_t\|_1 + \sum_{i\in\mathrm{supp}(z^*)} \ln\left(1 + \frac{|\widehat{z}_{t+1,i} - \widehat{z}_{t,i}|}{\min\{\widehat{z}_{t+1,i}, \widehat{z}_{t,i}\}}\right) \\
&\le \|\widehat{z}_{t+1} - \widehat{z}_t\|_1 + \sum_{i\in\mathrm{supp}(z^*)} \frac{|\widehat{z}_{t+1,i} - \widehat{z}_{t,i}|}{\min\{\widehat{z}_{t+1,i}, \widehat{z}_{t,i}\}} && (\ln(1+a) \le a) \\
&\le \frac{2}{\epsilon_{\mathrm{van}}} \|\widehat{z}_{t+1} - \widehat{z}_t\|_1; && (\text{Lemma 20 and } \epsilon_{\mathrm{van}} \le 1)
\end{aligned}$$

on the other hand, the right hand side of Eq. (37) for DOMWU is upper bounded by

$$\begin{aligned}
&\left(\nabla\Psi^{\mathrm{dil}}_{\boldsymbol{\alpha}}(\widehat{z}_{t+1}) - \nabla\Psi^{\mathrm{dil}}_{\boldsymbol{\alpha}}(\widehat{z}_t)\right)^\top (z' - \widehat{z}_{t+1}) \\
&= \sum_i \alpha_i (z_i' - \widehat{z}_{t+1,i}) \ln\frac{\widehat{q}_{t+1,i}}{\widehat{q}_{t,i}} && \text{(Eq. (39))} \\
&= \sum_i \alpha_i z_i' \ln\frac{\widehat{q}_{t+1,i}}{\widehat{q}_{t,i}} - \sum_i \alpha_i \widehat{z}_{t+1,i} \ln\frac{\widehat{q}_{t+1,i}}{\widehat{q}_{t,i}} \\
&= \sum_{i\in\mathrm{supp}(z^*)} \alpha_i z_i' \ln\frac{\widehat{q}_{t+1,i}}{\widehat{q}_{t,i}} - \sum_{h\in\mathcal{H}^{\mathcal{Z}}} \alpha_h \widehat{z}_{t+1,\sigma(h)} \sum_{j\in\Omega_h} \widehat{q}_{t+1,j} \ln\frac{\widehat{q}_{t+1,j}}{\widehat{q}_{t,j}} \\
&\le \|\boldsymbol{\alpha}\|_\infty \sum_{i\in\mathrm{supp}(z^*)} \left|\ln\frac{\widehat{q}_{t+1,i}}{\widehat{q}_{t,i}}\right| && \left(\textstyle\sum_{j\in\Omega_h} \widehat{q}_{t+1,j} \ln\frac{\widehat{q}_{t+1,j}}{\widehat{q}_{t,j}} \ge 0\right) \\
&\le \|\boldsymbol{\alpha}\|_\infty \sum_{i\in\mathrm{supp}(z^*)} \ln\left(1 + \frac{|\widehat{q}_{t+1,i} - \widehat{q}_{t,i}|}{\min\{\widehat{q}_{t+1,i}, \widehat{q}_{t,i}\}}\right) \\
&\le \|\boldsymbol{\alpha}\|_\infty \sum_{i\in\mathrm{supp}(z^*)} \frac{|\widehat{q}_{t+1,i} - \widehat{q}_{t,i}|}{\min\{\widehat{q}_{t+1,i}, \widehat{q}_{t,i}\}} && (\ln(1+a) \le a) \\
&\le \frac{\|\boldsymbol{\alpha}\|_\infty}{\epsilon_{\mathrm{dil}}} \sum_{i\in\mathrm{supp}(z^*)} |\widehat{q}_{t+1,i} - \widehat{q}_{t,i}|. && (\text{Lemma 20})
\end{aligned}$$

Since $z'$ can be chosen as any point in $\mathcal{V}^*(\mathcal{Z})$, we further lower bound the left-hand side of Eq. (37) using Lemma 15 and get for VOMWU,

$$\begin{aligned}
\eta C\|z^* - \widehat{z}_{t+1}\|_1 &\le \frac{2}{\epsilon_{\mathrm{van}}} \|\widehat{z}_{t+1} - \widehat{z}_t\|_1 + \frac{\|z_t - \widehat{z}_{t+1}\|_1}{4} \\
&\le \frac{2}{\epsilon_{\mathrm{van}}} \left(\|\widehat{z}_{t+1} - \widehat{z}_t\|_1 + \|z_t - \widehat{z}_{t+1}\|_1\right), \tag{40}
\end{aligned}$$

and for DOMWU,

$$\eta C \|\boldsymbol{z}^* - \widehat{\boldsymbol{z}}_{t+1}\|_1 \leq \frac{1}{4}\|\boldsymbol{z}_t - \widehat{\boldsymbol{z}}_{t+1}\|_1 + \frac{\|\boldsymbol{\alpha}\|_\infty}{\epsilon_{\text{dil}}} \sum_{i \in \text{supp}(\boldsymbol{z}^*)} |\widehat{q}_{t+1,i} - \widehat{q}_{t,i}|, \qquad \text{(Lemma 20)}$$

$$\leq \frac{1}{4}\|\boldsymbol{z}_t - \widehat{\boldsymbol{z}}_{t+1}\|_1 + \frac{\|\boldsymbol{\alpha}\|_\infty}{\epsilon_{\text{dil}}} \sum_{i \in \text{supp}(\boldsymbol{z}^*)} \frac{|\widehat{z}_{t+1,i} - \widehat{z}_{t,i}|}{\widehat{z}_{t+1,p_i}} + \widehat{z}_{t,i} \frac{|\widehat{z}_{t+1,p_i} - \widehat{z}_{t,p_i}|}{\widehat{z}_{t+1,p_i}\widehat{z}_{t,p_i}}$$

$$\leq \frac{1}{4}\|\boldsymbol{z}_t - \widehat{\boldsymbol{z}}_{t+1}\|_1 + \frac{\|\boldsymbol{\alpha}\|_\infty}{\epsilon_{\text{dil}}^2} \sum_{i \in \text{supp}(\boldsymbol{z}^*)} |\widehat{z}_{t+1,i} - \widehat{z}_{t,i}| + |\widehat{z}_{t+1,p_i} - \widehat{z}_{t,p_i}|$$

$$\text{(Lemma 20)}$$

$$\leq \frac{1}{4}\|\boldsymbol{z}_t - \widehat{\boldsymbol{z}}_{t+1}\|_1 + \frac{P\|\boldsymbol{\alpha}\|_\infty}{\epsilon_{\text{dil}}^2} \|\widehat{\boldsymbol{z}}_{t+1} - \widehat{\boldsymbol{z}}_t\|_1$$

$$\leq \left(\frac{1}{4} + \frac{P\|\boldsymbol{\alpha}\|_\infty}{\epsilon_{\text{dil}}^2}\right) \left(\|\widehat{\boldsymbol{z}}_{t+1} - \widehat{\boldsymbol{z}}_t\|_1 + \|\boldsymbol{z}_t - \widehat{\boldsymbol{z}}_{t+1}\|_1\right). \qquad (41)$$

Squaring both sides of Eq. (40), we get

$$\eta^2 C^2 \|\boldsymbol{z}^* - \widehat{\boldsymbol{z}}_{t+1}\|_1^2 \leq \frac{4}{\epsilon_{\text{van}}^2} \left(\|\widehat{\boldsymbol{z}}_{t+1} - \widehat{\boldsymbol{z}}_t\|_1 + \|\boldsymbol{z}_t - \widehat{\boldsymbol{z}}_{t+1}\|_1\right)^2$$

$$\leq \frac{8}{\epsilon_{\text{van}}^2} \left(\|\widehat{\boldsymbol{z}}_{t+1} - \widehat{\boldsymbol{z}}_t\|_1^2 + \|\boldsymbol{z}_t - \widehat{\boldsymbol{z}}_{t+1}\|_1^2\right). \qquad (42)$$

Using the strong convexity of the regularizers, the left hand side of Eq. (25) can be bounded by

$$D_\psi(\widehat{\boldsymbol{z}}_{t+1}, \boldsymbol{z}_t) + D_\psi(\boldsymbol{z}_t, \widehat{\boldsymbol{z}}_t)$$

$$\geq \frac{1}{2}\|\widehat{\boldsymbol{z}}_{t+1} - \boldsymbol{z}_t\|_1^2 + \frac{1}{2}\|\boldsymbol{z}_t - \widehat{\boldsymbol{z}}_t\|_1^2 \qquad (a^2 + b^2 \geq \tfrac{1}{2}(a+b)^2)$$

$$\geq \frac{1}{8}\|\widehat{\boldsymbol{z}}_{t+1} - \boldsymbol{z}_t\|_1^2 + \frac{1}{4}\left(\|\widehat{\boldsymbol{z}}_{t+1} - \boldsymbol{z}_t\|_1^2 + \|\boldsymbol{z}_t - \widehat{\boldsymbol{z}}_t\|_1^2\right)$$

$$\geq \frac{1}{8}\left(\|\widehat{\boldsymbol{z}}_{t+1} - \widehat{\boldsymbol{z}}_t\|_1^2 + \|\widehat{\boldsymbol{z}}_{t+1} - \boldsymbol{z}_t\|_1^2\right) \qquad (a^2 + b^2 \geq \tfrac{1}{2}(a+b)^2 \text{ and triangle inequality})$$

Combining this with Eq. (41) finishes the proof for VOMWU. The similar argument works for DOMWU by combining the inequality above with Eq. (42). □

### E.2 Proofs of Eq. (12) and Eq. (13)

In this subsection, we prove Eq. (12) and Eq. (13). We first show Lemma 16 and Lemma 17. Then we get Eq. (12) and Eq. (13) by substituting $\boldsymbol{z}$ with $\widehat{\boldsymbol{z}}_{t+1}$ in these lemmas.

**Lemma 16.** *For any $\boldsymbol{z} \in \mathcal{Z}$, we have*

$$D_{\Psi^{van}}(\boldsymbol{z}^*, \boldsymbol{z}) \leq \sum_{i \in supp(\boldsymbol{z}^*)} \frac{(z_i^* - z_i)^2}{\min_{i \in supp(\boldsymbol{z}^*)} z_i} + \sum_{i \notin supp(\boldsymbol{z}^*)} z_i \leq \frac{3P\|\boldsymbol{z}^* - \boldsymbol{z}\|}{\min_{i \in supp(\boldsymbol{z}^*)} z_i}.$$

*Proof.* By Eq. (14), we have

$$D_{\Psi^{van}}(\boldsymbol{z}^*, \boldsymbol{z}) \leq \sum_i \frac{(z_i^* - z_i)^2}{z_i} \leq \sum_{i \in \text{supp}(\boldsymbol{z}^*)} \frac{(z_i^* - z_i)^2}{\min_{i \in \text{supp}(\boldsymbol{z}^*)} z_i} + \sum_{i \notin \text{supp}(\boldsymbol{z}^*)} z_i$$

$$\leq \frac{\|\boldsymbol{z}^* - \boldsymbol{z}\|^2}{\min_{i \in \text{supp}(\boldsymbol{z}^*)} z_i} + \|\boldsymbol{z}^* - \boldsymbol{z}\|_1 \leq \frac{3P\|\boldsymbol{z}^* - \boldsymbol{z}\|}{\min_{i \in \text{supp}(\boldsymbol{z}^*)} z_i},$$

where the last inequality is because $\|\boldsymbol{z}^* - \boldsymbol{z}\| \leq 2P$ and $\|\boldsymbol{z}^* - \boldsymbol{z}\|_1 \leq P\|\boldsymbol{z}^* - \boldsymbol{z}\|$. □

**Lemma 17.** *For any $\boldsymbol{z} \in \mathcal{Z}$, we have*

$$D_{\Psi_{\boldsymbol{\alpha}}^{dil}}(\boldsymbol{z}^*, \boldsymbol{z}) \leq \|\boldsymbol{\alpha}\|_\infty \left(\sum_{i \in supp(\boldsymbol{z}^*)} \frac{4P}{z_i^*} \frac{(z_i^* - z_i)^2}{q_i} + \sum_{i \notin supp(\boldsymbol{z}^*)} z_{p_i}^* q_i\right) \leq \frac{4P\|\boldsymbol{\alpha}\|_\infty}{\min_{i \in supp(\boldsymbol{z}^*)} z_i^* z_i} \|\boldsymbol{z}^* - \boldsymbol{z}\|_1.$$

*Proof.* By direction calculation and [Wei et al., 2021, Lemma 16], we have

$$D_{\Psi^{\mathrm{dil}}_{\boldsymbol{\alpha}}}(\boldsymbol{z}^*, \boldsymbol{z}) = \sum_i \alpha_{h(i)} z^*_{p_i} q^*_i \ln\left(\frac{q^*_i}{q_i}\right) \tag{Eq. (24)}$$

$$\leq \|\boldsymbol{\alpha}\|_\infty \sum_i z^*_{p_i} \left( \mathbb{1}\{i \in \mathrm{supp}(\boldsymbol{z}^*)\} \frac{(q^*_i - q_i)^2}{q_i} + \mathbb{1}\{i \notin \mathrm{supp}(\boldsymbol{z}^*)\} q_i \right)$$

$$\leq \|\boldsymbol{\alpha}\|_\infty \sum_{i \in \mathrm{supp}(\boldsymbol{z}^*)} \frac{2z^*_{p_i}}{q_i} \left( \left(\frac{z^*_i - z_i}{z^*_{p_i}}\right)^2 + \left(\frac{z_i}{z^*_{p_i}} - \frac{z_i}{z_{p_i}}\right)^2 \right) + \|\boldsymbol{\alpha}\|_\infty \sum_{i \notin \mathrm{supp}(\boldsymbol{z}^*)} z^*_{p_i} q_i$$

$$= \|\boldsymbol{\alpha}\|_\infty \sum_{i \in \mathrm{supp}(\boldsymbol{z}^*)} \left( \frac{2}{q_i z^*_{p_i}}(z^*_i - z_i)^2 + \frac{2q_i}{z^*_{p_i}}(z^*_{p_i} - z_{p_i})^2 \right) + \|\boldsymbol{\alpha}\|_\infty \sum_{i \notin \mathrm{supp}(\boldsymbol{z}^*)} z^*_{p_i} q_i$$

$$\leq \|\boldsymbol{\alpha}\|_\infty \sum_{i \in \mathrm{supp}(\boldsymbol{z}^*)} \left( \frac{2}{z^*_{p_i}} \frac{(z^*_i - z_i)^2}{q_i} + \frac{2P}{z^*_i}(z^*_i - z_i)^2 \right) + \|\boldsymbol{\alpha}\|_\infty \sum_{i \notin \mathrm{supp}(\boldsymbol{z}^*)} z^*_{p_i} q_i$$

$$(p_i \in \mathrm{supp}(\boldsymbol{z}^*) \text{ for all } i \in \mathrm{supp}(\boldsymbol{z}^*))$$

$$\leq \|\boldsymbol{\alpha}\|_\infty \sum_{i \in \mathrm{supp}(\boldsymbol{z}^*)} \left( \frac{4P}{z^*_i} \frac{(z^*_i - z_i)^2}{q_i} \right) + \|\boldsymbol{\alpha}\|_\infty \sum_{i \notin \mathrm{supp}(\boldsymbol{z}^*)} z^*_{p_i} q_i.$$

This proves the first inequality. The second equality in the lemma follows from

$$D_{\Psi^{\mathrm{dil}}_{\boldsymbol{\alpha}}}(\boldsymbol{z}^*, \boldsymbol{z}) \leq \|\boldsymbol{\alpha}\|_\infty \sum_{i \in \mathrm{supp}(\boldsymbol{z}^*)} \left( \frac{4P}{z^*_i} \frac{(z^*_i - z_i)^2}{q_i} \right) + \|\boldsymbol{\alpha}\|_\infty \sum_{i \notin \mathrm{supp}(\boldsymbol{z}^*)} z^*_{p_i} q_i$$

$$\leq \|\boldsymbol{\alpha}\|_\infty \sum_{i \in \mathrm{supp}(\boldsymbol{z}^*)} \left( \frac{4P}{z^*_i z_i} \right) |z^*_i - z_i| + \|\boldsymbol{\alpha}\|_\infty \sum_{i \notin \mathrm{supp}(\boldsymbol{z}^*)} \frac{z^*_{p_i} z_i}{z_{p_i}} \qquad (|z^*_i - z_i| \leq 1)$$

$$\leq \|\boldsymbol{\alpha}\|_\infty \sum_{i \in \mathrm{supp}(\boldsymbol{z}^*)} \left( \frac{4P}{z^*_i z_i} \right) |z^*_i - z_i| + \|\boldsymbol{\alpha}\|_\infty \sum_{i \notin \mathrm{supp}(\boldsymbol{z}^*)} \frac{\mathbb{1}\{p_i \in \mathrm{supp}(\boldsymbol{z}^*)\}}{z_{p_i}} \cdot |z_i - 0|$$

$$\leq \frac{4P\|\boldsymbol{\alpha}\|_\infty}{\min_{i \in \mathrm{supp}(\boldsymbol{z}^*)} z^*_i z_i} \|\boldsymbol{z}^* - \boldsymbol{z}\|_1.$$

$\square$

### E.3 Lower Bounds on the Probability Masses

In this subsection, we show for all $i \in \mathrm{supp}(\boldsymbol{z}^*)$ and $t$, $\widehat{z}_{t,i}$ computed by VOMWU can be lower bounded by $\epsilon_{\mathrm{van}}$, while $\widehat{z}_{t,i}$ and $z_{t,i}$ computed by DOMWU can be lower bounded by $\epsilon_{\mathrm{dil}}$, where $\epsilon_{\mathrm{van}}$ and $\epsilon_{\mathrm{dil}}$ are defined in Definition 4. We state the results in Lemma 19 and Lemma 20, respectively. We first state the stability of $\widehat{\boldsymbol{q}}_t$ and $\boldsymbol{q}_t$, which directly follows from the stability of OMWU on simplex, for example, [Wei et al., 2021, Lemma 17].

**Lemma 18.** *For $\eta \leq \frac{1}{8P}$, DOMWU guarantees $\frac{3}{4}\widehat{q}_{t,i} \leq q_{t,i} \leq \frac{4}{3}\widehat{q}_{t,i}$ and $\frac{3}{4}\widehat{q}_{t,i} \leq \widehat{q}_{t+1,i} \leq \frac{4}{3}\widehat{q}_{t,i}$.*

**Lemma 19.** *For all $i \in \mathrm{supp}(\boldsymbol{z}^*)$ and $t$, VOMWU guarantees that $\widehat{z}_{t,i} \geq \epsilon_{\mathrm{van}}$.*

*Proof.* Using Eq. (5), we have

$$D_\psi(\boldsymbol{z}^*, \widehat{\boldsymbol{z}}_t) \leq \Theta_t \leq \cdots \leq \Theta_1 = \tfrac{1}{16} D_\psi(\widehat{\boldsymbol{z}}_1, \boldsymbol{z}_0) + D_\psi(\boldsymbol{z}^*, \widehat{\boldsymbol{z}}_1) = D_\psi(\boldsymbol{z}^*, \widehat{\boldsymbol{z}}_1), \tag{43}$$

where the last equality is because $\widehat{z}_1 = z_0$. Thus, $D_{\Psi^{\text{van}}}(z^*, \widehat{z}_t) \leq D_{\Psi^{\text{van}}}(z^*, \widehat{z}_1)$. Then, for any $i \in \text{supp}(z^*)$, we have

$$
\begin{aligned}
z_i^* \ln \frac{1}{\widehat{z}_{t,i}} &\leq \sum_j z_j^* \ln \frac{1}{\widehat{z}_{t,j}} = D_{\Psi^{\text{van}}}(z^*, \widehat{z}_t) + \sum_j \left( z_j^* - \widehat{z}_{t,j} - z_j^* \ln z_j^* \right) \\
&\leq D_{\Psi^{\text{van}}}(z^*, \widehat{z}_1) - \sum_j z_j^* \ln z_j^* + \sum_j (z_j^* - \widehat{z}_{t,j}) \\
&\leq \sum_j z_j^* \ln \frac{1}{\widehat{z}_{1,j}} + \sum_j (\widehat{z}_{1,j} - \widehat{z}_{t,j}) \\
&\leq P + \sum_j z_j^* \ln \frac{1}{\widehat{z}_{1,j}} = P(1 + \ln(1/z_{\min})).
\end{aligned}
$$

Therefore, we conclude for all $t$ and $i \in \text{supp}(z^*)$, $\widehat{z}_{t,i}$ satisfies

$$
\widehat{z}_{t,i} \geq \exp\left( -\frac{P(1 + \ln(1/z_{\min}))}{z_i^*} \right) \geq \min_{j \in \text{supp}(z^*)} \exp\left( -\frac{P(1 + \ln(1/z_{\min}))}{z_j^*} \right) = \epsilon_{\text{van}}.
$$

$\square$

**Lemma 20.** *For all $i \in \text{supp}(z^*)$ and $t$, DOMWU guarantees that $\widehat{q}_{t,i} \geq \widehat{z}_{t,i} \geq \epsilon_{dil}$ and $q_{t,i} \geq z_{t,i} \geq \epsilon_{dil}$.*

*Proof.* Similar to Lemma 19, applying Eq. (43) gives $D_{\Psi_{\alpha}^{\text{dil}}}(z^*, \widehat{z}_t) \leq D_{\Psi_{\alpha}^{\text{dil}}}(z^*, \widehat{z}_1)$. Then, for any $i \in \text{supp}(z^*)$, we have

$$
\begin{aligned}
z_i^* \ln \frac{1}{\widehat{q}_{t,i}} &\leq \sum_j \alpha_j z_j^* \ln \frac{1}{\widehat{q}_{t,j}} = D_{\Psi_{\alpha}^{\text{dil}}}(z^*, \widehat{z}_t) - \sum_j \alpha_j z_j^* \ln q_j^* \leq D_{\Psi_{\alpha}^{\text{dil}}}(z^*, \widehat{z}_1) - \sum_j \alpha_j z_j^* \ln q_j^* \\
&= \sum_j \alpha_j z_j^* \ln \frac{1}{\widehat{q}_{1,j}} \leq \|\alpha\|_\infty P \ln(1/z_{\min}).
\end{aligned}
$$

Therefore, we conclude for all $t$ and $i \in \text{supp}(z^*)$, $\widehat{q}_{t,i}$ satisfies

$$
\widehat{q}_{t,i} \geq \exp\left( -\frac{\|\alpha\|_\infty P \ln(1/z_{\min})}{z_i^*} \right) \geq \min_{j \in \text{supp}(z^*)} \exp\left( -\frac{\|\alpha\|_\infty P \ln(1/z_{\min})}{z_j^*} \right)
$$

and $\widehat{z}_{t,i}$ satisfies

$$
\widehat{z}_{t,i} = \widehat{z}_{t,p_i} \widehat{q}_{t,i} = \widehat{z}_{t,p_{p_i}} \widehat{q}_{t,p_i} \widehat{q}_{t,i} \geq \prod_{j \in \text{supp}(z^*)} \widehat{q}_{t,j} \geq \min_{j \in \text{supp}(z^*)} \exp\left( -\frac{\|\alpha\|_\infty P \ln(1/z_{\min})}{z_j^*} \right)^P.
$$

This finishes the first part of the proof. Finally, using Lemma 18, we have for $i \in \text{supp}(z^*)$, $q_{t,i} \geq \frac{3}{4} \widehat{q}_{t,i} \geq \epsilon_{\text{dil}}$ and

$$
z_{t,i} \geq \prod_{j \in \text{supp}(z^*)} q_{t,j} \geq \min_{j \in \text{supp}(z^*)} \exp\left( -\frac{\|\alpha\|_\infty P^2 \ln(1/z_{\min})}{z_j^*} \right) \cdot \left( \frac{3}{4} \right)^P \geq \epsilon_{\text{dil}}.
$$

$\square$

### E.4 Proof of Theorem 6

Based on the results in the previous subsections and the discussion in Section 6.3, we can get $\Theta_t = \mathcal{O}(1/t)$ for both VOMWU and DOMWU. In this subsection, we formally state the results in Theorem 21 for both VOMWU and DOMWU, and show the proof by combining all the components. In particular, the result for VOMWU implies Theorem 6.

**Theorem 21.** *Under the uniqueness assumption,* VOMWU *and* DOMWU *with step size* $\eta \leq \frac{1}{8P}$ *guarantee* $D_{\Psi^{\mathrm{van}}}(z^*, \widehat{z}_t) \leq \frac{C_{13}}{t}$ *and* $D_{\Psi_{\alpha}^{\mathrm{dil}}}(z^*, \widehat{z}_t) \leq \frac{C'_{13}}{t}$, *respectively, where* $C'_{13}, C_{13} > 0$ *are some constants depending on the game,* $\widehat{z}_1$, *and* $\eta$.

*Proof.* We start from Lemma 12. Using Lemma 16 and Lemma 19, the right hand side of Eq. (25) can be bounded by

$$\zeta_t \geq C_{12}\|z^* - \widehat{z}_{t+1}\|^2 \geq \frac{\epsilon_{\mathrm{van}}^2 C_{12}}{9P^2} D_{\Psi^{\mathrm{van}}}(z^*, \widehat{z}_{t+1})^2, \tag{44}$$

for VOMWU. Similarly, using Lemma 17 and Lemma 20, we have for DOMWU,

$$\zeta_t \geq C_{12}\|z^* - \widehat{z}_{t+1}\|^2 \geq \frac{\epsilon_{\mathrm{van}}^4 C_{12}}{16P^2\|\alpha\|_{\infty}^2} D_{\Psi_{\alpha}^{\mathrm{dil}}}(z^*, \widehat{z}_{t+1})^2.$$

On the other hand, applying Eq. (5) repeatedly, we get

$$D_{\psi}(z^*, \widehat{z}_1) = \Theta_1 \geq \cdots \geq \Theta_{t+1} \geq \frac{1}{16} D_{\psi}(\widehat{z}_{t+1}, z_t).$$

Thus, $\zeta_t \geq D_{\psi}(\widehat{z}_{t+1}, z_t) \geq C_{10} D_{\psi}(\widehat{z}_{t+1}, z_t)^2$ for some $C_{10} > 0$ depending on $D_{\psi}(z^*, \widehat{z}_1)$. Combining this with Eq. (44) gives

$$
\begin{aligned}
\zeta_t &= \frac{1}{2}\left( D_{\psi}(\widehat{z}_{t+1}, z_t) + D_{\psi}(z_t, \widehat{z}_t) \right) + \frac{1}{2}\left( D_{\psi}(\widehat{z}_{t+1}, z_t) + D_{\psi}(z_t, \widehat{z}_t) \right) \\
&\geq \frac{1}{2} \cdot \frac{\epsilon_{\mathrm{van}}^2 C_{12}}{9P^2} D_{\Psi^{\mathrm{van}}}(z^*, \widehat{z}_{t+1})^2 + \frac{1}{2} \cdot C_{10} D_{\Psi^{\mathrm{van}}}(\widehat{z}_{t+1}, z_t)^2 \qquad \text{(Eq. (44))} \\
&\geq \frac{\epsilon_{\mathrm{van}}^2 C_{12}}{36P^2} \cdot 2D_{\Psi^{\mathrm{van}}}(z^*, \widehat{z}_{t+1})^2 + \frac{C_{10}}{4} \cdot 2D_{\Psi^{\mathrm{van}}}(\widehat{z}_{t+1}, z_t)^2 \\
&\geq \min\left\{ \frac{\epsilon_{\mathrm{van}}^2 C_{12}}{36P^2}, \frac{C_{10}}{4} \right\} \Theta_{t+1}^2 \geq C_{11} \Theta_{t+1}^2,
\end{aligned}
$$

for some $C_{11} > 0$. Similarly, we have $\zeta_t \geq C'_{11} \Theta_{t+1}^2$ for DOMWU and constant $C'_{11} > 0$. Applying this to Eq. (5), we obtain the recursion $\Theta_{t+1} \leq \Theta_t - \frac{15}{16} C_{11} \Theta_{t+1}^2$. This implies $\Theta_t \leq \frac{C_{13}}{t}$ for some constant $C_{13}$ by [Wei et al., 2021, Lemma 12]. With the same argument, we can prove the case for DOMWU. $\qquad\square$

## E.5 Proof of Theorem 7

### E.5.1 The Significant Difference Lemma

In this subsection, we explain how to get the linear convergence result of DOMWU. As we discuss in the end of Section 6.3, this requires showing that $\sum_{i \notin \mathrm{supp}(z^*)} z_{p_i}^* \widehat{q}_{t+1,i}$ decreases significantly as $\widehat{z}_t$ gets close enough to $z^*$. The argument is shown in Lemma 24. Before that, we first show a lemma stating that for any information set $h \in \mathcal{H}^{\mathcal{Z}}$, indices $i, j \in \Omega_h$ such that $i \notin \mathrm{supp}(z^*)$ and $j \in \mathrm{supp}(z^*)$, $\widehat{L}_{t,i}$ is significantly larger than $\widehat{L}_{t,j}$ when $\widehat{z}_t$ is close to $z^*$.

**Lemma 22.** *Suppose* $\|z^* - z\| \leq \frac{\eta^2 \xi \epsilon_{\mathrm{dil}}^2}{40P^3\|\alpha\|_{\infty}}$. *Then for* $i \in \Omega_h$, $h \in \mathcal{H}^{\mathcal{X}}$, *we have*

$$\forall i \in \mathrm{supp}(x^*), \qquad L_i \leq V_h^* + \frac{9\eta\xi}{10}; \qquad \forall i \notin \mathrm{supp}(x^*), \qquad L_i \geq V_h^* + \frac{9\eta\xi}{10}, \tag{45}$$

*where* $L_i = (Gy)_i + \sum_{g \in \mathcal{H}_i} -\frac{\alpha_g}{\eta} \ln\left( \sum_{j \in \Omega_g} q_j \exp(-\eta L_j / \alpha_g) \right)$, $q_j = z_j / z_{p_j}$.

*Proof.* We first consider terminal index $i$. By the assumption $\|z^* - z\| \leq \frac{\eta^2 \xi \epsilon_{\mathrm{dil}}^2}{40P^3\|\alpha\|_{\infty}}$ and Lemma 13, we have $\|y - y^*\|_1 \leq \frac{\eta\xi}{10P}$, and

$$L_i = (Gy)_i \leq (Gy^*)_i + \frac{\eta\xi}{10P} = V_h^* + \frac{\eta\xi}{10P} \tag{46}$$

for $i \in \text{supp}(\boldsymbol{x}^*)$ and

$$L_i = (\boldsymbol{Gy})_i \geq (\boldsymbol{Gy}^*)_i - \frac{\eta\xi}{10P} \geq V_h^* + \xi - \frac{\eta\xi}{10P} \geq V_h^* + \frac{9\eta\xi}{10}$$

for $i \notin \text{supp}(\boldsymbol{x}^*)$ by the definition of $\xi$. Therefore, this shows Eq. (45) for terminal indices. In the following, we complete the proof by backward induction. Specifically, for nonterminal index $i \notin \text{supp}(\boldsymbol{x}^*)$, we assume $L_j \geq V_{h(j)}^* + \frac{9\eta\xi}{10}$ for every *descendant* $j$ (we say that index $j$ is a descendant of index $i$ if there exists a sequence of indexes $s_0, \ldots, s_K$ for some $K > 0$ such that $s_0 = j$, $s_K = i$, and $p_{s_{k-1}} = s_k$ for every $k \in [K]$). Note that we always have $j \notin \text{supp}(\boldsymbol{x}^*)$. We will prove $L_i$ satisfies Eq. (45), which completes the proof for $i \notin \text{supp}(\boldsymbol{x}^*)$ by induction. By assumption, we have

$$L_i = (\boldsymbol{Gy})_i + \sum_{g \in \mathcal{H}_i} -\frac{\alpha_g}{\eta} \ln \left( \sum_{j \in \Omega_g} q_j \exp(-\eta L_j/\alpha_g) \right)$$

$$\geq (\boldsymbol{Gy})_i + \sum_{g \in \mathcal{H}_i} -\frac{\alpha_g}{\eta} \ln \left( \max_{j \in \Omega_g} \exp(-\eta L_j/\alpha_g) \right)$$

$$= (\boldsymbol{Gy})_i + \sum_{g \in \mathcal{H}_i} \min_{j \in \Omega_g} L_j$$

$$\geq (\boldsymbol{Gy})_i + \sum_{g \in \mathcal{H}_i} V_g^* + \frac{9\eta\xi}{10} \qquad \text{(by the induction hypothesis)}$$

$$\geq (\boldsymbol{Gy}^*)_i - \xi + \sum_{g \in \mathcal{H}_i} V_g^* + \frac{9\eta\xi}{10} \qquad (\|\boldsymbol{y} - \boldsymbol{y}^*\| \leq \xi)$$

$$\geq V_{h(i)}^* + \frac{9\eta\xi}{10}. \qquad \text{(Lemma 13)}$$

Similarly, for nonterminal index $i \in \text{supp}(\boldsymbol{x}^*)$, we show for every descendant $j$,

$$L_j \leq V_{h(j)}^* + \frac{\eta\xi}{10P} f(h(j)), \tag{47}$$

where $f : \mathcal{H}^{\mathcal{X}} \to \mathbb{R}^+$ is defined recursively as follows. For information set (simplex) $g$ such that $\Omega_g$ contains terminal indices only, we let $f(g) = 1$. Otherwise, we define

$$f(g) = \max_{k \in \Omega_g} \sum_{s \in \mathcal{H}_k} \left( f(s) + \frac{1}{P} \right). \tag{48}$$

This shows Eq. (45) as Lemma 23 guarantees

$$f(h) \leq (P - 1) \cdot \left( 1 + \frac{1}{P} \right) < P,$$

for every simplex $h$. It remains to prove Eq. (48) by induction. For the base case that $i$ is a terminal index, Eq. (47) clearly holds by Eq. (46). For nonterminal index $i$, we have

$$L_i = (\boldsymbol{Gy})_i + \sum_{g \in \mathcal{H}_i} -\frac{\alpha_g}{\eta} \ln \left( \sum_{j \in \Omega_g} q_j \exp(-\eta L_j/\alpha_g) \right)$$

$$\leq (\boldsymbol{Gy})_i + \sum_{g \in \mathcal{H}_i} -\frac{\alpha_g}{\eta} \ln \left( \sum_{j \in \Omega_g \cap \text{supp}(\boldsymbol{x}^*)} q_j \exp(-\eta L_j/\alpha_g) \right)$$

$$\leq (\boldsymbol{Gy})_i + \sum_{g \in \mathcal{H}_i} -\frac{\alpha_g}{\eta} \ln \left[ \exp \left( -\eta \left( V_g^* + \frac{\eta\xi}{10P} f(g) \right) / \alpha_g \right) \sum_{j \in \Omega_g \cap \text{supp}(\boldsymbol{x}^*)} q_j \right]$$

$$\text{(by the assumption)}$$

$$= (\boldsymbol{Gy})_i + \sum_{g \in \mathcal{H}_i} \left[ V_g^* + \frac{\eta\xi}{20P} f(g) - \frac{\alpha_g}{\eta} \ln \left( \sum_{j \in \Omega_g \cap \text{supp}(\boldsymbol{x}^*)} q_j \right) \right]. \tag{49}$$

We continue to bound the last term. Let $c = \frac{\eta^2 \xi \epsilon_{\text{dil}}^2}{40 P^3 \|\boldsymbol{\alpha}\|_\infty}$. We have

$$-\frac{\alpha_g}{\eta} \ln \left( \sum_{j \in \Omega_g \cap \text{supp}(\boldsymbol{x}^*)} q_j \right) = -\frac{\alpha_g}{\eta} \ln \left( \frac{\sum_{j \in \Omega_g \cap \text{supp}(\boldsymbol{x}^*)} x_j}{x_{p_j}} \right)$$

$$\leq -\frac{\alpha_g}{\eta} \ln \left( \frac{\sum_{j \in \Omega_g \cap \text{supp}(\boldsymbol{x}^*)} (x_j^* - c)}{x_{p_j}^* + c} \right) \quad (\|\boldsymbol{z}^* - \boldsymbol{z}\| \leq \frac{\eta^2 \xi \epsilon_{\text{dil}}^2}{40 P^3 \|\boldsymbol{\alpha}\|_\infty})$$

$$\leq -\frac{\alpha_g}{\eta} \ln \left( \frac{x_{p_j}^* - c|\Omega_g|}{x_{p_j}^* + c} \right)$$

$$= -\frac{\alpha_g}{\eta} \ln \left( 1 - \frac{c(|\Omega_g| + 1)}{x_{p_j}^* + c} \right)$$

$$\leq -\frac{\alpha_g}{\eta} \ln \left( 1 - \frac{cP}{\epsilon_{\text{dil}}} \right) \quad (x_{p_j}^* + c \geq x_{p_j}^* \geq \epsilon_{\text{dil}} \text{ and } |\Omega_g| + 1 \leq P)$$

$$\leq -\frac{\alpha_g}{\eta} \ln \left( 1 - \frac{\eta^2 \xi}{40 \alpha_g P^2} \right) \quad \text{(by definition of } c\text{)}$$

$$\leq \frac{\eta \xi}{20 P^2}. \quad (-\ln(1 - x) < 2x \text{ for } 0 < x < 0.5)$$

Plugging this back to the original inequalities, we get

$$L_i \leq (\boldsymbol{Gy})_i + \sum_{g \in \mathcal{H}_i} \left( V_g^* + \frac{\eta \xi}{10 P} f(g) + \frac{\eta \xi}{20 P^2} \right)$$

$$\leq (\boldsymbol{Gy}^*)_i + \frac{\eta \xi}{20 P^2} + \sum_{g \in \mathcal{H}_i} \left( V_g^* + \frac{\eta \xi}{10 P} f(g) + \frac{\eta \xi}{20 P^2} \right) \quad (\|\boldsymbol{y} - \boldsymbol{y}^*\|_1 \leq \frac{\eta \xi}{10 P^2})$$

$$\leq (\boldsymbol{Gy}^*)_i + \sum_{g \in \mathcal{H}_i} \left( V_g^* + \frac{\eta \xi}{10 P} f(g) + \frac{\eta \xi}{10 P^2} \right) \quad (i \text{ is nonterminal})$$

$$= (\boldsymbol{Gy}^*)_i + \sum_{g \in \mathcal{H}_i} V_g^* + \frac{\eta \xi}{10 P} \sum_{g \in \mathcal{H}_i} \left( f(g) + \frac{1}{P} \right)$$

$$\leq V_h^* + \frac{\eta \xi}{10 P} \sum_{g \in \mathcal{H}_i} \left( f(g) + \frac{1}{P} \right) \quad \text{(Lemma 13)}$$

$$\leq V_h^* + \frac{\eta \xi}{10 P} f(h(i)), \quad \text{(Eq. (48))}$$

which shows Eq. (47) by induction, and thus shows Eq. (45). $\qquad \square$

**Lemma 23.** *Define* $f : \mathcal{H}^{\mathcal{X}} \to \mathbb{R}^+$ *as follows.*

$$f(g) = \begin{cases} 1, & \text{if } \Omega_g \text{ contains terminal indices only;} \\ \max_{k \in \Omega_g} \sum_{s \in \mathcal{H}_k} \left( f(s) + \frac{1}{P} \right), & \text{otherwise.} \end{cases}$$

*Then for every* $g \in \mathcal{H}^{\mathcal{X}}$*, we have*

$$f(g) \leq I_g \left( 1 + \frac{1}{P} \right), \tag{50}$$

*where* $I_g$ *is the number of indices that are the descendants of* $g$ *(we say that index* $j$ *is a descendant of simplex* $g$ *if* $j \in \Omega_g$ *or index* $j$ *is a descendant of index* $i$ *for some* $i \in \Omega_g$*).*

*Proof.* If $\Omega_g$ contains terminal indices only, since $I_g \geq 1$, Eq. (50) holds. Otherwise, suppose Eq. (50) holds for all simplexes that are descendants of $g$ (we say that simplex $h$ is a descendant of

simplex $g$ if there exists a sequence of simplexes $s_0, \ldots, s_K$ for some $K > 0$ such that $s_0 = h$, $s_K = g$, and $\sigma(s_{k-1}) \in \Omega_{s_k}$ for every $k \in [K]$). We define

$$k^* = \underset{k \in \Omega_g}{\operatorname{argmax}} \sum_{s \in \mathcal{H}_k} \left( f(s) + \frac{1}{P} \right).$$

Then we have

$$
\begin{aligned}
f(g) &= \sum_{s \in \mathcal{H}_{k^*}} \left( f(s) + \frac{1}{P} \right) && \text{(by definition of } f) \\
&\leq \sum_{s \in \mathcal{H}_{k^*}} \left[ I_s \left( 1 + \frac{1}{P} \right) + \frac{1}{P} \right] && \text{(by assumption)} \\
&\leq 1 + \sum_{s \in \mathcal{H}_{k^*}} \left[ I_s \left( 1 + \frac{1}{P} \right) \right] && (|\mathcal{H}_{k^*}| \leq P) \\
&\leq (I_g - 1) \left( 1 + \frac{1}{P} \right) + 1 \\
&\leq I_g \left( 1 + \frac{1}{P} \right),
\end{aligned}
$$

where the third inequality is because $k^*$ is not a descendant of any $s \in \mathcal{H}_{k^*}$, and thus

$$\sum_{s \in \mathcal{H}_{k^*}} I_s \leq I_g - 1.$$

Therefore, we show Eq. (50) by induction. $\qquad \square$

### E.5.2 The Counterpart of Eq. (11) for DOMWU

With Lemma 22, we can prove the following lemma, the counterpart of Eq. (11) for DOMWU.

**Lemma 24.** *Under the uniqueness assumption, there exists a constant $C_{14} > 0$ that depends on the game, $\eta$, and $\widehat{\boldsymbol{z}}_1$ such that for any $t \geq 1$, DOMWU with step size $\eta \leq \frac{1}{8P}$ guarantees*

$$D_{\Psi_{\boldsymbol{\alpha}}^{dil}}(\widehat{\boldsymbol{z}}_{t+1}, \boldsymbol{z}_t) + D_{\Psi_{\boldsymbol{\alpha}}^{dil}}(\boldsymbol{z}_t, \widehat{\boldsymbol{z}}_t) \geq C_{14} D_{\Psi_{\boldsymbol{\alpha}}^{dil}}(\boldsymbol{z}^*, \widehat{\boldsymbol{z}}_{t+1})$$

*as long as $\max\{\|\boldsymbol{z}^* - \widehat{\boldsymbol{z}}_t\|_1, \|\boldsymbol{z}^* - \boldsymbol{z}_t\|_1\} \leq \frac{\eta^2 \xi \epsilon_{dil}^2}{40 P^3 \|\boldsymbol{\alpha}\|_\infty}$.*

*Proof.* We define $\alpha_{\min} = \min_{h \in \mathcal{H}^z} \alpha_h > 0$. Note that

$$
\begin{aligned}
&D_{\Psi_{\boldsymbol{\alpha}}^{dil}}(\widehat{\boldsymbol{z}}_{t+1}, \boldsymbol{z}_t) + D_{\Psi_{\boldsymbol{\alpha}}^{dil}}(\boldsymbol{z}_t, \widehat{\boldsymbol{z}}_t) \\
&= \sum_{g \in \mathcal{H}^z} \alpha_g \cdot \widehat{z}_{t+1, \sigma(g)} \mathrm{KL}(\widehat{\boldsymbol{q}}_{t+1, g}, \boldsymbol{q}_{t, g}) + \alpha_g \cdot z_{t, \sigma(g)} \mathrm{KL}(\boldsymbol{q}_{t, g}, \widehat{\boldsymbol{q}}_{t, g}) && \text{(Eq. (24))} \\
&\geq \alpha_{\min} \sum_{g \in \mathcal{H}^z} \widehat{z}_{t+1, \sigma(g)} \mathrm{KL}(\widehat{\boldsymbol{q}}_{t+1, g}, \boldsymbol{q}_{t, g}) + z_{t, \sigma(g)} \mathrm{KL}(\boldsymbol{q}_{t, g}, \widehat{\boldsymbol{q}}_{t, g}) \\
&\geq \alpha_{\min} \epsilon_{\mathrm{dil}} \sum_{g \in \mathcal{H}, \, \sigma(g) \in \mathrm{supp}(\boldsymbol{z}^*)} \mathrm{KL}(\widehat{\boldsymbol{q}}_{t+1, g}, \boldsymbol{q}_{t, g}) + \mathrm{KL}(\boldsymbol{q}_{t, g}, \widehat{\boldsymbol{q}}_{t, g}) && \text{(Lemma 20)} \\
&\geq \frac{\alpha_{\min} \epsilon_{\mathrm{dil}}}{3} \sum_{i \notin \mathrm{supp}(\boldsymbol{z}^*), p_i \in \mathrm{supp}(\boldsymbol{z}^*)} \left( \frac{(\widehat{q}_{t+1, i} - q_{t, i})^2}{\widehat{q}_{t+1, i}} + \frac{(q_{t, i} - \widehat{q}_{t, i})^2}{q_{t, i}} \right) \\
&\hspace{8cm} \text{([Wei et al., 2021, Lemma 18])} \\
&\geq \frac{\alpha_{\min} \epsilon_{\mathrm{dil}}}{4} \sum_{i \notin \mathrm{supp}(\boldsymbol{z}^*), p_i \in \mathrm{supp}(\boldsymbol{z}^*)} \left( \frac{(\widehat{q}_{t+1, i} - q_{t, i})^2}{\widehat{q}_{t, i}} + \frac{(q_{t, i} - \widehat{q}_{t, i})^2}{\widehat{q}_{t, i}} \right) && \text{(Lemma 18)} \\
&\geq \frac{\alpha_{\min} \epsilon_{\mathrm{dil}}}{8} \sum_{i \notin \mathrm{supp}(\boldsymbol{z}^*), p_i \in \mathrm{supp}(\boldsymbol{z}^*)} \frac{(\widehat{q}_{t+1, i} - \widehat{q}_{t, i})^2}{\widehat{q}_{t, i}}. && (51)
\end{aligned}
$$

Below we continue to bound $\sum_{i\notin\text{supp}(\boldsymbol{z}^*),p_i\in\text{supp}(\boldsymbol{z}^*)}\frac{(\widehat{q}_{t+1,i}-\widehat{q}_{t,i})^2}{\widehat{q}_{t,i}}$.

By the assumption, we have $\|\widehat{\boldsymbol{q}}_t-\boldsymbol{q}^*\|_1\leq P\|\widehat{\boldsymbol{z}}_t-\boldsymbol{z}^*\|_1/\epsilon_{\text{dil}}^2\leq\frac{\eta\xi}{10\alpha_i}$ and for index $i$ such that $i\notin\text{supp}(\boldsymbol{x}^*)$ and $p_i\in\text{supp}(\boldsymbol{x}^*)$,

$$\sum_{j\in\Omega_{h(i)},j\notin\text{supp}(\boldsymbol{x}^*)}\widehat{q}_{t,j}\leq\frac{\eta\xi}{10\alpha_i}. \tag{52}$$

Moreover, by Lemma 22, we have (denote $h=h(i)$)

$$\begin{aligned}
\widehat{q}_{t+1,i}&=\frac{\widehat{q}_{t,i}\exp(-\eta\widehat{L}_i/\alpha_i)}{\sum_{j\in\Omega_h}\widehat{q}_{t,j}\exp(-\eta\widehat{L}_j/\alpha_i)}\leq\frac{\widehat{q}_{t,i}\exp(-\eta\widehat{L}_i/\alpha_i)}{\sum_{j\in\Omega_h\cap\text{supp}(\boldsymbol{z}^*)}\widehat{q}_{t,j}\exp(-\eta\widehat{L}_j/\alpha_i)}\\
&\leq\frac{\widehat{q}_{t,i}\exp(-\eta(V_h^*+\frac{9\xi}{10})/\alpha_i)}{\sum_{j\in\Omega_h\cap\text{supp}(\boldsymbol{z}^*)}\widehat{q}_{t,j}\exp(-\eta(V_h^*+\frac{\xi}{10})/\alpha_i)} &\text{(Lemma 22)}\\
&=\frac{\widehat{q}_{t,i}\exp\left(-\frac{8}{10}\eta\xi/\alpha_i\right)}{\left(1-\sum_{j\in\Omega_h,j\notin\text{supp}(\boldsymbol{z}^*)}\widehat{q}_{t,j}\right)}\\
&\leq\frac{\widehat{q}_{t,i}\exp\left(-\frac{8}{10}\eta\xi/\alpha_i\right)}{\left(1-\frac{\eta\xi/\alpha_i}{10}\right)} &\text{(Eq. (52))}\\
&\leq\widehat{q}_{t,i}\left(1-\frac{1}{2}\frac{\eta\xi}{\alpha_i}\right). &(\tfrac{\exp(-0.8u)}{1-0.1u}\leq 1-0.5u\text{ for }u\in[0,1])
\end{aligned}$$

Rearranging gives

$$\frac{|\widehat{q}_{t+1,i}-\widehat{q}_{t,i}|^2}{\widehat{q}_{t,i}}\geq\frac{\eta^2\xi^2}{4\|\boldsymbol{\alpha}\|_\infty^2}\widehat{q}_{t,i}\geq\frac{\eta^2\xi^2}{8\|\boldsymbol{\alpha}\|_\infty^2}\widehat{q}_{t+1,i},$$

where the last step uses Lemma 18. The case for $\widehat{\boldsymbol{y}}_t$ is similar. Combining this with Eq. (51), we get

$$\begin{aligned}
D_{\Psi_{\boldsymbol{\alpha}}^{\text{dil}}}(\widehat{\boldsymbol{z}}_{t+1},\boldsymbol{z}_t)+D_{\Psi_{\boldsymbol{\alpha}}^{\text{dil}}}(\boldsymbol{z}_t,\widehat{\boldsymbol{z}}_t)&\geq C_{12}'\sum_{i\notin\text{supp}(\boldsymbol{z}^*),p_i\in\text{supp}(\boldsymbol{z}^*)}\widehat{q}_{t+1,i}\\
&\geq C_{12}'\sum_{i\notin\text{supp}(\boldsymbol{z}^*)}z_{p_i}^*\widehat{q}_{t+1,i} \tag{53}
\end{aligned}$$

for some $C_{12}'>0$. Now we combine two lower bounds of $D_{\Psi_{\boldsymbol{\alpha}}^{\text{dil}}}(\widehat{\boldsymbol{z}}_{t+1},\boldsymbol{z}_t)+D_{\Psi_{\boldsymbol{\alpha}}^{\text{dil}}}(\boldsymbol{z}_t,\widehat{\boldsymbol{z}}_t)$. Using Lemma 12 and Eq. (53), we get

$$\begin{aligned}
&D_{\Psi_{\boldsymbol{\alpha}}^{\text{dil}}}(\widehat{\boldsymbol{z}}_{t+1},\boldsymbol{z}_t)+D_{\Psi_{\boldsymbol{\alpha}}^{\text{dil}}}(\boldsymbol{z}_t,\widehat{\boldsymbol{z}}_t)\\
&=\frac{1}{2}\left(D_{\Psi_{\boldsymbol{\alpha}}^{\text{dil}}}(\widehat{\boldsymbol{z}}_{t+1},\boldsymbol{z}_t)+D_{\Psi_{\boldsymbol{\alpha}}^{\text{dil}}}(\boldsymbol{z}_t,\widehat{\boldsymbol{z}}_t)\right)+\frac{1}{2}\left(D_{\Psi_{\boldsymbol{\alpha}}^{\text{dil}}}(\widehat{\boldsymbol{z}}_{t+1},\boldsymbol{z}_t)+D_{\Psi_{\boldsymbol{\alpha}}^{\text{dil}}}(\boldsymbol{z}_t,\widehat{\boldsymbol{z}}_t)\right)\\
&\geq\frac{C_{12}}{2}\|\boldsymbol{z}^*-\widehat{\boldsymbol{z}}_{t+1}\|_1^2+\frac{C_{12}'}{2}\sum_{i\notin\text{supp}(\boldsymbol{z}^*)}z_{p_i}^*\widehat{q}_{t+1,i}. \tag{54}
\end{aligned}$$

Also note that by Lemma 17, we have

$$\begin{aligned}
D_{\Psi_{\boldsymbol{\alpha}}^{\text{dil}}}(\boldsymbol{z}^*,\widehat{\boldsymbol{z}}_{t+1})&\leq\|\boldsymbol{\alpha}\|_\infty\left(\sum_{i\in\text{supp}(\boldsymbol{z}^*)}\frac{4P}{z_i^*}\frac{(z_i^*-\widehat{z}_{t+1,i})^2}{\widehat{q}_{t+1,i}}+\sum_{i\notin\text{supp}(\boldsymbol{z}^*)}z_{p_i}^*\widehat{q}_{t+1,i}\right)\\
&\leq\frac{4\|\boldsymbol{\alpha}\|_\infty P}{\epsilon_{\text{dil}}^2}\left(\sum_{i\in\text{supp}(\boldsymbol{z}^*)}(z_i^*-\widehat{z}_{t+1,i})^2+\sum_{i\notin\text{supp}(\boldsymbol{z}^*)}z_{p_i}^*\widehat{q}_{t+1,i}\right). \quad\text{(Lemma 20)}
\end{aligned}$$

Combining this with Eq. (54), we conclude that

$$\begin{aligned}
D_{\Psi_{\boldsymbol{\alpha}}^{\text{dil}}}(\widehat{\boldsymbol{z}}_{t+1},\boldsymbol{z}_t)+D_{\Psi_{\boldsymbol{\alpha}}^{\text{dil}}}(\boldsymbol{z}_t,\widehat{\boldsymbol{z}}_t)&\geq\frac{\min\{C_{12},C_{12}'\}}{2}\left(\|\boldsymbol{z}^*-\widehat{\boldsymbol{z}}_{t+1}\|_1^2+\sum_{i\notin\text{supp}(\boldsymbol{z}^*)}z_{p_i}^*\widehat{q}_{t+1,i}\right)\\
&\geq\frac{\epsilon_{\text{dil}}^2}{8\|\boldsymbol{\alpha}\|_\infty P}\min\{C_{12},C_{12}'\}D_{\Psi_{\boldsymbol{\alpha}}^{\text{dil}}}(\boldsymbol{z}^*,\widehat{\boldsymbol{z}}_{t+1}),
\end{aligned}$$

which finishes the proof.

$\square$

### E.5.3 Proof of Theorem 7

With Lemma 24, we are ready to prove Theorem 7.

*Proof of Theorem 7.* Set $T_0 = \frac{64C'_{13}}{c^2}$, where $c = \frac{\eta^2 \xi \epsilon_{\mathrm{dil}}^2}{40P^3 \|\boldsymbol{\alpha}\|_\infty}$. For $t \geq T_0$, we have using Theorem 21,

$$\|\boldsymbol{z}^* - \widehat{\boldsymbol{z}}_t\|_1^2 \leq 2D_{\Psi_{\boldsymbol{\alpha}}^{\mathrm{dil}}}(\boldsymbol{z}^*, \widehat{\boldsymbol{z}}_t) \leq \frac{2C'_{13}}{T_0} \leq c^2,$$

$$\|\boldsymbol{z}^* - \boldsymbol{z}_t\|_1^2 \leq 2\|\boldsymbol{z}^* - \widehat{\boldsymbol{z}}_{t+1}\|_1^2 + 2\|\widehat{\boldsymbol{z}}_{t+1} - \boldsymbol{z}_t\|_1^2$$

$$\leq 4D_{\Psi_{\boldsymbol{\alpha}}^{\mathrm{dil}}}(\boldsymbol{z}^*, \widehat{\boldsymbol{z}}_{t+1}) + 4D_{\Psi_{\boldsymbol{\alpha}}^{\mathrm{dil}}}(\widehat{\boldsymbol{z}}_{t+1}, \boldsymbol{z}_t)$$

$$\leq 64\Theta_{t+1} \leq \frac{64C'_{13}}{T_0} \leq c^2.$$

Therefore, when $t \geq T_0$, the condition of the second part of Lemma 24 is satisfied, and we have

$$\zeta_t \geq \frac{1}{2}D_{\Psi_{\boldsymbol{\alpha}}^{\mathrm{dil}}}(\widehat{\boldsymbol{z}}_{t+1}, \boldsymbol{z}_t) + \frac{1}{2}\zeta_t$$

$$\geq \frac{1}{2}D_{\Psi_{\boldsymbol{\alpha}}^{\mathrm{dil}}}(\widehat{\boldsymbol{z}}_{t+1}, \boldsymbol{z}_t) + \frac{C_{14}}{2}D_{\Psi_{\boldsymbol{\alpha}}^{\mathrm{dil}}}(\boldsymbol{z}^*, \widehat{\boldsymbol{z}}_{t+1}) \qquad \text{(by Lemma 24)}$$

$$\geq C_{15}\Theta_{t+1}.$$

for some constant $C_{15} > 0$. Therefore, when $t \geq T_0$, $\Theta_{t+1} \leq \Theta_t - \frac{15}{16}C_{15}\Theta_{t+1}$, which further leads to

$$\Theta_t \leq \Theta_{T_0} \cdot \left(1 + \frac{15}{16}C_{15}\right)^{T_0 - t} \leq \Theta_1 \cdot \left(1 + \frac{15}{16}C_{15}\right)^{T_0 - t} = D_{\Psi_{\boldsymbol{\alpha}}^{\mathrm{dil}}}(\boldsymbol{z}^*, \widehat{\boldsymbol{z}}_1) \cdot \left(1 + \frac{15}{16}C_{15}\right)^{T_0 - t},$$

where the second inequality uses Eq. (43). The inequality trivially holds for $t < T_0$ as well, so it holds for all $t$. We finish the proof by relating $D_{\Psi_{\boldsymbol{\alpha}}^{\mathrm{dil}}}(\boldsymbol{z}^*, \boldsymbol{z}_t)$ and $\Theta_{t+1}$. Note that by Lemma 17,

$$D_{\Psi_{\boldsymbol{\alpha}}^{\mathrm{dil}}}(\boldsymbol{z}^*, \boldsymbol{z}_t)^2 \leq \frac{16P\|\boldsymbol{\alpha}\|_\infty^2}{\epsilon_{\mathrm{dil}}^4}\|\boldsymbol{z}^* - \boldsymbol{z}_t\|_1^2$$

$$\leq \frac{32P\|\boldsymbol{\alpha}\|_\infty^2}{\epsilon_{\mathrm{dil}}^4}\left(\|\boldsymbol{z}^* - \widehat{\boldsymbol{z}}_{t+1}\|_1^2 + \|\widehat{\boldsymbol{z}}_{t+1} - \boldsymbol{z}_t\|_1^2\right)$$

$$\leq \frac{1024P\|\boldsymbol{\alpha}\|_\infty^2}{\epsilon_{\mathrm{dil}}^4}\Theta_{t+1}.$$

Therefore, we conclude

$$D_{\Psi_{\boldsymbol{\alpha}}^{\mathrm{dil}}}(\boldsymbol{z}^*, \boldsymbol{z}_t) \leq \sqrt{\frac{1024P\|\boldsymbol{\alpha}\|_\infty^2}{\epsilon_{\mathrm{dil}}^4}\Theta_{t+1}} \leq \sqrt{\frac{1024P\|\boldsymbol{\alpha}\|_\infty^2 D_{\Psi_{\boldsymbol{\alpha}}^{\mathrm{dil}}}(\boldsymbol{z}^*, \widehat{\boldsymbol{z}}_1)}{\epsilon_{\mathrm{dil}}^4}}\left(1 + \frac{15}{16}C_{15}\right)^{\frac{T_0 - t - 1}{2}},$$

which finishes the proof by setting

$$C_3 = \sqrt{\frac{1024P\|\boldsymbol{\alpha}\|_\infty^2 D_{\Psi_{\boldsymbol{\alpha}}^{\mathrm{dil}}}(\boldsymbol{z}^*, \widehat{\boldsymbol{z}}_1)}{\epsilon_{\mathrm{dil}}^4}}\left(1 + \frac{15}{16}C_{15}\right)^{\frac{T_0 - 1}{2}}, \quad C_4 = \left(1 + \frac{15}{16}C_{15}\right)^{\frac{1}{2}} - 1.$$

$\square$

### E.6 Remarks on DOGDA

In this subsection, we discuss the technical difficulties to get a convergence rate for DOGDA. This is challenging even if we assume the uniqueness of the Nash equilibrium. From the analysis of VOMWU and DOMWU, we can see that Lemma 19 and Lemma 20 play an important role. The lemmas lower bound $\widehat{z}_{t,i}$ with some game-dependent constants for $i$ in supp$(\boldsymbol{z}^*)$, and the proofs are based on the observation that $D_{\Psi^{\mathrm{van}}}(\boldsymbol{z}^*, \boldsymbol{z})$ and $D_{\Psi_\alpha^{\mathrm{dil}}}(\boldsymbol{z}^*, \boldsymbol{z})$ approach infinity as $z_i$ approaches zero for $i \in \mathrm{supp}(\boldsymbol{z}^*)$. This property of the entropy regularizers, however, does not hold for the dilated Euclidean regularizer $\Phi_\alpha^{\mathrm{dil}}$ in general. Lower bounding $\widehat{z}_{t,i}$ for DOGDA could be possible when $\widehat{z}_t$ is sufficiently close to $\boldsymbol{z}^*$. For example, when

$$\|\widehat{z}_t - \boldsymbol{z}^*\| \le \frac{1}{2} \min_{i \in \mathrm{supp}(\boldsymbol{z}^*)} z_i^*,$$

we can lower bound $\widehat{z}_{t,i}$ by $\frac{1}{2}\min_{i\in\mathrm{supp}(\boldsymbol{z}^*)} z_i^*$ for $i \in \mathrm{supp}(\boldsymbol{z}^*)$. This must happen when $t$ is large by Theorem 4, but the entire analysis will then depend on a potentially large "asymptotic" constant. Therefore, even though we know that asymptotically, DOGDA has linear convergence, getting a concrete rate as VOMWU and DOMWU is still an open question.

Another direction is to follow the analysis of VOGDA, which gives a linear convergence rate by Corollary 5. However, in the analysis of VOGDA, Wei et al. [2021] implicitly use the fact that $\Phi^{\mathrm{van}}$ is $\beta$-smooth, that is,

$$D_{\Phi^{\mathrm{van}}}(\boldsymbol{z}, \boldsymbol{z}') \le \frac{\beta}{2}\|\boldsymbol{z} - \boldsymbol{z}'\|^2,$$

for some $\beta > 0$. In fact, $\Phi^{\mathrm{van}}$ is 1-smooth. This property, unfortunately, does not hold for $\Phi_\alpha^{\mathrm{dil}}$. We believe one can still show that $\Phi_\alpha^{\mathrm{dil}}$ is $\beta$-smooth for some game-dependent $\beta$ once $\widehat{z}_t$ is sufficiently close to $\boldsymbol{z}^*$. However, this again involves the asymptotic result in Theorem 4 and may prevent us from getting a concrete rate. In summary, using the existing techniques, we met some difficulties to obtain a concrete convergence rate for DOGDA. However, DOGDA performs well in the experiments. Moreover, it reduces to VOGDA in the normal-form games and achieves linear convergence in this case. Therefore, we still believe that it is a promising direction to get a (linear) convergence rate for DOGDA in theory.