# OpenReview forum: "Last-iterate Convergence in Extensive-Form Games"
_NeurIPS.cc/2021/Conference — NeurIPS 2021 Poster_

### Official Review · Reviewer_rJS7 · 2021-07-06

**Rating:** 6
**Confidence:** 3

**Summary:**

In this paper, the authors prove the last-iterate convergence of several optimistic algorithms in two-player zero-sum extensive-form games with perfect recalls (by considering the treeplex formulation). In particular, they show that the last iterate of Dilated Optimistic Multiplicative Weight Update (DOMWU) converges geometrically when the Nash equilibrium is unique.

**Limitations And Societal Impact:**

The limitations of the work are discussed in the paper.

**Main Review:**

**Post rebuttal**
The authors have addressed my concerns, so I have raised my score accordingly. I agree with the other reviewers that proving last-iterate convergence for extensive-form games is an important contribution.

=======================================================================================================

My main concern is that the proof of Theorem 4 is wrong. The authors claim that $\mathbf{z_t}$ converges after showing that $||\mathbf{z}_{t+1}-\mathbf{z}_t||^2\rightarrow 0$, which is clearly wrong. I do not think there is a simple way to fix this error, and thus to evaluate the paper I rather focus on the remaining results (and the proofs thereof seem to form the real technical part of the work).

Without diving into the appendix, I assume all the other results presented in this work are correct (at least the arguments presented in 6.2 and 6.3 seem quite reasonable). Then, I believe the real contribution is in proving that DOMWU converges geometrically to the Nash equilibrium when the latter is assumed to be unique. This provides an efficient algorithm for finding Nash equilibrium of an EFG. The introduction of these algorithms is without doubt also an important contribution if they had never been considered by the community before. (They are quite straightforward to derive from existing methods, but the effectiveness of these methods is demonstrated in the experimental section.)

I also encourage the authors to explain in more detail the treeplex formulation. I did not find it easy to relate it to the usual definition of an extensive-form game. Providing an example, perhaps in the appendix for those games considered for the experiments, will surely help.


**Time Spent Reviewing:**

5

---

> ### Author Response · Authors · 2021-08-10
> **Response to Reviewer rJS7**
>
>
>
> **The proof of Theorem 4 is incorrect**
>
> We thank the reviewer for pointing out the bug of the proof of Theorem4.  Clearly, $\\|{\boldsymbol{z}}\_{t+1}−{\boldsymbol{z}}\_t\\| \\to0$ alone does not imply the convergence of the entire sequence.  However, our final statement is still correct under an additional condition that the regularizer $\\psi$ is continuously differentiable on the entire domain (see proof below), which is indeed the case for VOGDA and DOGDA. We do not know how to fix it when $\\psi$ is only continuously differentiable in the interior of the domain, which is unfortunately the case for the other two entropy-based algorithms.  However, note that we already provide finite-time last-iterate convergence for these two algorithms in Theorems 6 and 7 (albeit under the uniqueness assumption), which, as you pointed out, are the key technical contributions of this work.
>
> To prove the above claim, we first extract the display below Eq. (5) in the paper: $$
>         \Theta\_1\ge\frac{15}{32}\sum^{T-1}\_{t=1}\\|\widehat{\boldsymbol{z}}\_{t+1}- \boldsymbol{z}\_{t}\\|^2 +\\|\boldsymbol{z}\_t- \widehat{\boldsymbol{z}}\_t\\|^2\ge\frac{15}{64}\sum^{T-1}\_{t=2}\\|\boldsymbol{z}\_{t}-\boldsymbol{z}\_{t-1}\\|^2.
> $$
>
> Similar to the last inequality, we also have $\Theta\_1\ge  \frac{15}{64}\sum^{T-1}\_{t=1}\\|\widehat{\boldsymbol{z}}\_{t+1}-\widehat{\boldsymbol{z}}\_{t}\\|^2$ since $2\\|\widehat{\boldsymbol{z}}\_{t+1}- \boldsymbol{z}\_{t}\\|^2 +2\\|\boldsymbol{z}\_t- \widehat{\boldsymbol{z}}\_t\\|^2\ge \\|\widehat{\boldsymbol{z}}\_{t+1}-\widehat{\boldsymbol{z}}\_{t}\\|^2$. Therefore, we conclude that $\\|\boldsymbol{z}\_t- \widehat{\boldsymbol{z}}\_t\\|$, $\\|\boldsymbol{z}\_{t+1}- \boldsymbol{z}\_{t}\\|$, and $\\|\widehat{\boldsymbol{z}}\_{t+1}- \widehat{\boldsymbol{z}}\_{t}\\|$ all converge to $0$ as $t$ approaches infinity.
> On the other hand, since the sequence $\{\boldsymbol{z}\_1, \boldsymbol{z}\_2, \ldots,\}$ is bounded, by the Bolzano-Weierstrass theorem, there exists a convergent subsequence, which we denote by $\{\boldsymbol{z}\_{i\_1}, \boldsymbol{z}\_{i\_2}, \ldots,\}$.  Let $\boldsymbol{z}\_{\infty}=\lim\_{\tau \to \infty}\boldsymbol{z}\_{i\_{\tau}}$. By $\\|\widehat{\boldsymbol{z}}\_{t}-\boldsymbol{z}\_{t}\\|\to 0$ we also have $\boldsymbol{z}\_{\infty}=\lim\_{\tau \to \infty}\widehat{\boldsymbol{z}}\_{i\_{\tau}}$. Now, using the first-order optimality condition of $\widehat{\boldsymbol{z}}\_{t+1}$, we have for every $\boldsymbol{z}'\in\mathcal{Z}$,
> $$
> (\nabla\psi(\widehat{\boldsymbol{z}}\_{t+1}) - \nabla\psi(\widehat{\boldsymbol{z}}\_t) + \eta F(\boldsymbol{z}\_t))^\top (\boldsymbol{z}'-\widehat{\boldsymbol{z}}\_{t+1}) \geq 0.
> $$
> Applying this with $t=i\_{\tau}$ for every $\tau$ and let $\tau\to \infty$, we obtain
> $$
> \begin{align*}
>     0\le&\lim\_{\tau \to \infty}(\nabla\psi(\widehat{\boldsymbol{z}}\_{i\_{\tau}+1}) - \nabla\psi(\widehat{\boldsymbol{z}}\_{i\_{\tau}}) + \eta F(\boldsymbol{z}\_{i\_{\tau}}))^\top (\boldsymbol{z}'-\widehat{\boldsymbol{z}}\_{i\_{\tau}+1})\tag{by the first-order optimality}\\\\
>     =&\lim\_{\tau \to \infty}\eta F(\boldsymbol{z}\_{i\_{\tau}})^\top (\boldsymbol{z}'-\widehat{\boldsymbol{z}}\_{i\_{\tau}+1})\tag{by $\\|\widehat{\boldsymbol{z}}\_{t+1}- \widehat{\boldsymbol{z}}\_{t}\\|\to 0$ and the continuity of $\nabla\psi$}\\\\
>     =&~\eta F(\boldsymbol{z}\_{\infty})^\top (\boldsymbol{z}'-\boldsymbol{z}\_{\infty})\tag{by $\boldsymbol{z}\_{\infty}=\lim\_{\tau \to \infty}\boldsymbol{z}\_{i\_{\tau}}=\lim\_{\tau \to \infty}\widehat{\boldsymbol{z}}\_{i\_{\tau}}$}
> \end{align*}
> $$
> In other words, writing $\boldsymbol{z}\_{\infty}=(\boldsymbol{x}\_\infty, \boldsymbol{y}\_\infty)$ and $\boldsymbol{z}'=(\boldsymbol{x}', \boldsymbol{y}')$, we have for every $(\boldsymbol{x}', \boldsymbol{y}')$,
> $$
> (\nabla\psi(\widehat{\boldsymbol{z}}\_{t+1}) - \nabla\psi(\widehat{\boldsymbol{z}}\_t) + \eta F(\boldsymbol{z}\_t))^\top (\boldsymbol{z}'-\widehat{\boldsymbol{z}}\_{t+1})  \geq 0.
> $$
> which implies the duality gap $\max\_{\boldsymbol{y}'}\boldsymbol{x}\_\infty^\top \boldsymbol{G} \boldsymbol{y}'-\min\_{\boldsymbol{x}'}\boldsymbol{x}'^\top \boldsymbol{G} \boldsymbol{y}\_\infty\le 0$ and that $\boldsymbol{z}\_\\infty$ is a Nash equilibrium.
> Finally,
> choosing $\boldsymbol{z}^*=\boldsymbol{z}\_\infty$ in the definition of $\Theta\_{t}$ (right before Eq. (5)),
> we have $\lim\_{\tau\to\infty}\Theta\_{i\_\tau}=0$ because $\lim\_{\tau\to\infty}D_{\psi}(\boldsymbol{z}\_\infty,\widehat{\boldsymbol{z}}\_{{i\_\tau}})=0$ and $\lim\_{\tau\to\infty}\\|\widehat{\boldsymbol{z}}\_{{i\_\tau}}-\boldsymbol{z}\_{{i\_\tau}-1}\\|=0$.
> Additionally, by Eq. (5) we also have that $\lim\_{t\to\infty}\Theta\_{t}=0$ as $\Theta\_{t}$ is non-increasing.
> Therefore, we conclude that the entire sequence $\{\boldsymbol{z}\_1, \boldsymbol{z}\_2, \ldots\}$ converges to $\boldsymbol{z}\_\infty$.
>
> We hope that this convinces the reviewer that our claim in Theorem 4 still holds for a wide range of regularizers, albeit not as general as the original version. The only issue to generalize this proof to regularizers that are only continuously differentiable in the interior is that the step $\lim\_{\tau \rightarrow \infty} \nabla\psi(\widehat{\boldsymbol{z}}\_{i\_{\tau}+1}) - \nabla\psi(\widehat{\boldsymbol{z}}\_{i\_{\tau}}) = 0$ does not necessarily hold.
> We do not know how to fix this at the moment, but we believe that for DOMWU and VOMWU, one can still prove the claim by adopting the more complicated techniques in [Hsieh et al., 2021, Theorem 8], where the authors show a similar asymptotic convergence result for OMWU over simplex with a time-varying step size.
>
> **Detail on the treeplex formulation**
>
> We agree that an example of the treeplex would be helpful. We will add more explanations in the next version. However, we note that this treeplex formulation is not unusual in the literature. It is also called the sequence form, which can be found in game theory textbooks, for example, [Nisan et al., 2007, Section 3.10] (see also Page 72 for a concrete example).
>
> **References**
>
> Y.-G. Hsieh, K. Antonakopoulos, and P. Mertikopoulos. Adaptive learning in continuous games: Optimal regret bounds and convergence to nash equilibrium. Conference on Learning Theory, 2021.
>
> N. Nisan, T. Roughgarden, E. Tardos, and V. V. Vazirani. Algorithmic Game Theory. Cambridge University Press, 2007.

---

### Official Review · Reviewer_MuPP · 2021-07-06

**Rating:** 7
**Confidence:** 4

**Summary:**

This paper considers last-iterate convergence of Optimistic Online Mirror Descent in zero-sum perfect-recall extensive-form games. It includes a proof for a general case that the last iterate must convergence in the limit, and specific convergence rates for three particular choices of regularizer function based on commonly used choices of negative entropy and the Euclidean norm -- with inclusion of a fourth result that follows from prior work. The authors provide an experimental comparison of these four OOMD methods with two variants of a popular baseline algorithm, CFR.

--- after discussion ---
There was one issue raised, but there was also a proposed connection so my opinion remains positive and largely unchanged.
------

**Limitations And Societal Impact:**

Yes.

**Main Review:**

The results in this paper are novel, to the limits of my knowledge. I agree with the authors that similar results in prior work assume simplex decision sets, or standard entropy / Euclidean norm regularizer functions that are computationally inconvenient in large extensive form games. The similarity of those conclusions does not trivialize this work: they do not immediately apply to sequential decision making problems.

I found the details to be clearly laid out and reasonably easy to follow, given the paper is largely a theoretical analysis. There were only a few issues that might be addressed, listed below.

The paper does not introduce VOGDA, VOMWU, DOGDA, or DOWMU as novel algorithms. Are they new, strictly speaking, but the authors consider them obvious instantiations of existing OOMD framework using existing common regularizers? Are they previously explored from other work, which should be cited? For example, citation [16] seems at a glance to be looking at DOGDA. The current situation where the provenance is left unclear is awkward. Claim them, or give credit.

I found myself unsure of what conclusions to draw from the experimental section. Why not compare the last-iterate algorithms compared to the last-iterate solution from the CFR+ variants? What is the reader intended to draw from the CFR+ w/ RM+ line? Even more generally, is the reader intended to reach the conclusion that the theoretical analysis is empirically visible, even in some cases where uniqueness of eq'm does not hold? If so, how are they expected to do so, when the step size was too large in one experiment? Or is the reader being sold on the OOMD algorithms (which may or may not be new to this paper) as a method for solving games?

Can a reduced subset of the algorithms be run for longer on Leduc poker to use the theoretically justified step size?  Given the experiments apparently run in under an hour on standard PC hardware, getting two orders of magnitude more steps in four days seems quite reasonable.

**Time Spent Reviewing:**

8

---

> ### Author Response · Authors · 2021-08-10
> **Response to Reviewer MuPP**
>
> **Provenance of the algorithms**
>
> We are sorry for the confusion and will clarify this in the revision.  Indeed, we do not claim novelty for these algorithms since they either have been studied before or are instantiations of the standard OOMD framework using common regularizers. More specifically, DOGDA was studied in [16] and DOMWU was studied in [30]. On the other hand, VOGDA and VOMWU are direct extensions of standard algorithms to the treeplex domain, although to our knowledge they have not been explicitly considered for extensive-form games theoretically or empirically.
>
> **Conclusion regarding the experiments**
>
> We thank the reviewer for bringing up many great questions regarding the experiments. We clarify these questions below and will also add them to the next version.
> - **why not compare with the last-iterate of the CFR+ variants:** We use this version of CFR+ (regret matching+, alternating updates, and linear averaging) because this is the one considered in the original paper [42] and the most commonly used benchmark. We do not use its last-iterate because it does not always converge to a Nash equilibrium, which is also shown in Appendix C.4.
> - **the purpose of the CFR w/ RM+ line**: This is a CFR+ variant with simultaneous updates instead of alternative updates. The reason we consider this algorithm is that all of our OOMD algorithms use simultaneous updates. We introduce this additional benchmark to highlight the potential difference between these two updates. Secondly, it also adds another reference point for those familiar with the numerics of EFG-solving: on Leduc CFR+ does very well, and it may be useful to see a second baseline to better contextualize the performance of the OOMD algorithms with respect to known methods.
> - **conclusion of the experiments**: We followed your suggestion and rerun the experiments for Leduc poker with a smaller step size for VOGDA and DOMWU. With more iterations (10^5), we observed again that they exhibit linear convergence. Indeed, the main conclusion of the experiments is that OOMD algorithms tend to converge exponentially fast empirically, even though theoretically we only prove this for some cases. However, whether these algorithms can be made to beat the state-of-the-art such as CFR+ in practice clearly still requires more future investigations. We will update the paper with those additional plots.

---

> > ### Comment · Reviewer_MuPP · 2021-08-23
> > **Response to author response**
> >
> > Happy to see the issues addressed in the paper, and a patch to the issue raised by reviewer rJS7 -- good catch! -- even if it does weaken a theoretical claim. I stand by the original suggestion to accept.

---

### Official Review · Reviewer_YFCu · 2021-07-15

**Rating:** 7
**Confidence:** 3

**Summary:**

This paper appears to be the first to achieve last iterate convergence in extensive form games. Last iterate convergence results have recently been established for optimistic gradient descent algorithms on bilinear saddle point problems. This paper extends these results to extensive form games, which is a nontrivial extension. The authors provide solid theory and experiments to support the claim.


**Limitations And Societal Impact:**

This work is mainly theoretical so I think the authors have adequately addressed the societal impact.


**Main Review:**

I think this is a very strong paper. Last iterate convergence in extensive-form games has been an open problem in this field for some time, and I am happy to see it finally tackled. I wasn’t able to find an error in any of the theory. I was excited to see the very impressive performance on Kuhn poker in the experiments. I think this paper will open up a lot of future work, for example combining this approach with neural networks. I could see future versions of this method surpassing CFR as the go-to extensive form NE solver. One downside is that the techniques are exactly the same as in a recent past paper which solves bilinear constrained optimization problems but just applied to extensive form games, but from my understanding this application is nontrivial so I think the paper is still strong.


**Time Spent Reviewing:**

3

---

> ### Author Response · Authors · 2021-08-10
> **Response to Reviewer YFCu**
>
> **The techniques are exactly the same as in recent papers**
>
> We agree that the main idea of the proofs is very similar to [Wei et al. 2021]. However, as you also point out, this extension is highly nontrivial. Besides that, we also come up with important new ideas such as the strict complementary slackness in Section D.1 and a bottom-up induction in Lemma 22. We believe that these are of independent interest and may be useful for proving more results on EFGs in the future.

---

### Official Review · Reviewer_cpq1 · 2021-07-16

**Rating:** 7
**Confidence:** 4

**Summary:**

This paper studies learning algorithms in zero-sum extensive form games. While there has been a huge amount of interest in recent years in understanding the convergence of learning algorithms to Nash equilibria in continuous or normal form  zero-sum games, this has not extended as much to extensive form games. This paper makes use of prior works that allows such games to be formulated as a bilinear zero-sum games over action sets known as treeplexes that capture the sequential nature of the actions in the game. The authors then analyze the family of Optimistic Online Mirror Descent (OOMD) algorithms in such games due to their strong last-iterate guarantees in normal form games. In particular they first show that counterfactual regret minimization can diverge in extensive form games (using rock-paper-scissors as an example) before showing that OOMD algorithms with strongly convex regularizers enjoy last iterate convergence. The authors again invoke prior works to give convergence rates for vanilla GDA, give new convergence rates for vanilla optimistic multiplicative weights, and then provide a new analysis of the last-iterate convergence for a previously proposed algorithm known as Dilated Optimistic Multiplicative Weight Update which is specialized to the structure of a treeplex (and therefore actually implementable).

**Limitations And Societal Impact:**

The authors discuss some of the limitations of their work in the appendix (namely in the difficulties in provided similar rates for the dilated OGDA algorithm). Furthermore the authors discuss the benefits of the dilated algorithms over vanilla algorithms for normal form games.

**Main Review:**

This paper gives last iterate convergence (and evens some rates) in extensive form games, and though these results are not necessarily surprising given the formulation of extensive form games as bilinear games over polytopes, these results are--- to the best of my knowledge--- new and relevant to the literature. Furthermore, the paper is well written and easy to follow, and the analysis is well explained.

One main place where this paper could be improved I believe is in the preliminaries. Though I understand the space constraints require the authors to cut down some exposition, I think this paper would benefit greatly from a more in depth introduction to extensive form games and how they differ from normal form games (rather than jumping straight into the bilinear formulation and treeplexes). Indeed, the lengthy definition of a treeplex seems less important than introducing the concept of an extensive form game (especially since--- as the authors note--- this class of games is much less common in the learning literature). Furthermore, the actual structure of a treeplex (i.e., branching, parent variables, etc.) does not seem to be integral to the analysis. Also, the definition and exposition surrounding dilated optimistic GDA can also be moved to an appendix since there are no tangible results associated to the algorithm as far as I can see. I think these changes would make the paper much more self-contained.

Another question I have is in the constants in the convergence rates. Do your results capture the dependence on the `depth' (for lack of a better word) of the treeplex, or is this quantity completely hidden in the constants? I'm curious about how the computation of a Nash equilibrium in extensive form games scales with the number of stages in the game.



**Time Spent Reviewing:**

4 hours

---

> ### Author Response · Authors · 2021-08-10
> **Response to Reviewer cpq1**
>
> **Improvement on the preliminary**
>
> We thank the reviewer for the valuable suggestion on the preliminaries and the structure of the paper. We agree there is a room for improvement. We will add more background on extensive-form games and more intuitive introduction on the treeplex formulation in future versions. We note that in the proof sketch in the main text, we hide most details involving the structure of treeplex (i.e., branching, parent variables, etc.)  for simplicity. However, we do use it heavily in the complete analysis (see, for example, Lemma 22 in the appendix).Regarding DOGDA, we do have the tangible result that it has last-iterate convergence according to Theorem 4 (which must be amended, as pointed out by reviewer rJS7).
>
> **The dependence on the “depth”**
> Our convergence rates do not have an explicit dependence on the depth. However, our rate for DOMWU heavily depends on $\\|\boldsymbol{\\alpha}\\|_\infty$, where ${\boldsymbol{\alpha}}$ is the weight parameter such that $\Psi\_{\boldsymbol{\alpha}}^{\text {dil }}$ is 1-strongly convex. That ${\boldsymbol{\alpha}}$ itself is known to have an exponential dependence on the depth of the game [30], but that dependence seems tobe necessary, and is a feature inherent to the $\Psi\_{\boldsymbol{\alpha}}^{\text {dil }}$ regularizer (see the tightness claim in [30] and also their proof). Note also that in most games, the size of the treeplex itself is exponential in the depth of the game, so even just updating the iterates or computing gradients is linear in treeplex size, and thus exponential in game depth.

---

### Decision · Program_Chairs · 2021-09-27

**Decision:**

Accept (Poster)

**Comment:**

The reviewers were unanimously positive about the paper, and the rebuttal addressed all major concerns that remained; we are happy to recommend acceptance. Please make sure to implement the issues identified in the review process in the updated version of the paper.